# STING-dependent sensing of self-DNA drives silica-induced lung inflammation

Sulayman Benmerzoug[1,2], Stéphanie Rose[1,2], Badreddine Bounab[1,2], David Gosset[4], Laure Duneau[1,2], Pauline Chenuet[3], Lucile Mollet[4], Marc Le Bert [1,2], Christopher Lambers[5], Silvana Geleff[6], Michael Roth [7], Louis Fauconnier[3], Delphine Sedda[1,2], Clarisse Carvalho[1,2], Olivier Perche[1,2,8], David Laurenceau[1,2,8], Bernhard Ryffel[1,2], Lionel Apetoh [9], Ahmet Kiziltunc[10], Hakan Uslu[11], Fadime Sultan Albez[12], Metin Akgun[12], Dieudonnée Togbe[3] & Valerie F.J. Quesniaux [1,2]

Silica particles induce lung inflammation and fibrosis. Here we show that stimulator of interferon genes (STING) is essential for silica-induced lung inflammation. In mice, silica induces lung cell death and self-dsDNA release in the bronchoalveolar space that activates STING pathway. Degradation of extracellular self-dsDNA by DNase I inhibits silica-induced STING activation and the downstream type I IFN response. Patients with silicosis have increased circulating dsDNA and CXCL10 in sputum, and patients with fibrotic interstitial lung disease display STING activation and CXCL10 in the lung. In vitro, while mitochondrial dsDNA is sensed by cGAS-STING in dendritic cells, in macrophages extracellular dsDNA activates STING independent of cGAS after silica exposure. These results reveal an essential function of STING-mediated self-dsDNA sensing after silica exposure, and identify DNase I as a potential therapy for silica-induced lung inflammation.

[1] CNRS, UMR7355, Orleans 45071, France. [2] Experimental and Molecular Immunology and Neurogenetics, University of Orleans, Orleans 45071, France. [3] Artimmune SAS, rue Buffon, 45071 Orleans, France. [4] Center for Molecular Biophysics, CNRS UPR4301, Orleans 45071, France. [5] Department of Thoracic Surgery, Lung Transplantation Program, Vienna AT1090, Austria. [6] Clinical Institute of Pathology, Medical University of Vienna, Vienna 1090, Austria. [7] Pulmonary Cell Research & Pneumology, Department Biomedicine, University& University Hospital Basel, Basel 4000, Switzerland. [8] Genetics Department, Regional Hospital Orleans, Orleans 45100, France. [9] INSERM, U1231, Dijon 21000, France. [10] Department of Biochemistry, Atatürk University School of Medicine, Erzurum 25240, Turkey. [11] Department of Clinical Microbiology, Atatürk University School of Medicine, Erzurum 25240, Turkey. [12] Department of Pulmonary Medicine, Atatürk University School of Medicine, Erzurum 25240, Turkey. These authors contributed equally: Dieudonnée Togbe, Valerie F. J. Quesniaux. Correspondence and requests for materials should be addressed to V.F.J.Q. (email: quesniaux@cnrs-orleans.fr)

Originally associated with mining and stone industry, new causes of silicosis include denim sand blasting[1–5], and the handling of frac sand for shale gas industry[6]. Indeed, drilling and fracking processes produce fine particles, such as silica particles that are retained in the lungs of silica-exposed workers and urban residents, and may lead to severe lung damage, silicosis, or idiopathic pulmonary fibrosis[7,8]. Silicosis is a chronic progressive fibrotic lung inflammation associated with increased cancer, tuberculosis, and chronic obstructive pulmonary disease[9].

Inhaled crystalline silica affects several cell types, including macrophages, dendritic cells neutrophils, fibroblasts, and epithelial cells, leading to cell activation, inflammation, and oxidative stress[10–13]. Phagocytosis of crystalline silica induces lysosomal damage and efflux of intracellular potassium, which leads to NLRP3 inflammasome activation and IL-1-dependent inflammatory response with subsequent fibrosis[9,14–17]. Here, we hypothesized that airway silica exposure induced cell death, release of self-DNA, and triggered the stimulator of interferon genes (STING) pathway. The STING signaling pathway is activated by dsDNA or cyclic-dinucleotides (cDN) such as c-di-AMP, either through direct binding to cDNs or via DNA sensors[18]. Among them, cyclic GMP–AMP synthase (cGAS), IFN-γ-inducible protein 16 (IFI16), its mouse ortholog (IFI204), or DEAD-box helicase 41 (DDX41) trigger type 1 IFN response through STING, TANK-binding kinase 1 (TBK1), and IFN regulatory factor 3 (IRF3) activation.

Here, we show that STING is activated in the lung tissue from patients with fibrotic interstitial lung disease (ILD). Mouse airway exposure to silica microparticles induces cell death, self-dsDNA leakage, and inflammatory response through STING-dependent type 1 IFN signaling and downstream CXCL10 expression. Interestingly, patients with silicosis exhibit increased circulating self-dsDNA, together with increased concentrations of CXCL10 in sputum. DNA is central as degradation of extracellular DNA by DNase I in vivo prevents the STING pathway activation and silica-induced lung inflammation. DNA sensor cGAS contributes to STING activation after silica in vivo exposure. Thus, STING, by sensing dsDNA from dying cells plays a key role in silica-induced lung inflammation and DNase I treatment abrogates this response.

## Results

**Airway silica induces self-dsDNA release and IFN-I response.** Silica microparticles intratracheal exposure induced self-dsDNA release in the bronchoalveolar space (Fig. 1a). This was accompanied by the overexpression of STING (*Tmem173*) and cGAS (*Mb21d1*) genes in the lungs 7 days (Fig. 1b, c) and 4 weeks after silica exposure (Supplementary Fig. 1a–c). The overexpression of *Irf3*, *Irf7*, and *Ifnβ* genes was in line with an engagement of the STING pathway at 4 weeks (Supplementary Fig. 1d, e). Silica induced type I IFNs and downstream CXCL10 expression either at day 7 or 4 weeks following exposure (Fig. 1d, e; Supplementary Fig. 1e,f). At 4 weeks, silica induced inflammatory cytokines, including IL-1β, TNF, CXL10, and IFN-γ in the lung, together with lung inflammation (Supplementary Fig. 1f and g). The levels of extracellular dsDNA in the bronchoalveolar lavage fluid (BALF) correlated with *Ifnα* and *Ifnβ* overexpression in silica-exposed WT mice on day 7 (Fig. 1f), and already at 24 h (Supplementary Fig. 1h–j), suggesting that this early dsDNA release might induce type I IFN expression. We verified in TLR2/TLR4 double-deficient mice that the effect of silica was not due to endotoxin or other PAMPs (Supplementary Fig. 1k–n). We then assessed the contribution of type I IFN and IL-1 pathways by exposing IFNAR- and IL-1R1-deficient mice to silica.

Interestingly, the level of dsDNA in the BALF was not affected by the absence of functional IFNAR pathway, indicating that silica exposure induced dsDNA release upstream of type I IFN response. However, the inflammatory response in response to silica was reduced in the absence of either type I IFN or IL-1 signaling pathways, including neutrophil recruitment and CXCL10 release, while IFN-αβ production was unaffected by the absence of IL-1 signaling (Supplementary Fig. 1o–p), confirming the role of the type I IFN pathway for mounting the inflammatory response to silica.

We then asked whether self-dsDNA leakage was the result of cell death induced by silica exposure. Indeed, cell death was highly increased in the lung of WT mice exposed to silica microparticles (Fig. 1g), correlating with dsDNA release in the airways (Fig. 1h). We further characterized the source of self-dsDNA released in the BALF of WT mice exposed to silica. Nuclear DNA (nDNA) and mitochondrial DNA (mtDNA) were increased in the BALF after silica exposure (Fig. 1i). To better characterize the mode of cell death induced by silica microparticles in the lung, we first assessed caspase 3 cleavage as a marker of apoptosis. We show that silica exposure induced caspase 3 cleavage in WT mice, indicative of silica-induced apoptosis (Fig. 1j, k). The contribution of pyroptosis and necroptosis in silica-induced cell death was assessed by immunoblots of cleaved gasdermin D (GSDMD) and phosphorylated mixed lineage kinase domain-like pseudokinase (MLKL) (Fig. 1j). Indeed, silica exposure induced the cleavage of GSDMD and phosphorylation of MLKL in the lung of WT mice (Fig. 1j, l, m). In summary, airway silica exposure induces lung cell death through apoptosis, necroptosis, and pyroptosis, with nuclear and mitochondrial self-dsDNA release, triggering type I IFN pathway activation.

**DNase I prevents airway silica-induced STING activation.** To address the role of released endogenous dsDNA in silica-induced lung inflammation, we treated mice exposed to silica particles with DNase I (Fig. 1n). DNase I treatment reduced the amount of dsDNA recovered in the bronchoalveolar space 7 days post silica exposure (Fig. 1o). We show that silica induced the expression of STING protein and the activation of STING pathway, as documented by STING phosphorylation and dimerization, phosphorylation of TBK1 and IRF3 in the lung of exposed mice (Fig. 1p, q). STING dimer immunoblot was visualized under seminative conditions, while reducing conditions disrupted the dimeric form of STING (Fig. 1r). In vivo DNase I treatment prevented silica-induced STING expression, phosphorylation and dimer formation, TBK1 and IRF3 phosphorylation in lung homogenates (Fig. 1p, q), as well as STING overexpression in the lung parenchyma (Fig. 1s), demonstrating that extracellular dsDNA is central to STING activation pathway in the airways. Silica-induced cell death was reduced by DNase I treatment, in terms of caspase 3 cleavage and MLKL phosphorylation (Fig. 1p). Further, DNase I prevented silica-induced airway inflammation with a significant decrease of lung IFN-α and IFN-β, as well as of CXCL10 release (Fig. 1t). Importantly, the recruitment of neutrophils and macrophages, and protein extravasation in the BALF seen 7 days post silica exposure were largely reduced by DNase I treatment (Fig. 1u). Therefore, our data identify endogenous dsDNA as an integral actor of STING pathway activation and lung inflammation induced by silica exposure.

**Circulating dsDNA and CXCL10 in silicosis patients.** Although mining and stone industry were traditional sources of silica exposure, new sources such as sand blasting in the denim industry raise the expected prevalence of silicosis to over 96%[3,19].

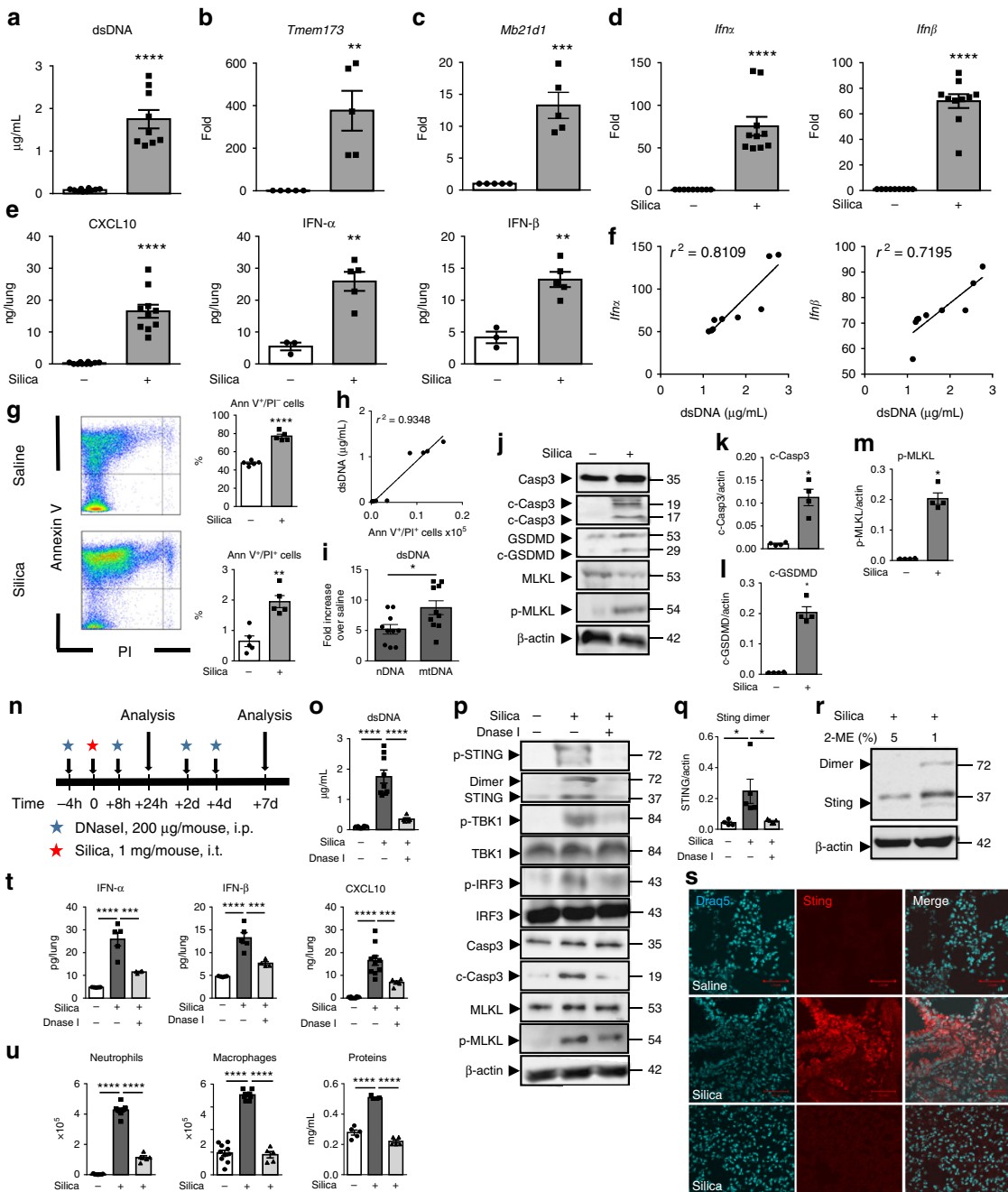

**Fig. 1** Self-dsDNA release is central to silica-induced lung inflammation and type I IFN response. **a–m** Silica microparticles (1 mg/mouse, i.t.) or saline were administered in WT mice and parameters were analyzed on day 7. **a** Concentration of extracellular dsDNA in the acellular fractions of bronchoalveolar lavage fluid (BALF). **b–d** *Tmem173, Mb21d1, Ifnα, and Ifnβ* transcripts in the lungs and normalized to *Gapdh* expression. **e** Lung CXCL10 determined by ELISA and IFN-αβ proteins quantified by multiplex immunoassay. **f** Correlation between extracellular dsDNA concentrations and type I IFN gene expression. **g** Annexin V/PI flow cytometry analysis pre-gated on singlet cells. **h** Correlation between dead cells and extracellular dsDNA. **i** Increased nuclear DNA (nDNA) and mitochondrial DNA (mtDNA) in the BALF after silica exposure. **j** Immunoblots of caspase 3 (Casp3), cleaved caspase 3 (c-Casp3), gasdermin D (GSDMD), MLKL, and phosphorylated-MLKL (p-MLKL), normalized to β-actin. Immunoblot quantifications of (**k**) c-Casp3, (**l**) c-GSDMD, and (**m**) p-MLKL. **n–u** Silica microparticles administered with DNase I (200 μg/mouse, i.p.) as indicated (**n**). **o** Extracellular dsDNA in BALF acellular fraction. **p** Lung immunoblots of phospho-STING, STING, phospho-TBK1, TBK1, phospho-IRF3, IRF3, caspase 3 (Casp3), cleaved caspase 3 (c-Casp3), MLKL, and phosphorylated-MLKL (p-MLKL), with β-actin as a reference. **q** Quantification of STING dimer, relative to β-actin. **r** STING immunoblots under 5% reducing (left) or 1% 2-mercaptoethanol seminative conditions (2-ME; right). **s** Lung confocal images of DNA dye Draq5 (cyan) and STING (red). Bars, 20 μm. **t** Pulmonary IFN-αβ and CXCL10. **u** Neutrophils, macrophages, and protein extravasation in the BALF. *p < 0.05, **p < 0.01, ***p < 0.001, ****p < 0.0001 (Kruskal–Wallis test followed by Dunn post test). The correlation analysis was nonparametric (Spearman correlation). Data are mean ± SEM representative of two independent experiments (**a–f**: mice per group: n = 5 or 10 (WT NaCl), n = 8 or 10 (WT silica); **g–h**: mice per group: n = 5 (WT NaCl), n = 5 (WT silica); **i–n**: mice per group, n = 5 or 10 (WT NaCl), n = 5 or 10 (WT silica), and n = 5 (WT silica + DNase I)). Immunoblots representative of n = 6 samples from three independent experiments (**j**, **p**), or n = 8 samples from two independent experiments (**r**), quantified in bargraphs with n = 4. Each symbol represents an individual mouse. Source data are provided as a Source Data file

Unique characteristics of these patients include their young age, a definite start of exposure, an intense exposure for ca. $4 \pm 2$ years, and a rapid progression of the disease, often fatal[3,19]. Here, we investigated a cohort of 21 young men suffering from silicosis after exposure to silica in denim industry in the mid-2000s (Table 1). We hypothesized that silica might have caused airway cell damage and release of self-DNA able to trigger the type I IFN response, as in the silica murine model. Interestingly, patients with silicosis exhibited increased plasma levels of dsDNA (Fig. 2a). Further, in their sputum, the concentration of CXCL10 was significantly higher than in healthy controls (Fig. 2b). Thus, in silicosis patients, a previous intense exposure to silica leads to high self-dsDNA circulating levels and CXCL10 response.

**Activated STING in fibrotic interstitial lung disease.** Silica particles can induce fibrotic ILD[8] and we next asked whether STING pathway is activated in this pathology. To address this question, we analyzed pneumonectomy samples from ILD patients undergoing lung transplantation. We show for the first time that the STING pathway is activated in the lung of ILD patients, as seen by STING overexpression, phosphorylation and dimer formation, TBK1 and IRF3 phosphorylation, and downstream CXCL10 production (Fig. 2c–e), correlating with epithelium damage (Fig. 2f). Interestingly, STING and CXCL10 expression were reduced (Fig. 2c–e) in the lung of ILD patients undergoing cortisone therapy (Table 2).

In vitro, silica exposure induced self-dsDNA release in human PBMCs, and this correlated with *IFNβ* overexpression (Fig. 2g–i) and CXCL10 production (Fig. 2j). STING was highly over-expressed in human PBMCs exposed to silica (Fig. 2k). This was similar to the response elicited by transfected cyclic dinucleotide c-di-AMP, a cDN that signals through STING to induce type I IFN response via a direct binding to STING protein. Silica-induced STING phosphorylation and dimerization, together with phosphorylation of TBK1 and IRF3 in PBMCs, were similar to cDN (Fig. 2l). To identify the source of self-dsDNA, we investigated cell death in silica-exposed PBMCs. Indeed, PBMCs stimulated with silica displayed a double Annexin V$^+$/PI$^+$-stained population, indicating that most cells were dead (Fig. 2m). Further, the level of Annexin V$^+$/PI$^+$ PBMCs correlated with self-dsDNA release and *IFNβ* expression (Fig. 2n). To investigate whether self-dsDNA drove STING-mediated type I IFN response, silica-exposed human PBMCs were treated with DNase I, to reduce extracellular self-dsDNA accumulation in vitro (Supplementary Fig. 2). DNase I– treatment of human PBMCs did not affect the Annexin V$^+$/PI$^+$ population, but decreased extracellular self-dsDNA concentration (Supplementary Fig. 2a–c). Degradation of extracellular self-dsDNA by DNase I reduced STING activation, as seen through the reduction of STING expression, phosphorylation and dimerization, *IFNβ* expression, and CXCL10 production in human PBMCs exposed to silica (Supplementary Fig. 2d–g).

**Table 1 Silicosis patient clinical characteristics**

| Patient | Age | First silica exposure year | Exposure duration (months) | FEV1 (% predicted) | FVC (% predicted) | FEV1/ FVC ratio | Chest X-ray severity score | Chest X-ray large opacities | Smoking history (Pack years) |
|---|---|---|---|---|---|---|---|---|---|
| C1 | 38 | NA | | | | | | | 15 |
| C2 | 43 | NA | | | | | | | 5 |
| C3 | 29 | NA | | | | | | | 2 |
| C4 | 27 | NA | | | | | | | 0 |
| C5 | 29 | NA | | | | | | | 2 |
| C6 | 40 | NA | | | | | | | 18 |
| C7 | 38 | NA | | | | | | | 10 |
| C8 | 40 | NA | | | | | | | 9 |
| C9 | 42 | NA | | | | | | | 15 |
| C10 | 28 | NA | | | | | | | 0 |
| P1 | 29 | 2003 | 72 | 83 | 83 | 86 | 8 | - | 8 |
| P2 | 36 | 1996 | 48 | 62 | 93 | 56 | 11 | A | 15 |
| P3 | 35 | 2000 | 60 | 42 | 68 | 53 | 12 | C | 7 |
| P4 | 28 | 2000 | 60 | 44 | 90 | 40 | 11 | A | 0 |
| P5 | 34 | 2002 | 48 | 65 | 73 | 75 | 12 | - | 9 |
| P6 | 31 | 2001 | 60 | 58 | 76 | 66 | 10 | B | 20 |
| P7 | 38 | 1996 | 24 | 92 | 102 | 75 | 11 | - | 5 |
| P8 | 34 | 1996 | 84 | 73 | 91 | 84 | 9 | - | 25 |
| P9 | 37 | 2000 | 60 | 64 | 78 | 68 | 12 | C | 5 |
| P10 | 36 | 2004 | 24 | 96 | 110 | 74 | 11 | - | 7 |
| P11 | 36 | 2001 | 24 | 78 | 87 | 76 | 9 | - | 10 |
| P12 | 34 | 1999 | 41 | | | | 8 | - | 5 |
| P13 | 38 | 2003 | 24 | 78 | 95 | 69 | 7 | - | 40 |
| P14 | 49 | 2000 | 60 | 76 | 102 | 60 | 5 | - | 40 |
| P15 | 33 | 2001 | 12 | 92 | 91 | 85 | 9 | - | 17 |
| P16 | 36 | 1997 | 72 | 39 | 57 | 58 | 12 | B | 0 |
| P17 | 30 | 2000 | 60 | 53 | 73 | 62 | 12 | B | 6 |
| P18 | 29 | 2003 | 24 | 66 | 83 | 69 | 10 | A | 15 |
| P19 | 33 | 2001 | 48 | 42 | 48 | 73 | 12 | C | 10 |
| P20 | 35 | 2000 | 60 | 38 | 71 | 45 | 12 | C | 5 |
| P21 | 34 | 2000 | 36 | 18 | 33 | 46 | 12 | C | 1 |

All 21 patients with silicosis (P1–P21) were males, as were the 10 healthy controls (C1–C10)
Radiological severity score according to International Labor Organization (ILO) small profusion subcategories (1–12, 12 represents the most extensive disease, which is 3/ + , highest profusion of small opacities subcategory). A large opacity is defined as an opacity having the longest dimension exceeding 10 mm. The size increases from A to C and reflects the severity of radiological involvement FEV1, forced expiratory volume at 1.0 s; FEV1%, ratio of FEV1 to FVC, forced vital capacity; FEV1% predicted, FEV1% of the patient divided by the average FEV1% in the equivalent population; NA: not applicable in the healthy controls

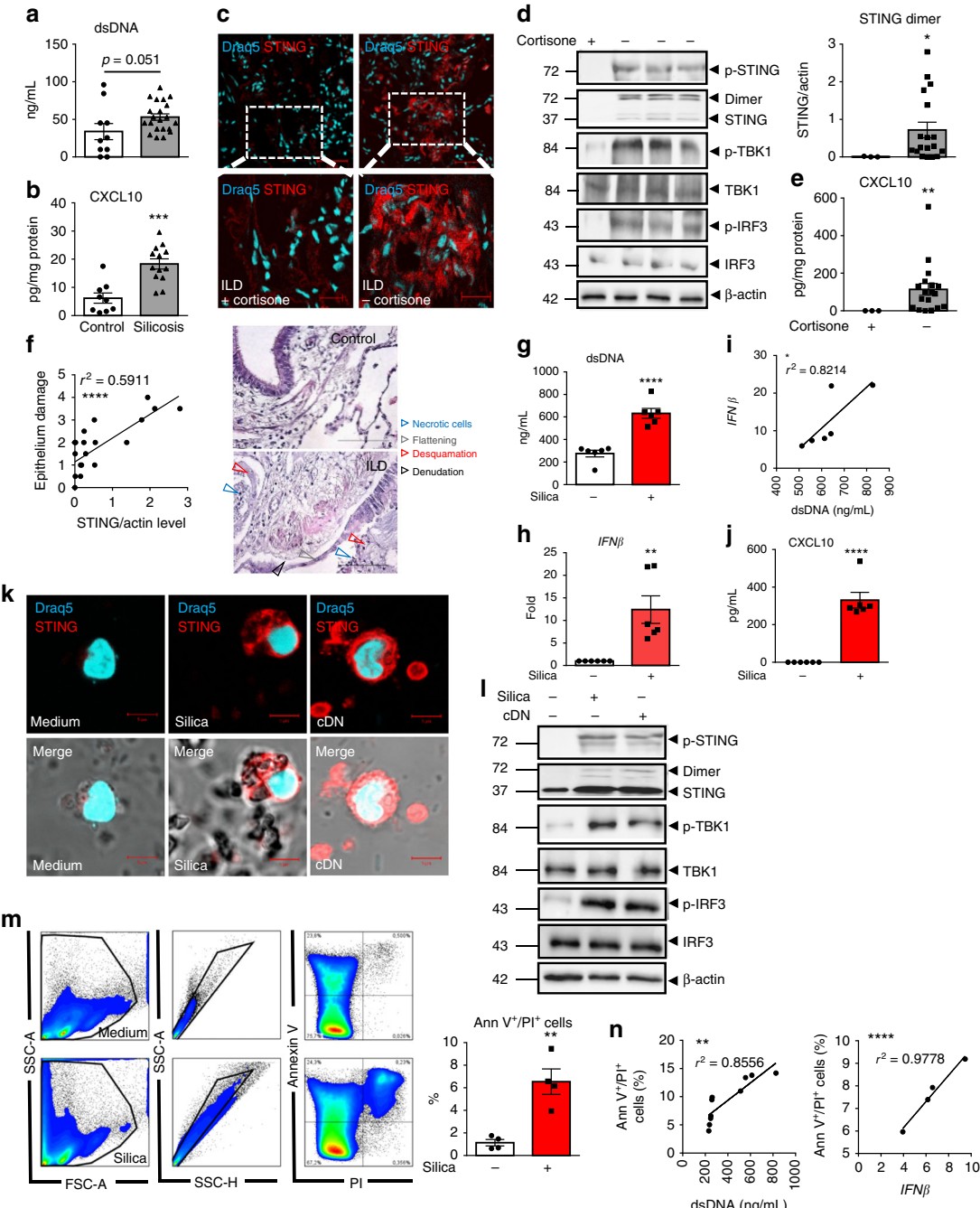

**Fig. 2** Circulating DNA and CXCL10 in the airways of silicosis patients, STING, and type I IFN pathway activation in ILD patient lungs. **a**, **b** Presence of (**a**) dsDNA in the plasma and (**b**) CXCL10 in the sputum of patients with silicosis (ILO 5–12, see Table 1), as compared with healthy individuals (ILO 0). **c**–**f** Human lung tissue samples of ILD patients treated or not with cortisone (see Table 2). **c** Confocal images of DNA dye Draq5 (cyan) and STING (red). Bars, 50 μm. Bars, 20 μm for zoomed regions. **d** Immunoblots of phosphorylated-STING (p-STING), STING, TBK1, phospho-TBK1, IRF3, phospho-IRF3, and β-actin. STING dimers quantification. **e** CXCL10 levels. **f** Correlation between STING dimers and epithelial damage scored on human histological tissue sections for necrotic cells, desquamation or denudation, and flattening of the epithelial barrier (indicated by arrows). Control lung tissue (patient G) and fibrotic area (patient D). Bars, 100 μm. **g**–**n** Human PBMCs were stimulated with silica microparticles (250 μg/mL) for 18 h, transfected with c-di-AMP (6 μg/mL) or unstimulated. **g** Extracellular dsDNA in cell supernatant, **h** *IFNβ* transcript, **i** correlation between *IFNβ* transcripts and released dsDNA. (**j**) CXCL10. (**k**) Confocal images of DNA dye Draq5 (cyan) and STING (red). Bars, 5 μm. **l** Immunoblots of phospho-STING, STING, phospho-TBK1, TBK1, phospho-IRF3, IRF3, and β-actin. **m** Flow cytometry annexin V/PI analysis gated on singlet cells. **n** Correlations between dead cells and extracellular dsDNA or *IFNβ* transcripts. *$p < 0.05$, **$p < 0.01$, ***$p < 0.001$, ****$p < 0.0001$ (Student's t test). The correlation analysis was nonparametric (Spearman correlation). Data are mean ± SEM representative of two independent experiments. **a**, **b** Plasma samples from $n = 10$ healthy controls and $n = 21$ silicosis patients; **b** sputum samples from $n = 9$ healthy controls and $n = 13$ silicosis patients. **c**–**f** c: Confocal image representative of three patients; **d**: immunoblots shown from left to right for ILD patients H, G, C, A (see Table 2), representative of $n = 3$ cortisone-treated ILD patients and $n = 18$ untreated-ILD patients, from three independent experiments, shown in **d**–**f**. **g**–**k** PBMC from $n = 6$ donors per group. **l** Immunoblots representative of $n = 6$ donors from two independent experiments. **m**–**n** Four donors per group, and are representative of at least two independent experiments with similar results (**g**–**n**). Each symbol represents an individual. Source data are provided as a Source Data file

**Table 2 Clinical characteristics of patients with fibrotic interstitial lung disease**

| Patient | Age | Gender | Treatment | FEV1 l | FEV1% predicted | Diagnosis | Smoking history: pack years/ years of smoking cessation |
|---|---|---|---|---|---|---|---|
| A | 63 | Male | Esbriet | 1.93 | 52.5 | Fibrotic ILD | 25 PY/15 Y cess |
| B | 40 | Female | No | No | No | Fibrotic ILD | n.d. |
| C | 48 | Male | No | n.d. | n.d. | Fibrotic ILD | 0 PY |
| D | 41 | Female | No | n.d. | n.d. | Fibrotic ILD | 0 PY |
| E | 57 | Male | Esbriet | 1.5 | 43 | Fibrotic ILD | 35 PY/7 Y cess |
| F | 65 | Male | Cortisone 25 mg | 1.1 l | 39 | Fibrotic ILD | 70 PY/ 6Y cess |
| G | 63 | Male | Esbriet | 1.8 | 48 | Fibrotic ILD | 30 PY/12Y cess |
| H | 60 | Male | Cortisone 7.5 mg | 2.3 | 64 | Fibrotic ILD | 30 PY/15Y cess |
| I | 54 | Female | n.d. | n.d. | n.d. | Fibrotic ILD | n.d. |
| J | 68 | Male | No | n.d. | n.d. | Fibrotic ILD | 0 PY |
| K | 52 | Female | Ofev | 1.6 | 58 | Fibrotic ILD | 35 PY/2Y cess |
| L | 60 | Male | No | 2.4 | 60 | Fibrotic ILD | 40 PY/5Y cess |
| M | 56 | Male | Esbriet | 1.51 | 43 | Fibrotic ILD | 0 PY |
| N | 52 | Male | No | 2.7 | 70 | Fibrotic ILD | 45 PY/2Y cess |
| O | 65 | Male | No | 1.65 | 57 | Fibrotic ILD | 25 PY/20Y cess |
| P | 59 | Male | No | 1.6 | 45 | Fibrotic ILD | 0 PY |
| Q | 50 | Male | Ofev | 1.73 | 435 | Fibrotic ILD | 20 PY/12Y cess |
| R | 68 | Male | Ofev | 1.3 | 41.4 | Fibrotic ILD | 0 PY |
| S | 63 | Male | Esbriet | 1.99 | 53 | Fibrotic ILD | 30 PY/3Y cess |
| T | 65 | Male | Ofev | 1.48 | 38 | Fibrotic ILD | 10 PY/15Y cess |
| U | 59 | Male | Cortisone 5 mg | 0.93 | 26.2 | Fibrotic ILD | 10 PY/35Y cess |

*FEV1* forced expiratory volume at 1.0 s, *FEV1%* ratio of FEV1 to FVC, forced vital capacity, *FEV1% predicted* FEV1% of the patient divided by the average FEV1% in the equivalent population, *ILD* interstitial lung disease, *PY* pack year, *Y cess* years of smoking cessation, *n.d.* not determined

Thus, STING pathway is activated in the lung of ILD patients, and in vitro silica exposure induced cell death, self-dsDNA release, STING activation, and type I IFN response in human PBMCs. In addition, DNase I treatment shows that self-dsDNA is central for STING-mediated type I IFN response to silica.

**Self-DNA sensing by cGAS/STING is key for silica response**. We next investigated whether self-dsDNA sensing was specific of STING activation pathway after in vivo silica exposure. As TLR9 is involved in self-DNA recognition and type 1 IFN expression[20], we tested the susceptibility of TLR9-deficient mice (TLR9$^{-/-}$) to silica microparticles. TLR9$^{-/-}$ mice responded to silica exposure similarly to WT mice in terms of cell recruitment, protein leak, IFN-β, and inflammatory cytokines release, although IFN-α protein was reduced in TLR9$^{-/-}$ mice (Fig. 3a, b).

Since STING is activated in the lung of silica-exposed mice (Fig. 1p–s), we next investigated how central is STING activation in silica-induced lung inflammation. STING$^{-/-}$ mice exhibited a reduction of all parameters of silica-induced lung inflammation, including neutrophil and macrophage recruitment in the airways, total protein extravasation, CXCL10, IL-1β, and TNF (Fig. 3a), recapitulating the phenotype of IFNAR$^{-/-}$ mice to silica exposure (Supplementary Fig. 1o-p). Silica-induced IFN-α and IFN-β were reduced in STING$^{-/-}$ mice (Fig. 3b), as were the phosphorylation of TBK1 and IRF3 (Fig. 3c), while dsDNA levels were unchanged in the bronchoalveolar space (Fig. 3d).

As cGAS is one of the main DNA sensors associated with STING activation, we next assessed the contribution of cGAS in silica-induced lung inflammation. cGAS$^{-/-}$ mice exhibited reduced inflammatory cell recruitment and protein extravasation in response to silica (Fig. 3a). CXCL10, IL-1β, and TNF and type I IFNs (Fig. 3a, b) were also reduced in cGAS$^{-/-}$ mice 7 days after silica exposure. The upregulation of STING (*Tmem173*) transcripts seen in WT mice 7 days after silica exposure was further increased in cGAS$^{-/-}$ mice, while cGAS (*Mb21d1*) transcript upregulation was not affected by the absence of STING (Fig. 3b). STING pathway is essential for the early response to silica, as STING$^{-/-}$ mice were protected 24 h post silica exposure, while

the contribution of cGAS was less pronounced in cGAS$^{-/-}$ mice at this early time point (Supplementary Fig. 3a), suggesting the contribution of other DNA sensors. Interestingly, the level of dsDNA in the BALF 24 h post silica exposure was not affected by the absence of functional STING pathway, confirming that silica induced dsDNA release upstream of STING and type I IFN responses (Supplementary Fig. 3a). STING pathway was still crucial at day 35 post silica exposure, as type I IFN response was abolished in the absence of either STING or cGAS (Supplementary Fig. 3b). Lung inflammation was also decreased in STING- and cGAS-deficient mice, with reduced cell infiltration and granuloma 35 days post exposure (Supplementary Fig. 3c). Our findings indicate that STING drives lung inflammation at both early and late phases of silica exposure, while the role of cGAS DNA sensor was less prominent early after silica exposure.

**Silica-induced inflammatory cell death depends on cGAS/ STING**. DNA sensing might differ according to DNA source and localization, and we next sought to identify the cellular source of self-DNA in the airways after silica exposure. We characterized the inflammatory cell populations recruited to the lung and their cellular fate 7 days after silica exposure. Neutrophils, and to a lesser extent interstitial macrophages, and dendritic cells were increased in the lung after silica exposure, while alveolar macrophages were unaffected (Supplementary Fig. 4). The source of self-dsDNA might include dying/dead cells. Using Annexin V/PI staining, we quantified early apoptotic (Ann V$^+$/PI$^-$) versus dead cells (Ann V$^+$/PI$^+$) (Fig. 4a–e). Silica exposure induced cell death of interstitial and alveolar macrophages and tenfold more of neutrophils, as well as early apoptosis in DCs from WT mice, while this response was reduced in the lung of STING$^{-/-}$ and cGAS$^{-/-}$ mice (Fig. 4a–e). Interestingly, silica exposure also induced cell death in epithelial and non-hematopoietic, resident lung cells independently of the cGAS/STING pathway (Supplementary Fig. 5). This is in line with the fact that self-dsDNA release was not affected by the absence of the cGAS/STING pathway (Fig. 3d). We next questioned whether STING could be involved in apoptosis, necroptosis, and/or pyroptosis-related cell

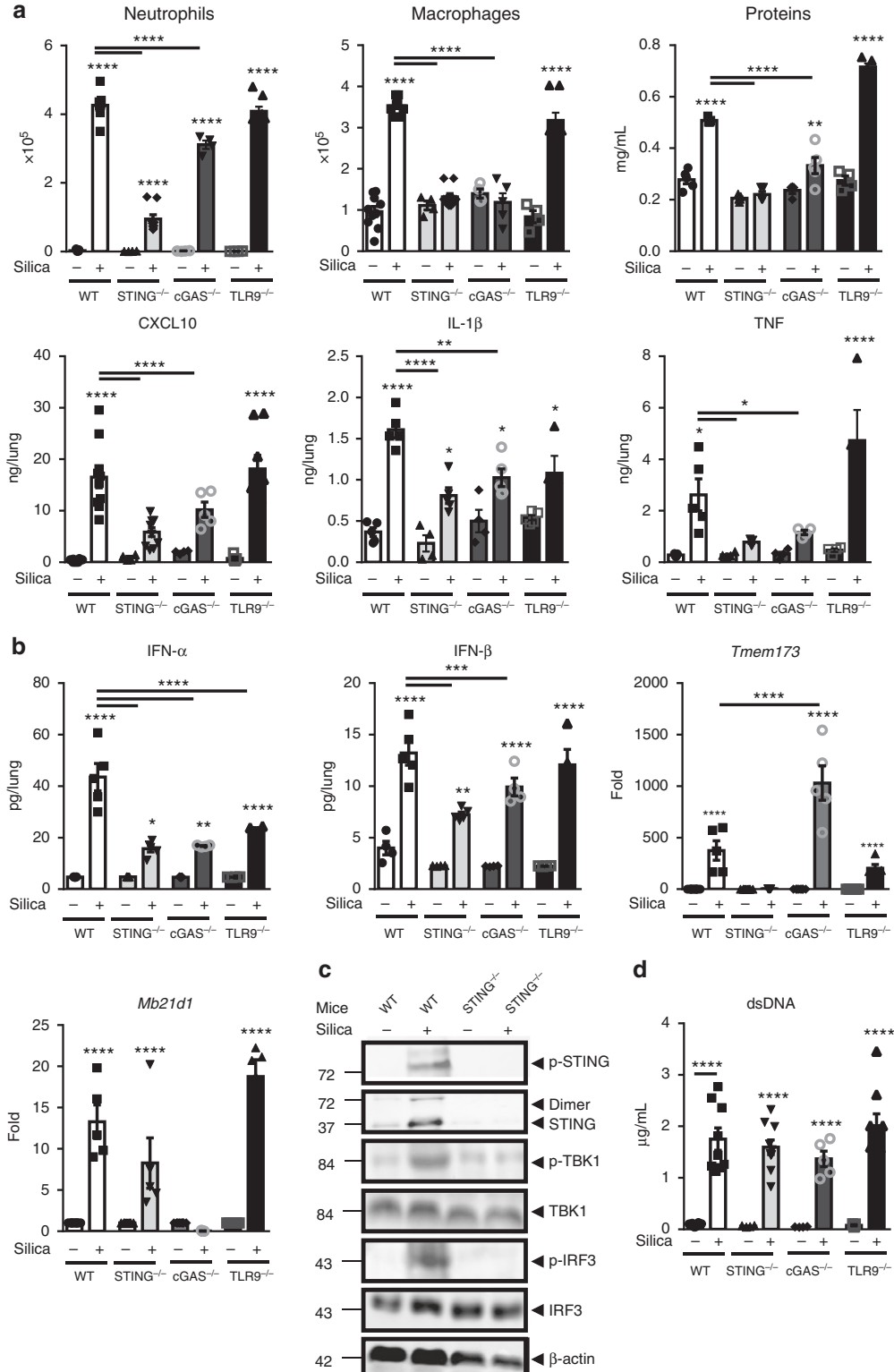

**Fig. 3** DNA sensors STING and cGAS are essential, while TLR9 is dispensable, for silica-induced lung inflammation. Silica microparticles (1 mg/mouse, i.t.) or saline vehicle were administered in WT, STING$^{-/-}$, cGAS$^{-/-}$, and TLR9$^{-/-}$ mice and the different parameters were analyzed on day 7. **a** Neutrophils, macrophages, and protein extravasation were measured in the BALF and the lung levels of CXCL10, IL-1β, and TNF were determined by ELISA. **b** Pulmonary IFN-α and IFN-β protein concentrations determined by multiplex immunoassay, and *Tmem173* and *Mb21d1* transcripts measured by real-time PCR. **c** Immunoblots of STING/IRF3 axis in the lung of WT and STING$^{-/-}$ mice, including phospho-STING, STING, phospho-TBK1, TBK1, phospho-IRF3, and IRF3, with β-actin as a reference. **d** Concentration of extracellular dsDNA in the acellular BALF fraction. *$p < 0.05$, **$p < 0.01$, ****$p < 0.0001$ (Kruskal–Wallis test followed by Dunn post test). Data are presented as mean ± SEM and are representative of three (**a**, **b**, **d**) and two (**c**) independent experiments. (**a**, **b**, **d**): mice per group: $n = 10$ (WT NaCl), $n = 8$ (WT silica), $n = 4$ (STING$^{-/-}$ NaCl), $n = 7$ (STING$^{-/-}$ silica), $n = 4$ (cGAS$^{-/-}$ NaCl), $n = 5$ (cGAS$^{-/-}$ silica), $n = 5$ (TLR9$^{-/-}$ NaCl), $n = 6$ (TLR9$^{-/-}$ silica). Immunoblots are representative of $n = 6$ samples from two independent experiments (**c**). Each symbol represents an individual mouse. Source data are provided as a Source data file

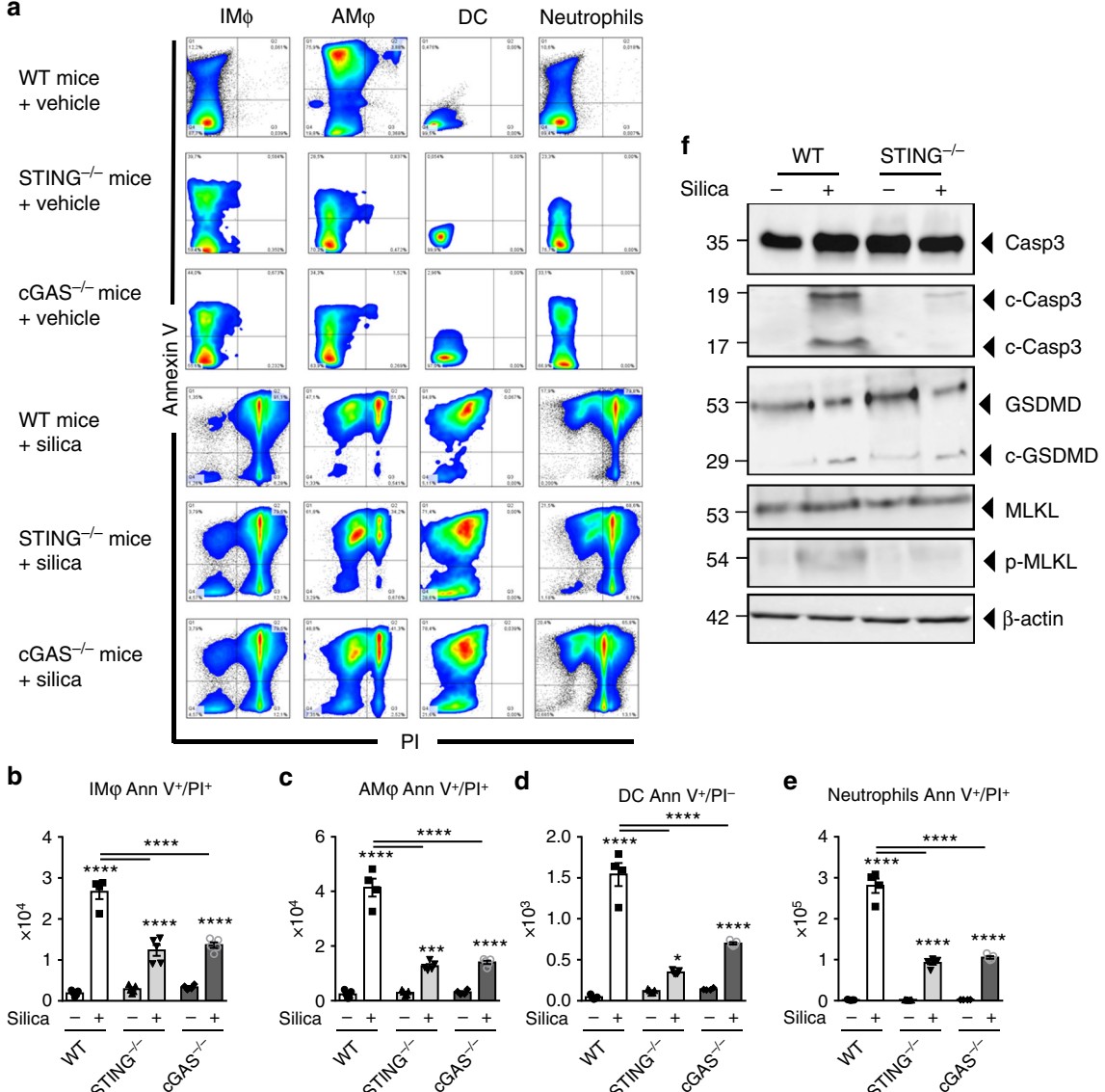

**Fig. 4** Reduced silica-induced inflammatory cell death in the absence of cGAS/STING. Silica microparticles (1 mg/mouse, i.t.) or saline vehicle were administered in WT, STING$^{-/-}$, and cGAS$^{-/-}$ mice and the different parameters were analyzed on day 7, as in Fig. 3. **a** Flow cytometry representative dot blots showing Annexin V/PI staining of F4/80$^+$CD11c$^-$ interstitial macrophages (IM), F4/80$^+$CD11c$^+$ alveolar macrophages (AM), CD11c$^+$F4/80$^-$ dendritic cells (DC), and Ly6G$^+$F4/80$^-$ neutrophils among CD45$^+$CD11b$^+$ cells in WT, STING$^{-/-}$, and cGAS$^{-/-}$ mice. Bargraphs of the Annexin V/PI-stained cells among **b** interstitial macrophages, **c** alveolar macrophages, **d** DCs, and **e** neutrophils, expressed as absolute cell number per lung. **f** Immunoblots of caspase 3, cleaved caspase 3, gasdermin D, MLKL, and phospho-MLKL in the lung of WT and STING$^{-/-}$ mice with β-actin as a reference. *$p < 0.05$, ***$p < 0.001$, ****$p < 0.0001$ (Kruskal–Wallis test followed by Dunn post test). Data are presented as mean ± SEM and are representative of two independent experiments (**a–e**: mice per group: $n = 5$ (WT NaCl), $n = 4$ (WT silica), $n = 5$ (STING$^{-/-}$ NaCl), $n = 5$ (STING$^{-/-}$ silica), $n = 4$ (cGAS$^{-/-}$ NaCl), $n = 5$ (cGAS$^{-/-}$ silica)). Immunoblots are representative of $n = 6$ samples from two independent experiments (**f**). Each symbol represents an individual mouse. Source data are provided as a Source Data file

death. Indeed, silica induced apoptosis and necroptosis in a STING-dependent manner, since caspase 3 cleavage and MLKL phosphorylation were highly reduced in the lung of STING$^{-/-}$ mice (Fig. 4f). By contrast, silica-induced gasdermin D cleavage was still present in STING$^{-/-}$ mice, showing that STING is not essential for pyroptosis cell death after silica exposure (Fig. 4f). We further addressed the involvement of the type I IFN pathway in early silica-induced cell death. Lung cell death was partially reduced in IFNAR$^{-/-}$ and STING$^{-/-}$ mice 24 h post silica exposure (Supplementary Fig. 6a–c), although there was already a STING/IFN-I-independent release of dsDNA in the airways (Supplementary Fig. 6d). At this early stage, caspase 3 cleavage and MLKL phosphorylation occurred in the absence of the IFN-I

pathway (Supplementary Fig. 6e). These results show that, beside causing cGAS/STING-independent epithelial cell death, silica induces necroptosis and apoptosis in a STING-dependent manner in the lung 7 days post silica exposure.

**STING modulates NLRP3 expression after silica exposure**. The NLRP3 inflammasome has been recognized as an innate immune signaling receptor important for mediating cell responses and inflammation to crystals, such as silica[14,16,21–23]. To dissect the effect of STING adaptor molecule on NLRP3 expression and pyroptosis, NLRP3-deficient (NLRP3$^{-/-}$) and gasdermin D-deficient (GSDMD$^{-/-}$) mice were exposed to silica

(Supplementary Fig. 7 and 8). The absence of NLRP3 led to a reduction of cell death in interstitial and alveolar macrophages but also of apoptosis in DCs after in vivo silica exposure (Supplementary Fig. 7a–c), which however did not translate into a reduction of dsDNA release in the airways (Supplementary Fig. 7d). Silica-induced lung inflammation was reduced in NLRP3$^{-/-}$ mice as compared with WT mice (Supplementary Fig. 7e–f). Interestingly, type I IFN response was also reduced in the absence of NLRP3 after in vivo silica exposure (Supplementary Fig. 7g, h). In order to understand how type I IFN signaling is reduced in NLRP3$^{-/-}$ mice, we evaluated STING/TBK1 signaling in the absence of NLRP3 in vivo. Indeed, *Tmem173* transcripts, STING protein expression, phosphorylation, and dimerization but also phosphorylation of TBK1 were partially reduced in the absence of NLRP3 in vivo (Supplementary Fig. 7i–k). Conversely, STING is important for NLRP3 expression, as seen by the decrease of NLRP3 protein and gene expression in the lung of STING-deficient mice after silica exposure (Supplementary Fig. 7j, l, m). Although inflammatory cell death was reduced in GSDMD$^{-/-}$ mice, dsDNA release and lung inflammation were not affected 7 days post silica exposure (Supplementary Fig. 8). Thus, STING and NLRP3 pathways are linked to induce lung inflammatory response to silica.

**Silica activates ROS, cell death, and STING in macrophages**. Macrophages play a key role in silicosis and inflammatory diseases, especially via their ability to internalize microparticles, nucleic acids, and dying cell-associated intracellular content[24]. To determine the role of macrophages following silica stimulation, we first evaluated their expression of co-stimulatory molecules in response to silica in vitro. We showed that the overexpression of CD40, C86, and MHC-II after macrophage stimulation with silica or cDN was abrogated in the absence of STING (Supplementary Fig. 9a). Following silica stimulation, macrophages underwent cellular stress with reactive oxygen species (ROS) production detected by MitoSOX immunofluorescence which co-localized with mitochondria revealed by MitoTracker (Fig. 5a, b). Interestingly, the induction of ROS after silica stimulation was impaired in STING$^{-/-}$ macrophages (Fig. 5a, b). Using live in vitro video microscopy, we documented that silica microparticles present in the culture were internalized by macrophages which underwent drastic morphological changes to engulf silica particles up to 12–24 h before cell death and silica particles externalization (Supplementary online video). Indeed, there was a sharp increase in Annexin V/PI double-positive macrophages 18 h post silica exposure (Fig. 5c), together with self-dsDNA release in the cell supernatant (Fig. 5d). Macrophage cell death after silica exposure was dependent on ROS induction as it could be partially prevented in the presence of ROS scavenger (Supplementary Fig. 10). Silica exposure induced apoptosis and necroptosis in macrophages, as revealed by cleaved caspase 3 and phosphorylated-MLKL (Fig. 5e). Interestingly, in silica-exposed STING$^{-/-}$ macrophages, caspase 3 cleavage and phospho-MLKL were reduced, as compared with WT macrophages (Fig. 5e). Gasdermin D was not cleaved after in vitro silica exposure, which may reflect the lack of inflammasome activation, due to the absence of the first NFκB signal (Fig. 5e)[25]. Interestingly, macrophages stimulated with silica exhibited Draq5-positive DNA-containing cytoplasmic structures co-localized with silica microparticles (Fig. 5f). The STING signaling pathway was activated in macrophages exposed to silica, including STING phosphorylation and dimerization, and phosphorylation of TBK1 and IRF3 (Fig. 5g). Indeed, STING was essential for silica-induced macrophage death, as STING$^{-/-}$ macrophages were protected (Fig. 5c). Further, intracellular DNA associated with silica microparticles induced STING overexpression and aggregation, as seen after transfection of c-di-AMP (cDN; Fig. 5h). Thus, in vitro silica exposure induced ROS, macrophage cell death, dsDNA release, intracytosolic DNA associated with silica microparticles, and activation of the STING pathway.

**Silica induces STING-dependent apoptosis in DCs**. Dendritic cells were recently shown to sense tumor dsDNA differently to macrophages[26]. To dissect whether the STING-dependent activation mechanism was different in DCs and macrophages, we next investigated the response of DCs to silica microparticles and the role of STING in this response. In vivo, DCs recruited to the airways after silica exposure initiated a cell death program (Fig. 4a, d). In vitro, BMDCs stimulated with silica microparticles for 24 h were activated, and the overexpression of CD40, CD86, and MHC class II was reduced in STING-deficient DCs (Supplementary Fig. 6b). Mitochondrial ROS expression was increased in silica-exposed wild-type DCs, but not in STING$^{-/-}$ DCs (Fig. 6a). This oxidative stress was associated with an increase in apoptotic Annexin V$^+$/PI$^-$ and dead cells (Annexin V$^+$/PI$^+$) 18 h post silica exposure (Fig. 6b), together with extracellular dsDNA release (Fig. 6c). Silica exposure induced STING-dependent apoptosis in DCs, as revealed by cleaved caspase 3, reduced in STING$^{-/-}$ DCs, while MLKL was not phosphorylated and gasdermin D was not cleaved (Fig. 6d). To examine whether dsDNA derived from DCs could be sensed by STING, DCs were double-stained with DNA dye Draq5 and STING-specific antibody. Interestingly, DCs stimulated with silica exhibited multiple DNA-containing subcellular structures inside the cytosol that co-localized with silica microparticles. There was a strong overexpression of STING and speck formation in these cells, beyond the perinuclear region (Fig. 6e). Further, the STING signaling pathway was activated in DCs after silica exposure via STING phosphorylation and dimerization and also phosphorylation of the TBK1/IRF3 axis (Fig. 6f). Together, these results indicate that silica is an activator of the STING signaling pathway in DCs through induction of mitochondrial stress, apoptosis-mediated cell death, and self-dsDNA release.

**Extracellular DNA mediates silica macrophage activation**. Recent work has indicated that immune cells can sense self-dsDNA present in the extracellular milieu[27]. We therefore asked whether the dsDNA released upon silica exposure could activate bystander macrophages, using DNase I to degrade extracellular dsDNA (Fig. 7a, b). Extracellular self-dsDNA accumulation after macrophage exposure to silica microparticles was prevented by DNase I treatment at 18 h (Fig. 7b). DNase I-treated macrophages showed a reduced pro-inflammatory response such as CXCL10, a complete abolition of *Ifnα* and *Ifnβ* message, and IFN-αβ protein overexpression in response to silica (Fig. 7b). To assess whether self-dsDNA is the major DAMP induced by silica, we next transfected DNA from silica-exposed lung into macrophages in vitro (Supplementary Fig. 11a). DNA from silica-exposed lung, but not from vehicle-treated mice, induced TNF pro-inflammatory cytokine release, and type I IFN responses, recapitulating silica macrophage activation, similar to c-di-AMP transfection (Supplementary Fig. 11b). This response was STING-dependent as it was abrogated in STING$^{-/-}$ macrophages (Supplementary Fig. 11b).

We next addressed the relevance of extracellular DNA sensing to DCs, by treating bone marrow-derived dendritic cells with extracellular DNase I (Fig. 7a). Surprisingly, extracellular self-dsDNA degradation did not affect silica-induced IFN-α and IFN-β protein overexpression by DCs (Fig. 7c). We next verified whether transfected self-dsDNA from silica-exposed lung could

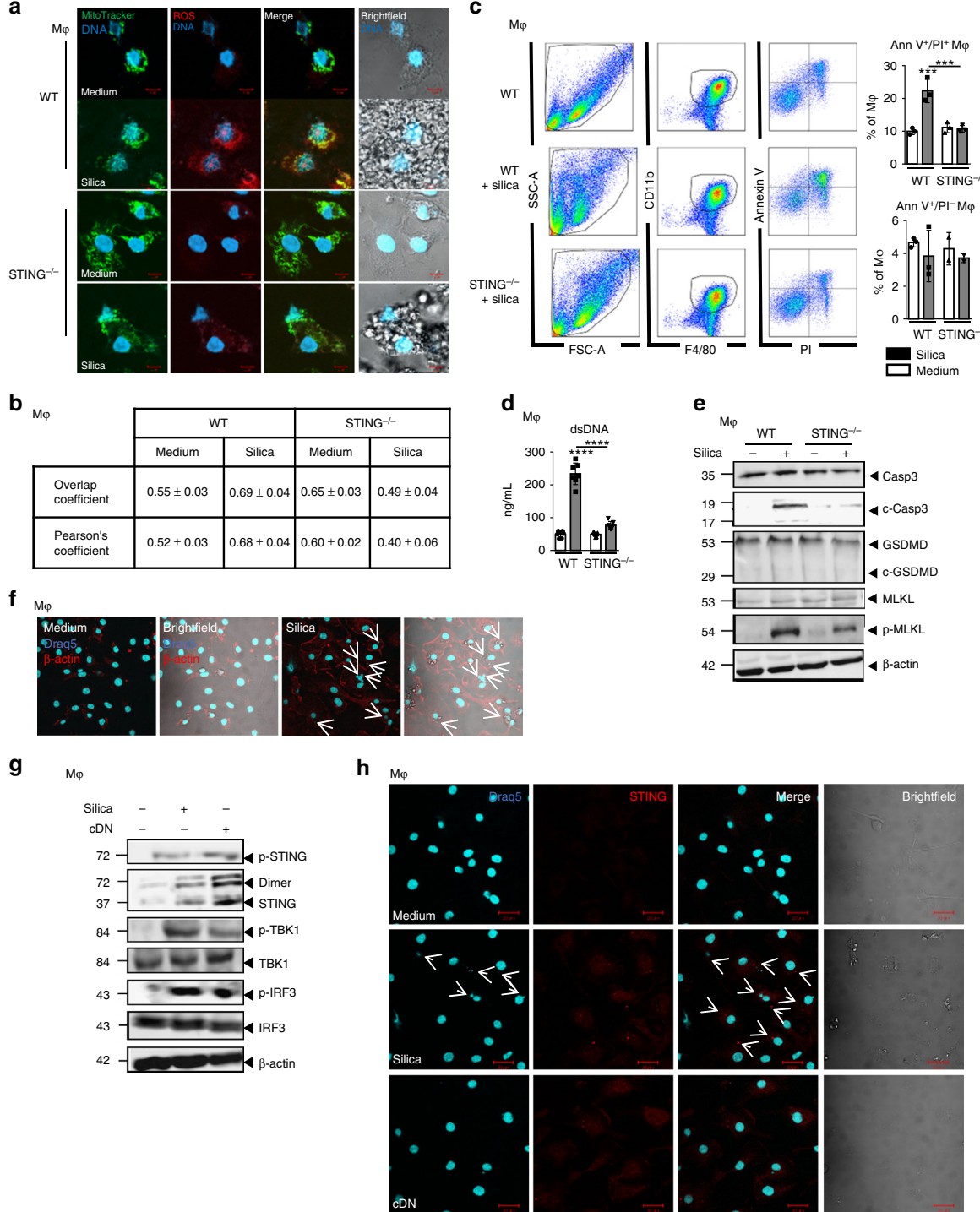

**Fig. 5** Silica induces macrophage mitochondrial stress and necrosis, dsDNA leakage, and STING pathway activation. **a, b** WT and STING$^{-/-}$ bone marrow-derived macrophages were unstimulated or stimulated with silica (250 µg/mL) for 18 h. **a** Brightfield confocal microscopy showing intracellular silica microparticles and MitoTracker (green) labeled mitochondria co-localizing with superoxide production detected by MitoSOX staining (red). Images are representative of five slides from $n = 3$ cell cultures. Bars, 5 µm. **b** Colocalization analysis of MitoTracker versus MitoSOX staining from the slides shown in **a**. Overlap coefficient and Pearson correlation coefficient were determined using ImageJ. **c** Flow cytometry Annexin V/PI staining of pre-gated singlets (SSC-A/SSC-H) and CD11b$^+$F4/80$^+$CD11c$^-$ cells. **d** Concentration of extracellular dsDNA in the culture supernatant. **e** Immunoblots of caspase 3, cleaved caspase 3, gasdermin D, MLKL, and phospho-MLKL in WT and STING$^{-/-}$ macrophages, with β-actin as a reference. **f** Confocal images of DNA Draq5 (cyan) and β-actin (red) staining in WT macrophages unstimulated or stimulated with silica (250 µg/mL) for 18 h. **g** Immunoblots of STING/IRF3 axis, including phospho-STING, STING, phospho-TBK1, TBK1, phospho-IRF3, IRF3, and β-actin as a reference, in WT macrophages stimulated as in **a–b** or transfected with c-di-AMP (6 µg/mL; cDN) for 18 h as a positive control. **h** Confocal images of DNA dye Draq5 (cyan) and STING-specific antibody (red) in WT BMDMs stimulated as in **f**. Bars, 20 µm. ***$p < 0.001$, ****$p < 0.0001$ (Kruskal–Wallis test followed by Dunn post test). Data are presented as mean ± SEM and are representative of three independent experiments with similar results (**a**, **c**, **d**, and **e**) or with n = 3 independent cultures (**b**). Immunoblots are representative of $n = 9$ samples from three independent experiments (**e**, **g**). Source data are provided as a Source Data file

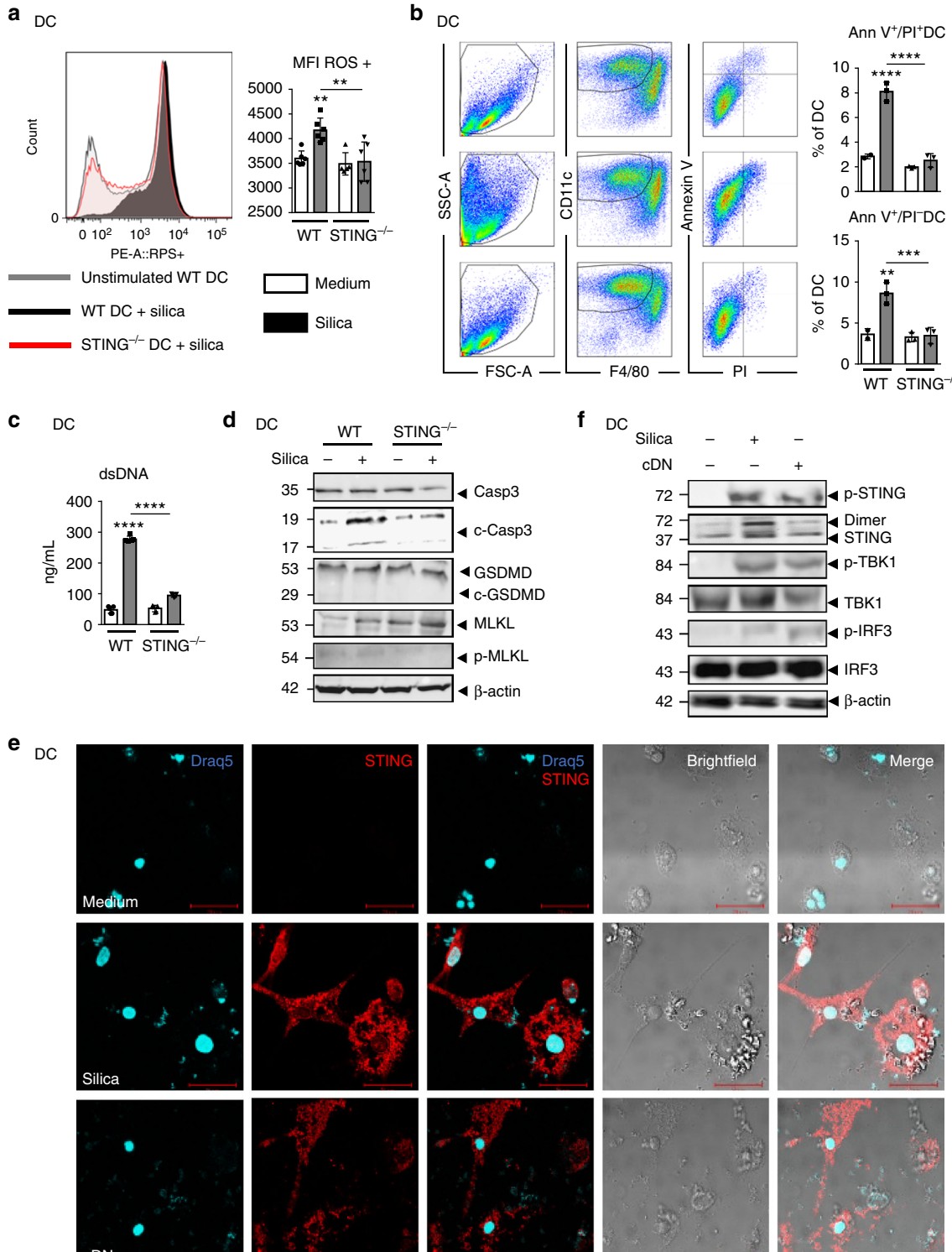

**Fig. 6** Dendritic cells respond to silica by mitochondrial and cellular stress associated with dsDNA leakage and STING activation. **a**, **b** WT and STING$^{-/-}$ bone marrow-derived DCs were unstimulated or stimulated with silica (250 µg/mL) for 18 h. **a** Flow cytometry using MitoSOX staining of mitochondria-derived superoxide production on pre-gated singlet cells (SSC-A/SSC-H) and CD11b$^+$CD11c$^+$F4/80$^-$ DCs. Histograms and mean fluorescence intensity (MFI) quantification. **b** Flow cytometry Annexin V/PI staining showing early apoptotic (Ann V$^+$/PI$^-$) versus late apoptotic/necrotic (Ann V$^+$/PI$^+$) DCs. **c** Concentration of extracellular dsDNA in culture supernatant. **d** Immunoblots of caspase 3, cleaved caspase 3, gasdermin D, MLKL, and phospho-MLKL in WT and STING$^{-/-}$ DCs with β-actin as a reference. **e** Confocal images of DNA dye Draq5 (cyan) and STING-specific antibody (red) in WT DCs stimulated as in **a**, **b** or transfected with c-di-AMP (6 µg/mL; cDN) for 18 h as a positive control. Bars, 20 µm. **f** WT DC immunoblots of STING/IRF3 axis including phospho-STING, STING, phospho-TBK1, TBK1, phospho-IRF3, and IRF3, with β-actin as a reference. *$p < 0.05$, **$p < 0.01$, ****$p < 0.0001$ (Mann–Whitney U analysis). Data are presented as mean ± SEM and are representative of two independent experiments with similar results (**a**, **c**, **d**) or with $n = 3$ independent cultures (**a**, **b**, **e**). Immunoblots are representative of $n = 9$ samples from three independent experiments (**d**, **f**). Source data are provided as a Source Data file

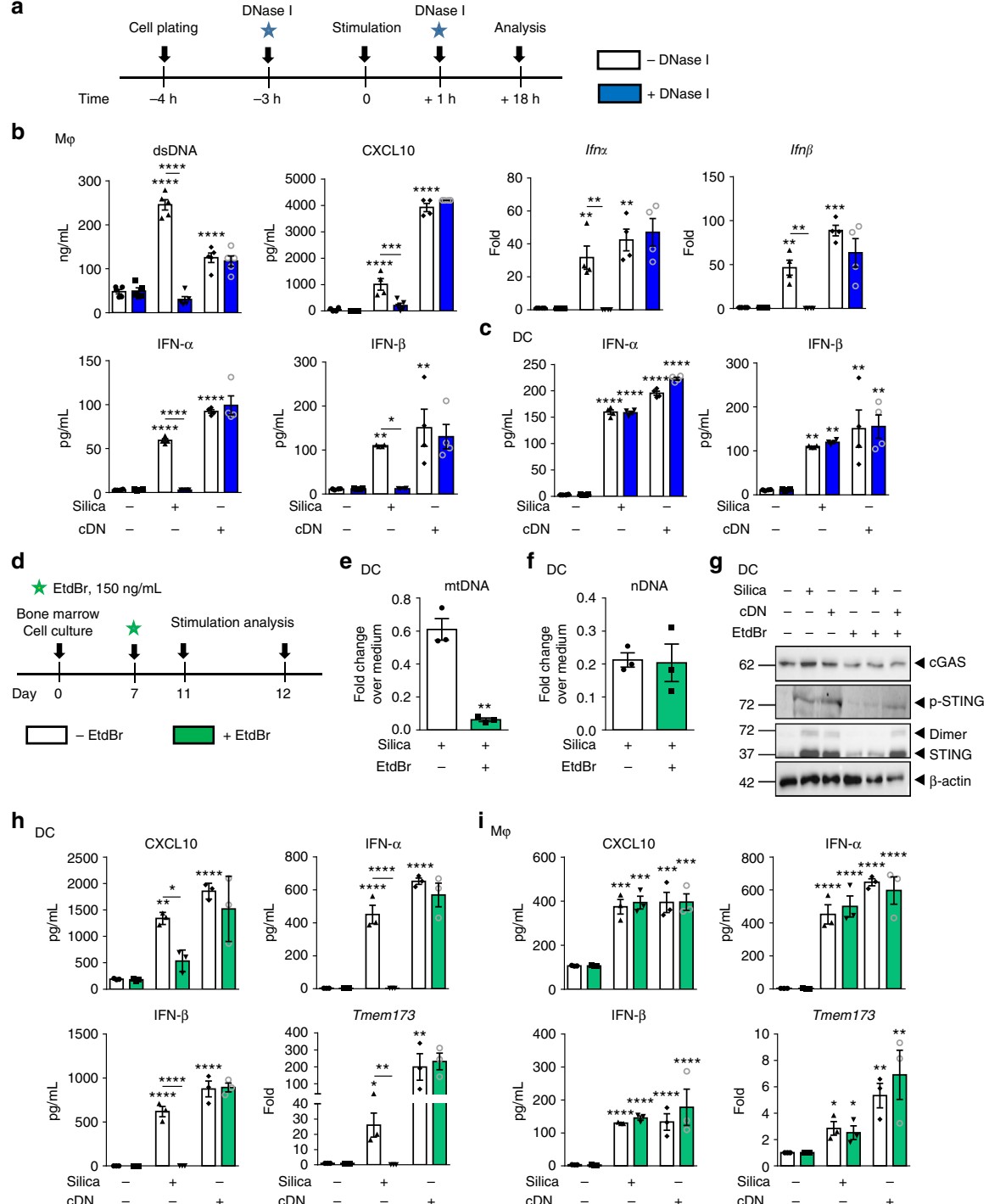

**Fig. 7** Extracellular self-dsDNA is key to silica-induced inflammatory response in macrophages, not in DCs. **a–c** Extracellular DNase I treatment (1 µg/mL) was applied 3 h prior and 1 h after silica exposure (250 µg/mL) or transfection with c-di-AMP (6 µg/mL; cDN) for 18 h in bone marrow-derived macrophages (**b**) and dentritic cells (**c**). **b** Macrophage concentration of extracellular dsDNA and CXCL10 in culture supernatant. *Ifnα* and *Ifnβ* transcripts measured by real-time PCR on cell fractions and IFN-α and IFN-β protein concentrations determined in culture supernatants by multiplex immunoassay. **c** Dendritic cell IFN-α and IFN-β protein concentrations determined in culture supernatants by multiplex immunoassay. **d–i** Mitochondrial DNA replication was inhibited using a low concentration of EtdBr (150 ng/mL) on day 7 of bone marrow cell culture. On day 11, bone marrow-derived DCs (**e–h**) and macrophages (**i**) were unstimulated, stimulated with silica (250 µg/mL), or transfected with c-di-AMP (6 µg/mL; cDN) for 18 h. Fold change of (**e**) mitochondrial DNA (mMitoF1 and mMitoR1) and (**f**) nuclear DNA (mB2MF1 and mB2MR1) in untreated versus EtdBr-treated DC exposed to silica, as compared to untreated unstimulated cells. **g** Immunoblot showing cGAS, phospho-STING, STING protein expression, and dimerization in DCs after EtdBr treatment, with β-actin as a reference. **h** CXCL10 level in DC supernatant quantified by ELISA and IFN-α and IFN-β concentrations determined by multiplex immunoassay. *Tmem173* transcripts measured by real-time PCR. **i** CXCL10 level in macrophage supernatant quantified by ELISA and IFN-α and IFN-β concentrations determined by multiplex immunoassay. *Tmem173* transcripts measured by real-time PCR. *$p < 0.05$, **$p < 0.01$, ***$p < 0.001$, ****$p < 0.0001$ (Mann–Whitney U analysis). Data are presented as mean ± SEM and are representative of two independent experiments with $n = 4$ independent cultures. Immunoblots are representative of $n = 2$ samples per condition (**g**). Source data are provided as a Source Data file

activate DCs (Supplementary Fig. 11a). Indeed, transfection of DNA from silica-exposed lung, but not from vehicle-treated mice, induced a DC type I IFN response similar to c-di-AMP together with some induction of TNF and CXCL10 (Supplementary Fig. 11c), and this response was STING dependent as it was abrogated in STING$^{-/-}$ DCs (Supplementary Fig. 11c). These data demonstrate that silica particles induce type I IFN response through extracellular self-dsDNA release in macrophages, while the response of dendritic cells to silica is not affected by extracellular dsDNA degradation.

**Silica-induced mitochondrial DNA sensed by STING in DC.** As DCs were not affected by extracellular DNA degradation (Fig. 7c), we next hypothesized that mtDNA might be the source of self-dsDNA activating STING in DCs. To test this hypothesis, we depleted DCs mitochondrial DNA using a well-established protocol with low-dose ethidium bromide (EtdBr; Fig. 7d). EtdBr intercalating into mtDNA base pairs, prevents its replication, yielding a tenfold reduction of mtDNA while sparing genomic DNA replication[28,29]. EtdBr-treated DCs displayed a drastic reduction of mtDNA without affecting nDNA (Fig. 7e, f). STING protein expression, phosphorylation, and dimerization in response to silica was strongly reduced in DCs after EtdBr treatment, while the response to control c-di-AMP (cDN) was spared (Fig. 7g). cGAS protein levels were similar after silica or cDN stimulation, without or with EtdBr treatment (Fig. 7g). In mtDNA-depleted WT DCs, type I IFNs expression was totally abolished and CXCL10 strongly decreased (Fig. 7h), while the response to c-di-AMP was unaffected, excluding a cytotoxic effect of the EtdBr treatment. The abolishment of type I IFNs in mtDNA-depleted DCs may be a consequence of the impaired cGAS/STING pathway, as we documented no expression of *Tmem173* transcript after EtdBr treatment (Fig. 7h). Moreover, the abolishment of *Tmem173* transcript in mtDNA-depleted WT DCs indicates that mtDNA is central for regulating the cGAS/STING pathway in DCs. However, mtDNA depletion did not affect the response of macrophages to silica (Fig. 7i), in line with the fact that the macrophage response was abolished by extracellular DNA depletion (Fig. 7b). Thus, mtDNA is the main source of DAMP activating STING pathway in DCs exposed to silica.

**Silica-induced DC response depends on both cGAS and STING.** To identify the DNA-binding sensor upstream of STING signaling in DCs, we next investigated the involvement of cGAS that drives type I IFN response in a STING-dependent manner. The response of DCs to in vitro silica stimulation by the induction of TNF, CXCL10, IFN-αβ message, and proteins, together with overexpression of STING and cGAS-coding genes was abolished in STING$^{-/-}$ and cGAS$^{-/-}$ DCs (Fig. 8a). DCs stimulation by c-di-AMP was dependent on both STING and cGAS pathways (Fig. 8a). Thus, in response to silica microparticles, DCs sense the endogenous dsDNA through both cGAS and STING promoting type 1 IFNs response.

**cGAS-independent macrophage response to silica.** Macrophages rely on extracellular DNA, but not on mtDNA to respond to silica exposure, however, they fail to upregulate cGAS coding gene (Fig. 8b), and we next questioned the contribution of cGAS for STING-mediated macrophage response. The increased release of TNF, CXCL10, together with *Ifnα, Ifnβ,* and STING-coding gene *Tmem173* overexpression induced by silica microparticles in wild-type macrophages was abolished in STING$^{-/-}$ macrophages, or partially reduced for TNF (Fig. 8b). Interestingly, this response was largely independent of cGAS, since cGAS$^{-/-}$

macrophages responded to silica microparticles with a release of TNF, CXCL10, and an overexpression of *Ifnα, Ifnβ*, and *Tmem173* similar to wild-type macrophages (Fig. 8b). Transfection with c-di-AMP induced a type 1 IFN response in wild-type macrophages that was abolished in STING$^{-/-}$ macrophages (Fig. 8b), while cGAS$^{-/-}$ macrophages responded to c-di-AMP, as expected (Fig. 8b). Altogether, in vitro macrophage stimulation by silica activates type 1 IFNs signaling in a STING-dependent manner, promoting inflammations, but this response is largely independent of cGAS DNA sensor.

**Silica macrophage response involves STING, DDX41, and IFI204.** We thus hypothesized that other upstream DNA sensors might mediate the response to silica and self-dsDNA in macrophages. Among DNA sensors activating STING, IFI204, the mouse ortholog of human IFI16, and the helicase DDX41 genes were overexpressed in silica-exposed macrophages (Supplementary Fig. 12a). Using siRNA (Supplementary Fig. 12b), we could show that silica-induced activation of the STING pathway, including STING dimerization and phosphorylation of TBK1 and IRF3, was partly reduced after knockdown of IFI204 and more importantly after DDX41 knockdown (Supplementary Fig. 12c), as were TNF, CXCL10, and type I IFN expression (Supplementary Fig. 12d). As expected, the response to c-di-AMP was largely dependent on DDX41[30]. IRF3 phosphorylation, which can be induced by different signaling pathways including TLR3, TLR4, or MAVS[31] was less affected by DDX41 knockdown. Thus, in macrophages in vitro, DDX41 and IFI204 DNA sensors contribute to STING-dependent type 1 IFN response to silica.

**Hematopoietic cell STING mediates silica airway response.** STING protein is expressed in endothelial and epithelial cell types, as well as in hematopoietic cells including macrophages, T cells, and DCs[32]. To study the cellular source of STING leading to silica-induced lung inflammation, bone marrow reconstituted STING$^{-/-}$ or wild-type mice were exposed to silica microparticles (Supplementary Fig. 13). The inability of STING-deficient mice to activate type I pathway in response to silica was corrected by WT hematopoietic cell reconstitution, and it was partially transferred to WT mice after reconstitution with STING$^{-/-}$ bone marrow cells. Indeed, silica-induced cell death, neutrophil recruitment in the airways, type I IFN expression, and the release of CXCL10 and TNF were reduced after reconstitution of WT mice with STING$^{-/-}$ hematopoietic cells (Supplementary Fig. 13a–k). Conversely, the reconstitution of STING$^{-/-}$ mice with WT BM restored the response to silica in terms of cell death, neutrophil recruitment in the airways, type I IFN expression, and CXCL10 and TNF release (Supplementary Fig. 13c–k). Self-dsDNA release in the BALF after silica exposure was unaffected in the chimeric mice (Supplementary Fig. 13i), similar to STING$^{-/-}$ mice (Fig. 3d). Thus, STING pathway in hematopoietic cells is crucial for silica-induced type I IFN response and lung inflammation. Nevertheless, the role of STING in stromal cells after silica exposure should not be excluded, since STING-competent stromal cells provide some CXCL10 response in WT mice reconstituted with STING$^{-/-}$ bone marrow mice, as compared with total STING deficiency (Supplementary Fig. 13g).

**Discussion**

In this study, we show that self-DNA released by lung damage plays a key role for inducing a pro-inflammatory type I IFN response after experimental silica exposure. Indeed, silica particles induced reactive oxygen species, cell death, and release of self-dsDNA correlating with the type I IFN response. We identified STING as a critical mediator for self-dsDNA sensing and type I

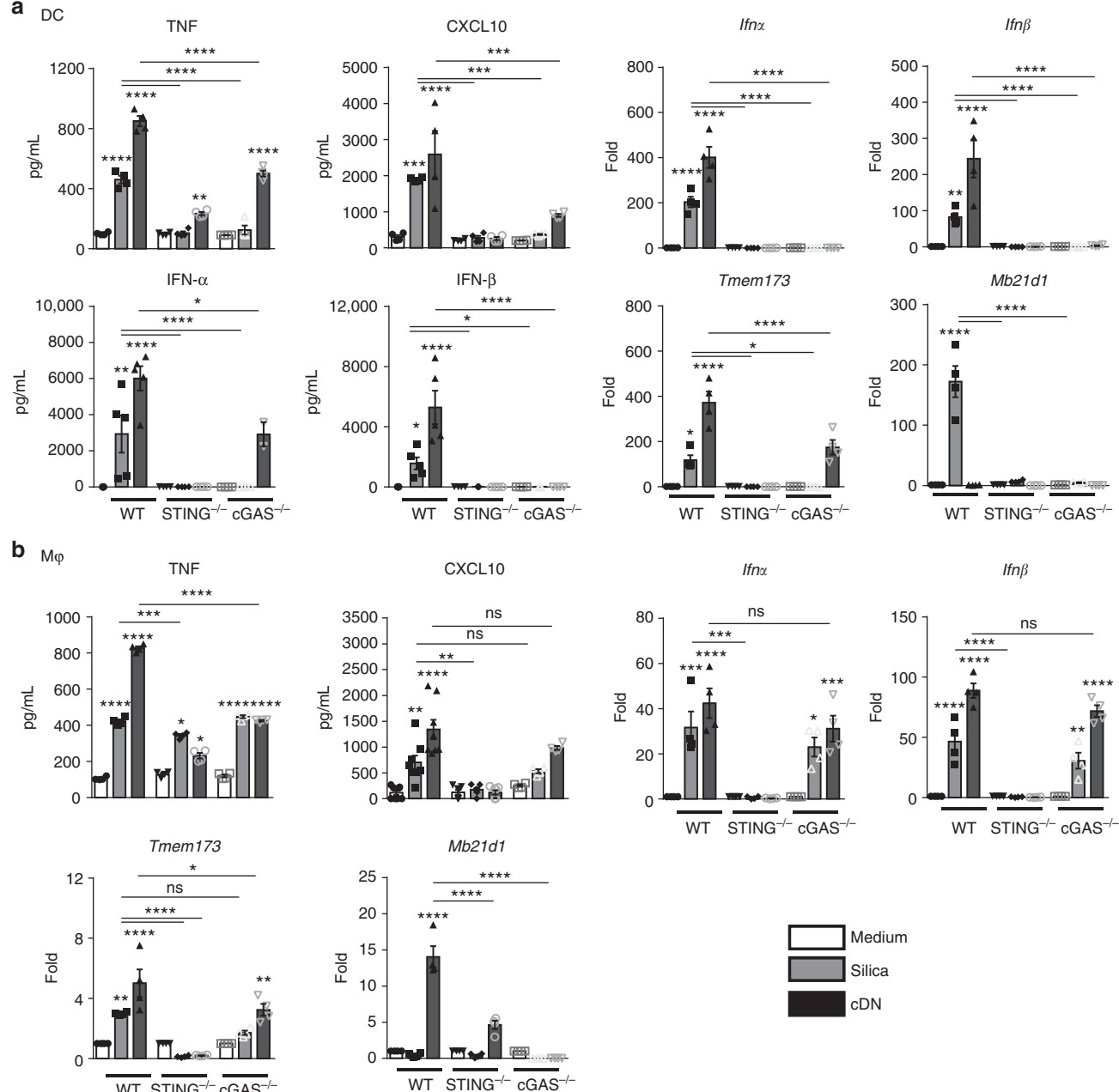

**Fig. 8** Different cGAS requirement for silica-induced STING-dependent activation in macrophages and DCs. Bone marrow-derived DCs (**a**) or macrophages (**b**) from WT, STING$^{-/-}$, and cGAS$^{-/-}$ were unstimulated, stimulated with silica (250 μg/mL), or transfected with c-di-AMP (6 μg/mL; cDN) for 18 h. **a** Levels of TNF and CXCL10 quantified by ELISA in DC supernatant. *Ifnα, Ifnβ, Tmem173,* and *Mb21d1* transcripts measured by real-time PCR on DC cell fractions. IFN-α and IFN-β protein concentrations determined in the DC supernatant by multiplex immunoassay. **b** Levels of TNF and CXCL10 quantified by ELISA in macrophage supernatant. *Ifnα, Ifnβ, Tmem173,* and *Mb21d1* transcripts measured by real-time PCR on macrophage cell fractions. ns *p* > 0.05 not significant, **p* < 0.05, ***p* < 0.01, ****p* < 0.001, *****p* < 0.0001 (Kruskal–Wallis test followed by Dunn post test). Data are presented as mean ± SEM (*n* = 6) and are representative of two independent experiments. Source data are provided as a Source Data file

IFN response upon silica exposure. Importantly, silicosis patients had increased circulating dsDNA and CXCL10 in sputum, and STING was activated in lung tissue from ILD patients. Moreover, silica induced human PBMC cell death, and activation of STING and type I IFN pathways, which were dependent on extracellular dsDNA release.

Using mice defective for different signaling pathways, we show that the pro-inflammatory type I IFN response is elicited by STING pathway activation via dsDNA released into the airways after silica in vivo exposure. We further show that silica exposure leads to activation of programmed cell death, apoptosis,

necroptosis, and pyroptosis inducing dsDNA leakage. We identify both nuclei and mitochondria as a source of released self-dsDNA activating STING in vivo after silica exposure, with an increased proportion of mtDNA in the airways. Degradation of extracellular self-dsDNA by DNase I considerably reduced lung inflammation and type I IFN response in a STING-dependent manner. In vitro DNase I treatment showed that extracellular DNA is the main source of DNA activating STING pathway and type I IFN response in macrophages. Indeed, macrophages stimulated with silica promoted a cellular stress followed by cell death and extranuclear dsDNA leakage in a STING-dependent

manner. mtDNA is another important source of self-DNA DAMP[20]. Using mtDNA inhibition by EtdBr intercalation, we excluded mtDNA as the source of DNA responsible for STING activation in macrophages after silica exposure. In DCs, mitochondria was the main source of self-DNA as mtDNA inhibition by EtdBr abolished type I IFN response, while extracellular dsDNA degradation with DNase I had no effect on silica-induced DC response. We showed that STING is more strongly activated in DCs than in macrophages, in line with the fact that self-dsDNA is more persistent inside DCs than macrophages. Indeed, in macrophages, endonucleases rapidly degrade cytosolic DNA, while DCs control mtDNA clearance via SIRPα signaling[26,33].

STING binds cDN, but whether it directly interacts with dsDNA is still a matter of debate[34,35]. The main cytosolic DNA sensors upstream of STING identified is cGAS, and other molecules contributing to cytosolic DNA sensing and type I IFN pathway activation in a STING-dependent manner include DDX41, IFI16 or mouse IFI204, DAI, RNA pol III, LRRFIP1, ZCCHC3 or meiotic recombination 11 homolog A (MRE11), IFI16 and IFI204 cooperating with cGAS, or ZCCHC3 acting as co-factor of cGAS[36–44]. Our findings indicate a role for cGAS DNA sensor for inducing type I IFN production after silica in vivo exposure. However, cGAS was dispensable for STING-mediated type I IFN response in the first 24 h post silica exposure, suggesting an early contribution of other DNA sensors. We thus investigated the DNA sensors triggered by silica exposure in vitro.

In DCs, silica exposure induced STING-dependent ROS production, cell death, and self-mtDNA release within DCs leading to type I IFN response. mtDNA engages cGAS–STING axis[20,33]. Indeed, in DCs, we identified the DNA sensor upstream of STING as cGAS, since cGAS-deficient DCs displayed a strong inhibition of type I IFN response after silica exposure. How mtDNA is released from the mitochondria to be sensed by cytosolic DNA sensors is still unresolved[20]. Several potential mechanisms were proposed: mtDNA may access the cytosol through initiation of apoptosis in a BAX and BCL-2-dependent manner[29,45], activation of mitochondrial permeability transition pore[46,47], or deficient control by transcription factor A mitochondria (TFAM)[45].

In vitro, silica-exposed macrophages display extracellular dsDNA release, not from mitochondrial origin but likely from the nucleus, leading to STING-dependent type I IFN response, independent of cGAS but mainly via DDX41 and IFI204 DNA sensors. Furthermore, the question arises as to how self-dsDNA released extracellularly could be sensed by intracellular STING adaptor molecule in macrophages. Self-dsDNA co-localized with silica microparticles in the intracellular compartments of macrophages. Extracellular DNA bound to silica is likely transported from the extracellular milieu into macrophage cytosol[48]. For instance, nucleic acid receptors could be involved in promoting extracellular DNA uptake after silica exposure. Indeed, RAGE protein has been shown to interact with DNA promoting DNA uptake by cells via the endosomal route, inducing Nf-κB activation through TLR9[49]. Further, in pathological conditions, extracellular self-dsDNA can also get access to the cytosol through Fc receptors, forming a chromatin-immune complex able to be internalized by the host cell[50]. Beside, CARGO proteins or peptides such as LL37 may contribute to dsDNA transport into the cell[51]. Micronuclei formed after silica exposure may also act as DAMPs activating type I IFN response in a STING-dependent manner. Indeed, Mackenzie et al. recently reported a role for DNA sensors in sensing micronuclei arising from genome instability or micronuclear envelope breakdown[52]. Micronuclei formation might occur in macrophages after silica exposure as free dsDNA was found next to the nucleus. The remaining open questions include whether dsDNA–silica complexes activate STING and whether dsDNA dissociates from silica inside the cell.

NLRP3 inflammasome activation occurs after silica exposure inducing apoptosis/pyroptosis and the release of the pro-inflammatory cytokine IL-1β[14,16,53]. Here, we show that STING activation by self-dsDNA sensing is an important pathway to induce lung inflammation after silica exposure, similarly to NLRP3 inflammasome, and that the pathways are linked. Indeed, Gaidt et al. recently showed that STING activation orchestrates a lysosomal cell death program engaging NLRP3 inflammasome after cytosolic DNA recognition[17]. In silica-induced lung inflammation, both STING and NLRP3 are activated eliciting a cell death program and pro-inflammatory cytokines release including type I IFN. Interestingly, NLRP3 expression seems to be dependent on STING activation since STING$^{-/-}$ mice displayed a lower expression of the inflammasome NLRP3 after silica exposure. Conversely, in the absence of NLRP3, silica-induced STING expression and type I IFN response was reduced. We could speculate that STING by inducing reactive oxygen species and cell death leads to the release of DAMPs, including K$^+$ ions and self-DNA activating the inflammasome NLRP3. Further, our results indicate that NLRP3 inflammasome could interfere with the cytosolic surveillance system for activating DNA sensor pathways.

STING adaptor molecule is highly investigated in response to bacterial, viral, or self-DNA but the cellular source of STING activating the cytosolic surveillance system is poorly documented. Indeed, STING is highly expressed in hematopoietic cells but also in resident cells such as endothelial and epithelial cells[32]. To discriminate the role of STING in hematopoietic cells versus resident cells, chimeric bone marrow reconstituted STING$^{-/-}$ mice were exposed to silica microparticles. We found that STING-competent hematopoietic cells are crucial for inducing inflammatory cell death, type I IFN response, and the release of pro-inflammatory cytokines, including CXCL10 after silica exposure. STING expression by stromal cells seems to play a minor role, which may be indirect, in response to type I IFNs through IFNAR activation, leading to CXCL10 production. Indeed, silica microparticles cause an initial stress to the airways tissue, with connective tissue breakdown, lung epithelial cell death, and dsDNA release in the airway space independently of the STING pathway, and the persistence of microcrystals in the lung can entertain this original cellular stress[12,54]. Our results indicate that this first response of lung epithelial cell death to silica microparticles is cGAS/STING-independent, while a cGAS/STING-dependent inflammatory response is induced by the released DNA, inducing ROS and further cell death.

The presence of nucleic acids in the cytosol represents a danger signal activating the cytosolic surveillance system but cells develop strategies to avoid an excessive immune response leading to autoimmunity: endo- and exonucleases such as MUS81, ERCC1-XPF, DNaseII, and TREX1[55–58]. Investigation of the nucleases present in hematopoietic and stromal cells could be relevant to appreciate a direct or indirect activation of STING by nucleic acids or cGAMP, possibly through bystander cells, as suggested by Ablasser et al.[59]

Self-DNA release including mtDNA has been implicated in various pathologies, such as infections, cancer, autoimmune, or inflammatory diseases, with so far limited implication of the STING pathway[20]. Extracellular mtDNA was recently detected in the BALF of idiopathic pulmonary fibrotic patients[60], and mtDNA concentration in plasma was identified as a marker of disease progression. However, no mechanistic link has been discussed between mtDNA release and disease. On the other hand, STING has been implicated in pediatric ILD[61], and gain-of-function mutations in *Tmem173* were identified in a new

autoinflammatory syndrome[59] (STING-associated vasculopathy with onset in infancy, SAVI), associated with ILD in five out of six patients[62]. Here, we show that silicosis patients with a history of intense silica exposure had increased circulating dsDNA and CXCL10 levels in sputum. Further, we document STING activation in the lung of patients with ILD. Airway silica exposure induced mitochondrial and nuclear self-dsDNA release that triggered STING-dependent type I IFN responses in mice. We propose that the DNA present in the airways activates STING pathway and contributes to the progressive lung inflammation.

In conclusion, the present findings establish a role for self-dsDNA released upon silica-induced injury in activating pro-inflammatory type I IFN response in a STING-dependent manner. DNase I treatment or targeting the sensing of self-dsDNA by STING may provide a therapeutic strategy to reduce lung inflammation after silica exposure and prevent the associated long-term development of silicosis, cancers, or COPD.

## Methods

**Tissue donors**. Plasma and sputum samples were obtained from patients with silicosis and healthy controls, after written informed consent and with the approval of the local ethics committee (registration number 2018/41; ethics committee, Atatürk University, Faculty of Medicine, Erzurum, Turkey). The relevant patient data are displayed in Table 1.

Lung tissue explants were obtained from patients with proven diagnosis of fibrotic ILD undergoing lung transplantation. All tissue samples were obtained after written informed consent of each patient and with the approval of the local ethical committee (EC-N° 1147/2015; ethics committee, Medical University of Vienna, Austria). The relevant patient data are displayed in Table 2.

**Mice**. Wild-type (WT) C57BL/6 mice and mice deficient for STING (STING$^{-/-}$)[58], cGAS (cGAS$^{-/-}$)[63], IFNAR (IFNAR$^{-/-}$)[64], TLR9 (TLR9$^{-/-}$)[65], IL-1R1 (IL-1R1$^{-/-}$)[66], TLR2/4 (TLR2/4$^{-/-}$)[67,68], NLRP3 (NLRP3$^{-/-}$)[69], and gasdermin D (GSDMD$^{-/-}$)[70] were bred in our specific pathogen-free animal facility at CNRS (TAAM UPS44, Orleans, France). For experiments, adult (8–12-week-old) animals were kept in ventilated cages in our animal unit and monitored daily. All animal experiments complied with the French Government animal experiment regulations and were approved by the "Ethics Committee for Animal Experimentation of CNRS Campus Orleans" (CCO) under number CLE CCO 2015-1087.

**Particles**. Crystalline silica particles (DQ12, $d_{50}$ = 2.2 μm, DMT GmbH and Co. KG, Essen, Germany) powders were sterilized by heating at 200 °C for 4 h in an autoclave[71]. The freshly prepared suspension was autoclaved at 121 °C for 30 min before use.

**In vivo animal experiments**. Crystalline silica particles (1 mg/mouse in 0.9% saline) or saline vehicle was administered intratracheally (i.t.) to mice under isoflurane anesthesia. BALF was collected and cells were counted[72]. The supernatant of the first lavage was collected after centrifugation and stored at −80 °C for dsDNA quantification (Quant-iT™ PicoGreen™). Lung tissues were fixed in 4% buffered formaldehyde overnight, paraffin-embedded, and 3-μm sections were stained with hematoxylin and eosin (HE). Lung IL-1β, TNF, and CXCL10 were quantified by ELISA (R&D systems). Protein extravasation in the BALF was measured by DC™ Protein Assay Kit (Bio-Rad®) and IFN-α and IFN-β proteins (Life Technologies®) were quantified in the lungs by Luminex Immunoassay System (MAGPIX-BIORAD®).

**Cell culture**. Blood was collected from 12 healthy volunteers after written informed consent by the "EtablissementFrançais du Sang Centre Atlantique" (n °CA-PLER-2016 045). Peripheral blood mononuclear cell (PBMC) was purified after Ficoll–Hypaque separation (GE Healthcare Life Sciences). After isolation, PBMCs were stimulated with silica microparticles (250 μg/mL), c-di-AMP (6 μg/mL) was transfected using Lipofectamine 2000 according to the manufacturer's instructions (Thermofisher Scientific) in RPMI 1640 (GIBCO, Life Technologies) supplemented with 2 mM L-glutamine, 10% heat-inactivated fetal calf serum (FCS), and 50 μg/mL of gentamicin, for 18 h at 37 °C and 5% CO$_2$.

Bone marrow-derived macrophages (BMDMs) were generated by differentiating mouse bone marrow cells for 7 days in DMEM medium (GIBCO, Life Technologies) supplemented with 10% FCS (GIBCO, Life Technologies), 100 U/mL penicillin and 100 μg/mL streptomycin (Gibco), 2 mM glutamine (Sigma-Aldrich), plus 20% horse serum, and 30% (v/v) L929 conditioned medium as a source of M-CSF. After washing and re-culturing for 3 days in fresh medium, BMDMs were stimulated with silica microparticles (250 μg/mL), c-di-AMP (6 μg/

mL), or purified lung DNA from WT mice (6 μg/mL) in supplemented DMEM. Nucleic acids were transfected using Lipofectamine 2000 as above.

Bone marrow-derived dendritic cells (BMDCs) were differentiated in RPMI 1640 medium supplemented with HEPES 25 mmol/L, 1 mM sodium pyruvate, 40 μg/mL gentamicin, 50 μmol/L 2-mercaptoethanol, 10 mmol/L lL-glutamine, 0.2% vitamins (Gibco®), 100 U/mL penicillin and 100 μg/mL streptomycin, plus 10% FCS, and 5% (v/v) J558 cell conditioned medium as a source of GM-CSF. BMDCs were stimulated as BMDMs.

**Measurement of double-stranded DNA**. Double-stranded DNA was measured in the BALF using Quant-iTPicoGreen dsDNA reagent (Invitrogen, Carlsbad, CA), according to the manufacturer's protocol. For dsDNA measurement using Quant-iTPicoGreen dsDNA reagent, cells were stimulated in a DMEM high glucose without phenol red medium (ThermoFisher Scientific, Waltham, USA).

**Quantitative RT-qPCR analysis**. Total RNA was collected and extracted in TRI-Reagent (Sigma). RNA integrity and quality was controlled using Agilent RNA 6000 Nanopuces kit®. Reverse transcription was performed with SuperScript®III Kit (Invitrogen), and cDNA was subjected to quantitative real-time PCR using primers for Ifnα4 (#QT01774353), Ifnβ1 (#QT00249662), Irf3 (#QT00108759), Irf7 (#QT00245266), Tlr2 (#QT00122458), Tmem173 (#QT00261590), and Mb21d1 (#QT00131929; all from Qiagen) and GoTaq® qPCR-Master Mix (Promega). RNA expression was normalized to Gapdh expression (Qiagen). Data were analyzed using the comparative analysis of relative expression by $^{\Delta\Delta}$Ct methods[73].

**Quantification of mitochondrial versus nuclear DNA**. Total DNA released in the BALF or from cell culture was purified using DNeasy Blood & Tissue kit (Qiagen) and quantified by real-time PCR (2.5 ng/well). Primers for mouse mtDNA (mMitoF1, mMitoR1) and mouse B2M (mB2MF1, mB2MR1) that do not co-amplify nuclear mitochondrial insertion sequences (NumtS), mouse fragments of mitochondrial genome present in the nuclear genome in the form of pseudogenes, were used[74]. The quantitative real-time PCR were performed in AriaMx Real-Time PCR System.

**In vivo and in vitro DNase I treatment**. Mice were treated with extracellular DNase I (Sigma® 200 μg/mouse, i.p.) 4 h prior and 8 h, 2 and 4 days after silica fine particles exposure (1 mg/mouse, i.t.).

BMDMs and BMDCs were treated with DNase I (1 μg/mL) 3 h prior and 1 h after silica fine particles exposure (250 μg/mL).

**Immunofluorescence microscopy**. Lung tissues were kept in freezing tissues OCT®, 10-μm lung sections were performed in cryostat (Leica®) and heated at 80 °C for 40 min in citrate 10 mM pH = 6. Lung cells were permeabilized with 0.5% Triton X-100 in 2% BSA in PBS-SVF (v/v) 10% for 1 h, three washes in TTBS and incubated overnight with rabbit anti-STING (ab92605 1/50, Abcam) in PBS containing 2% BSA, 10% FCS and 0.5% Triton X-100. Lung tissues were washed with TTBS and incubated with anti-rabbit IgG Alexa 532 secondary antibody (1/100) for 1 h. Following washing, cells were stained with DNA dye Draq5 (1/1000) for 3 min, washed with PBS, and mounted onto microscope slides (Fluoromount). Cells were observed by using a Zeiss Axiovert 200 M microscope coupled with a Zeiss LSM 510 Meta scanning device (Carl Zeiss Co. Ltd., Jena, Germany). The inverted microscope was equipped with a Plan-Apochromat 63X objective (NA = 1.4) Images were acquired using Zeiss LSM Image Browser (Carl Zeiss Co. Ltd., Jena, Germany).

In vitro, BMDMs were harvested, washed twice in cold-PBS and plated on coverslips overnight at 37 °C 5% CO$_2$ in DMEM complemented with 5% (v/v) FCS. For PBMCs and BMDCs, cells were adhered on coverslips using Poly-L-Lysine (0.1X in water) for 6 h at 37 °C 5% CO$_2$. Cells were then fixed with 4% paraformaldehyde in PBS for 15 min, permeabilized with 0.2% Triton X-100, blocked with 3% BSA in PBS for 30 min, and incubated overnight with rabbit anti-STING (ab92605 1/200 for PBMCs and 1/50 for BMDCs, Abcam) containing 1% BSA and 0.1% Triton X-100 in PBS at RT. Cells were washed with PBS and incubated with anti-rabbit IgG Alexa 532 fluorescence secondary antibody (1/100) for 1 h at RT. After washing, cells were stained with DNA dye Draq5 as above.

For ROS-mitochondria double staining, BMDMs were incubated at 37 °C during 15 min with MitoSOX (1 μM), washed, fixed with 4% paraformaldehyde in PBS for 15 min and counterstained with MitoTracker™ Green FM (100 nM) at 37 °C for 45 min, before washing and DNA dye Draq5 staining as above.

**Immunoblots**. The left part of lung tissues, BMDCs, BMDMs, and PBMCs were lysed with 0.5 ml of 1X RIPA lysis buffer (50 mMTris pH 7.5, 150 mM sodium chloride, 0.5% sodium deoxycholate, and 1% NP-40) containing 1X complete EDTA-free Protease inhibitor cocktail tablets (Roche Diagnostics), 1X Phosphatase Inhibitor Cocktail 2 (Sigma-Aldrich), 1 mM sodium orthovanadate (Sigma-Aldrich), 25 mM sodium fluoride (Sigma-Aldrich), and 1 mM phenylmethylsulfonyl fluoride (Sigma-Aldrich). The lysates were centrifuged and protein concentration was quantified in the supernatant by using DC™ Protein Assay Kit (Bio-Rad®). Total protein (30 μg per sample) were heated 5 min at 95 °C

in low reducing conditions (1% 2-mercaptoethanol) unless otherwise stated, before loading on 6–15% polyacrylamide gel and run at 160 V for 45 min using the Bio-Rad Mini-PROTEAN Tetra Cell. Proteins were transferred from the gel to a nitrocellulose membrane using a Trans-Blot SD. Transfer System (Bio-Rad) at 100 V for 45 min. Successful protein transfer was confirmed by using Ponceau S staining (Sigma-Aldrich). Membranes were blocked with 10% non-fat milk in 1X TBS-T (20 mM Tris Base, 150 mM sodium chloride, and 0.05% Tween-20 pH 7.6). Primary antibodies used were from rabbit anti-STING (#ab92605 1/1000; Abcam), rabbit anti-TBK1/NAK (#D1B4 1/1000; Cell Signaling), rabbit anti-IRF-3 (#D83B9 1/1000; Cell Signaling), rabbit anti-MLKL (#MABC604 1/1000; Merck), rabbit anti-GSDMD (#ab209845 1/500; Abcam), rabbit anti-NLRP3 (#15101 1/1000; Cell Signaling), rabbit anti-phospho-STING (Ser366; #19781 1/500; Cell Signaling), mouse anti-phospho-MLKL (Ser345; #MABC1158 1/500; Merck), rabbit anti-Phospho-TBK1/NAK (Ser172; #D52C2 1/500; Cell Signaling), rabbit anti-Phospho-IRF-3 (Ser396; #4D4G 1/500; Cell Signaling), goat anti-mouse alpha smooth muscle actin (#ab21027 1/3000; Abcam), and rabbit anti-human beta actin (#ab227387 1/3000; Abcam). Antibodies were used according to the manufacturer instructions. Portions of the membrane were incubated 2 h at RT for rabbit anti-STING, rabbit anti-IRF-3, rabbit anti-TBK1/NAK, goat anti-β-actin, rabbit anti-β actin and overnight at 4 °C for rabbit anti-Phospho-TBK1/NAK and rabbit anti-Phospho-IRF-3 in 5% non-fat milk in TBS-T at the appropriate dilutions. Membranes were washed in TBS-T three times for 10 min each at room temperature. Membranes were incubated with goat anti-rabbit-IgG-HRP-conjugated (#ab97057 1/4000; Abcam) or mouse anti-mouse-IgG-HRP-conjugated (#7076 S 1/4000; Ozyme) secondary antibodies (Bio-Rad) diluted in 5% non-fat milk in TBS-T for 2 h at RT. The membranes were washed three times in TBS-T and Amersham ECL Prime Western Blotting Detection Reagent (Thermofischer) was used to develop the blots on film (PXi gel doc system®). Uncropped immunoblot gels are displayed in Supplementary Fig. 14.

**Flow cytometry.** The left lung lobe was harvested and lung cells were isolated by enzymatic digestion using liberase (20 μg/mL; Roche) and DNase I (5 mg/mL; Sigma-Aldrich) in a final volume of 5 mL of RPMI at 37°C for 30 min under agitation. Single cells were isolated, counted, and extracellular staining was performed. The following antibodies were used: anti-CD16/CD32 antibodies (2.4G2; BD Biosciences) to avoid nonspecific binding. All staining reactions were performed at RT for 20 to 30 min. Different subsets of cells were detected by flow cytometry in cell suspensions of lung using a combination of the following fluorochrome-conjugated antibodies against mouse CD45-AlexaFluor 700 (1:100; 30⁻F11), CD11b-BV421 (1:300; M1/70), CD11c⁻APC-Cy7 (1:400; N418), F4/80-PE-Cy7 (1:400; BM8), Ly6G-BV605 (1:100; 1A8), Ly6C-FITC (1:200; HK1.4), CD4-APC (1:200; GK1.5), TCRβ-BV510 (H57-597 and CD170-PerCp-Cy5.5 (1:100; E50-2440). To verify BMDM and BMDC culture purity and activation, cells were stained as above and using in addition antibodies to mouse MHCII-BV785 (1:300; M5/114), CD40-PE (1:100; 3/23), and CD86-BV605 (1:300; GL-1). All antibodies were from BioLegend except CD170-PerCp-Cy5.5 (BD Biosciences®). In all experiments, FITC Annexin V-PE Propidium Iodide Apoptosis Detection Kit I (BD Bioscience) was used to discriminate early apoptosis to late apoptosis/necrosis. Optimal PMT voltages and antibody titrations were performed to properly separate negative and positive staining populations. Flow cytometry analyses were performed on LSR Fortessa X-20 flow cytometer (Becton Dickinson). Gating strategy was set up according to FMO control for all antibodies. One million events were recorded. Final analysis and graphical output were performed using FlowJo software (Tree Star, Ashland, OR).

For in vitro mitochondria-derived superoxide detection, BMDCs were washed twice in cold-PBS and then stained with MitoSOX(1 μM) for 15 min at 37 °C 5% $CO_2$[75]. BMDCs were then washed and flow cytometry analyses were performed on CANTO II flow cytometer (Becton Dickinson)

**Mitochondrial DNA depletion.** On day 7 of differentiation, BMDMs and BMDCs were cultured for 4 days in the presence of EtdBr (150 ng/mL), washed, and plated overnight in the absence of EtdBr[28,29]. Cells were then stimulated with silica (250 μg/mL) or transfected with c-di-AMP (6 μg/mL).

**Lung DNA extraction, purification, and in vitro stimulation.** Total DNA was purified from lungs of silica-exposed and saline vehicle-treated mice with DNeasy Blood & Tissue kit (Qiagen)[76]. DNA quantity and purity were controlled using Nanodrop 1000 (Thermo Scientific). Purified DNA was transfected into BMDMs and BMDCs (6 μg/mL) with lipofectamine 2000 according to manufacturer instructions (Thermofisher Scientific).

**Statistical analysis.** Coefficients of correlation (r) were presented as measures of linear association for regression relationships. All data are shown as mean ± SEM. Statistical analysis was performed using Student $t$ test, Mann–Whitney U analysis, nonparametric Kruskal–Wallis test followed by the Dunn post test or using a one-way analysis of variance (ANOVA) with Tukey post hoc test to determine significant differences between groups, as indicated. $p$-values < 0.05 were considered statistically significant. All analyses were performed using GraphPad Prism version 6.0 (GraphPad Software, La Jolla, CA).

**Reporting summary**. Further information on research design is available in the Nature Research Reporting Summary linked to this article.

## Data availability
All data are available from the corresponding authors on request.

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

## Acknowledgements

The authors thank Glen N. Barber for helpful discussions and sharing the STING-deficient mice, to Zhijian J. Chen for providing cGAS-deficient mice, to Peter Broz for sharing GSDMD-deficient mice, and to Yavuz Alper for his contribution in collecting and handling human samples. This work was supported by CNRS, University of Orleans, and European funding in Region Centre-Val de Loire (FEDER N° 2016-00110366).

## Author contributions

S.B. performed most of the experiments with the assistance of B.B., S.R., D.G., L.M., C.C., M.L.B., O.P., D.L., L.D. S.B., D.T., B.R. and V.Q. conceived the project, designed the experiments, analyzed and interpreted the data. A.K., H.U., F.S.A. and M.A. analyzed and interpereted human silicosis data. L.M., C.L., M.R., O.P., D.L., M.L.B. and L.A. provided resources. S.B., D.T., L.A., B.R. and V.Q. wrote the paper. All authors had the opportunity to discuss the results and comment on the paper.

## Additional information

**Competing interests:** The authors declare no competing interests.

