## [Peer Review File · Nature Communications]

Reviewers' comments:

Reviewer #1 (dsDNA, type I IFN response)(Remarks to the Author):

In this manuscript, Benmerzoug et al. implicate the STING pathway as an important contributor to lung inflammation and damage after silica instillation. It is well known that silica internalization leads to lysosomal damage, NLRP3 inflammasome activation, and IL-1b/IL-18 secretion. However, the role for other damage-associated immune signaling events after silica exposure remain uncharacterized. Here, the authors report roles for the cytosolic DNA sensing machinery in triggering type I interferon and pro-inflammatory responses in the lung after silica exposure. They further delineate unique aspects of self-DNA sensing and signaling in lung-relevant macrophage and dendritic cell populations. The use of in vivo and in vitro mouse models in addition to human samples increases the impact of this work; however, there are several important areas of weakness that should be addressed to strengthen the conclusions and broaden the relevance of the main findings.

Major concerns:

1. Several experiments lack critical controls that would further support the authors' narrative. First, the authors use autoclaved silica, which could contain residual LPS (Sandle 2013 American Pharm Rev, Miyamoto et al. 2009 Appl Environ Microbiol, Ravikumar et al. 2013 J Mater Chem B Mater Biol Med). To control for this the authors could use TLR4-/- mice/macrophages to ensure that the silica being instilled does not contain endotoxin that will surely induce IRF3-dependent CXCL10, type I interferon, and ISGs. Furthermore, do silica exposed mice have increased or altered amounts of lung bacteria compared to vehicle treated mice, which could be a source of increased dsDNA? Much is made of the role for self-DNA as the key inflammatory agonist of the DDX41/cGAS/STING pathway after silica exposure, but there is no effort to determine the source of the dsDNA observed in the lung BAL. Bacterial 16S ribosomal RNA could be used to quantify bacterial load, and similarly, qPCR (or sequencing) to screen for nuclear and mitochondrial sequence should be performed to more directly clarify the origin of the dsDNA.

2. Since ethidium bromide treatment of dendritic cells in figure 6e decreases the transcript levels of STING and cGAS, it is important to confirm that the pathway is still active to the same level as in vehicle exposed cells. The authors should therefore control for the activity of the cGAS/STING pathway by utilizing ISD transfection to document that ethidium bromide treatment does not generally impair cGAS and or DDX41 dependent interferon/inflammatory responses. Furthermore, as the authors do not quantify whether their cells lose mitochondrial DNA after EtBr, it is inconclusive as to whether the changes in the immune phenotype are due to loss of mitochondrial DNA, and thus this key self-ligand, or due to decreased levels/activity of cGAS and STING.

3. In figure 6 and 7 it is established that the critical ligand for macrophages is extracellular self DNA, whereas extracellular self DNA is dispensable for dendritic cell activation after silica exposure. In contrast, internal self DNA, presumed to be mitochondrial DNA, is the crucial ligand of dendritic cells after silica exposure. Is there more mitochondrial DNA in the cytosol of dendritic cells compared to macrophages after silica exposure?

4. In figures 1, 2, 3, 4, 5, and supplemental figure 2 high AnnexinV and PI are used as markers for late apoptosis/necrosis. Are the macrophages or dendritic cells undergoing inflammatory cell death or programmed cell death? As cell death is an important aspect of this manuscript, the authors should query specific markers of necroptosis (MLKL phosphorylation) and pyroptosis (caspase-1 cleavage). Silica will certainly drive pyroptosis via NLRP3 inflammasome activation, so do alveolar macrophages release self-DNA in an NLRP3-dependent manner? The timing and mode of cell death of all relevant cell types after silica exposure needs to be further clarified, and the inclusion of NLRP3-/- mice and or

macrophages would provide important insight into how the NLRP3 inflammasome and cGAS-STING pathways are intertwined to contribute to lung inflammation in this model.

5. The authors indicate that mitochondrial stress and subsequent ROS production are involved in DAMP release in response to silica exposure. However, the mitochondrial ROS measurements by IF staining in figure 4a lacks distinct mitochondrial localization as seen in the literature (Crnkovic et al. 2012 Free Radic Biol Med, Zhang et al. 2013 Cancer Research). An increase in the quality and magnification of the authors' IF images is suggested to enhance the visibility of changes and support the authors' claims. Moreover, it is suggested that the authors use additional markers of mitochondrial stress, such as examining mitochondrial morphology and distribution, or analysis of mitochondrial membrane potential.

6. The STING dimerization immunoblots and the STING puncta staining (4c-d) are not entirely convincing. In fact, the quality of many of the immunoblots is low and would benefit from some sort of quantitative analysis. The authors should utilize a phospho-STING antibody to confirm STING activation in figures 2, 3, 4, 5, suppl 2, and suppl 7. Moreover, the immunoblotting of the human clinical samples in figure 2 should be expanded to more clearly document cGAS/STING pathway activation by examining phospho-TBK1 and phospho-IRF3, as is done in mouse lung samples.

7. In figure 7, cGAS^{-/-} cells have a lower response to the addition of cDN. This is an interesting, yet somewhat confusing finding. Can the authors speculate as to why the response to cDN, which directly activate STING, is impaired in cGAS KO cells and mice? Absence of cGAS should have no effect on the activation of STING by exogenous cDNs, a finding that has been well-documented in other reports.

Minor concerns:

1. The work presented here is commendable in that both macrophages and dendritic cells are queried in the experiments, however the figures lack clearly defined labels for cell type and it is ambiguous as to which is used, as both are often presented side by side. Additionally, there are several inconsistencies in labels, presentation of statistical significance, and a lack of information of abbreviations used in the table, which collectively diminish the clarity of the information presented. The authors should indicate which panels are macrophages and which are dendritic cells through labels or color/pattern differences, unify the manner in which statistical significance is presented, and create a legend for the table to enhance readability. Moreover, the authors should indicate which clinical samples noted in the table are used in Fig 2a. blots.

2. There are several grammatical and spelling errors present in the text (i.e. oxidative is misspelled as oxydative).

Reviewer #2 (Lung inflammation, nanomedicine)(Remarks to the Author):

This is an interesting and detailed investigation of the role of STING in the development of silica-induced fibrosis. Although STING pathways are increasingly being described to be important in lung disease, this observation is, as far as I know, novel and will be of interest to scientists in the field. The authors have used human material and both animal models and in vitro methods to address their questions.

The authors show that the STING pathway is activated in human IPF tissue and correlates with epithelial damage; I may have missed this, but it is not clear how epithelial damage was evaluated. This needs to be clarified and proven to be rigorous and not subjective. It is also important that the

reader appreciates that these individuals have not been diagnosed with silicosis. That this phenomenon is likely related to the disease process, rather than specifically silica exposure. In fact, the details of patient demographics describe fibrosis or IPF; the difference between these need to be shown. The term IPF suggests nothing is known of the cause of the lung fibrosis, but what about the other patients? There are emerging studies of the role of STING in pulmonary fibrosis which should be included in the discussion/introduction(eg Qiu et al, Front Immunol 2017, 8:1756; Liu et al NEJM 2014, 371(6):507; Nathan et al Curr Opinions in Pulm Med 2018, 24:253). Indeed, the latter includes the role of genetic status of STING in lung fibrosis.

The mouse studies indicate that release of self dsDNA, likely by dying macrophages, into the airways is an important mediator of STING activation and type 1 IFN pathway activation. This is supported by studies of human PBMCs. Much work is performed to establish whether the extracellular or intracellular dsDNA is responsible and evidence suggests that for macrophages, the extracellular dsDNA is important, possibly internalised during uptake/phagocytosis of dsDNA coated silica particles. Phagocytosis usually contains (endosomal) and destroys material that is taken up, so it is interesting that the authors think it is subsequently cytosolic and activating STING pathways. What would be the mechanism? Presumably, there is not a receptor for dsDNA at the macrophage cell surface which internalises the DNA directly into the cytosol?

Regarding the in vitro macrophage models, a relatively large amount of silica was used in the exposure and it is perhaps not surprising that there was significant programmed cell death. Normally, apoptotic cells would be internalised by efferocytosis by macrophages. My understanding is that the DNA would not normally be released into the extracellular compartment. Is efferocytosis compromised by silica? So that the cells go into secondary necrosis and release DNA? Certainly the videos show cells stuffed with particles, which one assumes are present in exposed lungs, but which are supplemented by recruitment of fresh macrophages to the affected site.

The authors show that DCs behave differently with respect to STING pathway induction, where mitochondrial DNA drives the process. It should be noted that the authors have recently published on DC and the mechanism of STING activation (J Innate Immunology 2018, 10:239) and some of the work is similar, though using different environmental factors. How likely would they be exposed to silica particles of this size, and would they internalise the particles in vivo? Because it would seem that this is part of the process leading to mtDNA release and STING activation. Which of the cellular responses is most important - macrophage or DCs? And importantly, are there other cells at the gas-liquid interface (eg epithelial cells) that might also have the STING pathway that would be therapeutic targets (particles do not need to get into cells to activate them)?

Whilst the authors have meticulously taken the reader through a barrage of studies to establish the import of STING in the fibrotic pathway, there is no control for the effects of these particles, such as a non-fibrogenic (or less fibrogenic) particulate. Consequently, it is not clear whether this is an effect of particles, or the unique effect of crystalline silica.

There are many typographical errors that need to be addressed

Macrophages are notoriously difficult to transfect. More details of the measure of efficacy are needed.

It is not always clear how many samples were used for Western blotting experiments and not always shown bar graphs and significances of the Western blotting work.

Reviewer #3 (cGAS/STING signalling, DAMP/PAMP)(Remarks to the Author):

In this work Benmerzoug have explored the role of DNA in stimulation of pathological responses in silica-induced inflammation. The work is based on data from a strong mouse model, and the basic observation of a cGAS-STING dependent response upon silica treatment is strong. The subsequent cellular characterization of silica-induced cell death macrophages and DCs and macrophages reveals interesting differences, but is not fully developed. Finally, the attempt to perform detailed cellular/molecular characterization of the pathways for silica-induced cell death has significant flaws, and requires much more work to allow the conclusions drawn by the authors.

SPECIFIC POINTS:

1. Figure 1. The authors should conclusively establish the mode of cell death: apoptosis (cleaved caspase 3, DNA fragmentation), necroptosis (pMLKL), piroptosis (cleaved GasderminD).
2. The functional immunological data should be complemented with results on how cGAS and STING-deficiency impacts on disease development upon silica-treatment
3. Figure 2a. Data from a healthy donor not receiving cortisone treatment should be shown. It is surprising that no STING expression (not even STING monomer) is observed in the cortisone-treated sample. Therefore, the data would also gain if STING immunoblot from a reduced SDS-PAGE was shown.
4. Figure 3b (Immunoblot data). There is no detectable STING expression in the WT, silica- sample. This is very surprising and somewhat worrying.
5. Figure 3c. The V+/PI- and V+/PI+ data from all conditions/cell types under investigation should be shown.
6. The reduced cell death in the cGAS and STING KO cells is interesting, but the consequence (namely that silica-induced cell death and not only the downstream inflammatory response is dependent on cGAS-STING) is quite poorly characterized. This represents a weakness of the study.
7. Figure 4b. The authors show that the levels of free DNA is lower after silica-stimulation in STING KO cells. Is this data not in conflict with the data in the final display item in Figure 3b? Or does the in vitro system not mimic the in vivo situation?
8. Figure 5. As I understand it, the authors proposed that silica-induced ROS formation triggers STING-dependent cell death. What is the effect of ROS inhibitors/scavengers on silica-induced cell death?
9. Figure 5c. The IF data on Draq5-STING colocalization should be quantified.
10. Figure 6. The characterization of the difference in the response by macrophages and DCs is interesting but somewhat superficial. For instance, in panel 6, how do the authors explain that EtdBr treatment abolishes STING expression, but still allows response to CDNs? As a very minimum, the transcript data should be complemented by immunoblot data.
11. Supplementary Figure 7. This part is very underdeveloped. Non-cGAS cytosolic DNA sensors are very controversial, particularly in mice (e.g. Immunity 45(2):255-66). Therefore, the results presented should be confirmed/replaced by data from KO cells (e.g. using CRISPR). Moreover, the authors should show that the proposed sensors colocalize with DNA in the cytoplasm.

MINOR POINTS

12. It is very unusual that data referred to in the abstract are not shown in the main figures (data on DDX41 and IFI204).
13. Each display item should have its own panel. It is very difficult to follow the data presentation in the text.

Response to the Referees comments:

Reviewer #1 (dsDNA, type I IFN response)(Remarks to the Author):

In this manuscript, Benmerzoug et al. implicate the STING pathway as an important contributor to lung inflammation and damage after silica instillation. It is well known that silica internalization leads to lysosomal damage, NLRP3 inflammasome activation, and IL-1b/IL-18 secretion. However, the role for other damage-associated immune signaling events after silica exposure remain uncharacterized. Here, the authors report roles for the cytosolic DNA sensing machinery in triggering type I interferon and pro-inflammatory responses in the lung after silica exposure. They further delineate unique aspects of self-DNA sensing and signaling in lung-relevant macrophage and dendritic cell populations. The use of in vivo and in vitro mouse models in addition to human samples increases the impact of this work; however, there are several important areas of weakness that should be addressed to strengthen the conclusions and broaden the relevance of the main findings.

Major concerns:

1. Several experiments lack critical controls that would further support the authors' narrative. First, the authors use autoclaved silica, which could contain residual LPS (Sandle 2013 American Pharm Rev, Miyamoto et al. 2009 Appl Environ Microbiol, Ravikumar et al. 2013 J Mater Chem B Mater Biol Med). To control for this the authors could use TLR4^{-/-} mice/macrophages to ensure that the silica being instilled does not contain endotoxin that will surely induce IRF3-dependent CXCL10, type I interferon, and ISGs.

To ensure elimination of residual endotoxin, the freshly prepared suspension of crystalline silica particles was autoclaved at 121°C for 30 min before use. This point is now added in the M&M section p25.

As suggested, we controlled for a potential residual endotoxin contamination in mice deficient for TLR4 pathway. TLR2/TLR4 double deficient mice were instilled with silica microparticles and analyzed at day 7. Lung inflammation assessed by the number of inflammatory cells in the BAL or TNF expression in the lung was similar in TLR2/TLR4^{-/-} mice and WT mice. Further, the expression of *Ifnβ* and the concentration of CXCL10 in the lung, a type I interferon induced gene, was also similar in TLR2/TLR4^{-/-} mice as in WT mice after silica exposure.

Thus we are confident that the lung inflammation induced by silica instillation is not due to a LPS/TLR4 response in our model.

The results are described on page 5 and have been added as supplementary Figure 1 k-n.

New Supplementary Figure 1 k-n legend:

"(k-n) Silica microparticles (1mg/mouse i.t.) or saline vehicle as in (a) were administered to WT and TLR2/TLR4^{-/-} mice and parameters analyzed on day 7 post-exposure. (k) Total cells in the BALF. (l) Lung levels of TNF measured by ELISA, (m) *Ifnβ1* transcripts measured by real-time PCR, and (n) CXCL10 measured by ELISA."

Revised Result section on page 5:

"We verified in TLR2/TLR4 double deficient mice that the effect of silica was not due to endotoxin or other PAMPs (supplementary Fig. 1 k-n)."

Revised M&M page 25:

"Crystalline silica particles (DQ12, d50=2,2μm, DMT GmbH and Co. KG, Essen, Germany) powders were sterilized by heating at 200°C for 4h in autoclave, as described previously (Giordano et al., 2010). The freshly prepared suspension of crystalline silica particles was autoclaved at 121°C for 30 min before use."

Furthermore, do silica exposed mice have increased or altered amounts of lung bacteria compared to vehicle treated mice, which could be a source of increased dsDNA?

Much is made of the role for self-DNA as the key inflammatory agonist of the DDX41/cGAS/STING pathway after silica exposure, but there is no effort to determine the source of the dsDNA observed in the lung BAL. Bacterial 16S ribosomal RNA could be used to quantify bacterial load, and similarly, qPCR (or sequencing) to screen for nuclear and mitochondrial sequence should be performed to more directly clarify the origin of the dsDNA.

We quantified bacteria in the lung of silica exposed vs saline treated mice and found no difference. There was no increase of lung colony-forming bacteria enumerated after 24h to 72h on 5% sheep blood containing Columbia ANC agar specific for Gram + cocci, TSS Trypcase Soy Agar containing 5% sheep blood to identify staphylococcus in the airways, chromogenic CPSE translucent agar, or CET cetrimide-containing agar culture medium specific for pseudomonas. We thus excluded bacteria as a potent source of increased dsDNA.

To better characterize the source of dsDNA, we quantified the relative amount of nuclear and mitochondrial DNA using specifically designed qPCR primers that do not co-amplify nuclear mitochondrial insertion sequences (NumtS), mouse fragments of mitochondrial genome present in the nuclear genome in the form of pseudogenes (Malik et al., 2016). These authors reported the lowest mitochondrial genome to nuclear genome ratio in the lung tissue (6 copies Mt/nuclear genome).

Here we see a 8-fold increase in Mt genome in the BAL fluid after silica exposure, as compared to a 4-increase in nuclear genome. Thus, we think that there is no major contribution of lung bacteria as a source of dsDNA, and that the release of mitochondrial DNA is more prominent over nuclear DNA in the airways after silica exposure.

The new data have been added in Figure 1i and reported in the Results section on page ...

New Figure 1i:

Revised Figure 1i legend: page 39

"(i) Increase in nuclear DNA (nDNA) and mitochondrial DNA (mtDNA) in the BALF, expressed as the fold increase after silica exposure over saline control."

Revised Result section on page 5-6:

"We further characterized the source of self-dsDNA released in the BALF of WT mice exposed to silica. Nuclear DNA (nDNA) and, even more, mitochondrial DNA (mtDNA) were increased in the BALF after silica exposure (Figure 1i)."

Revised M&M section on page 27:

"Quantification of mitochondrial versus nuclear DNA

Total DNA released in the BALF or from cell culture was purified using DNeasy Blood & Tissue kit (Qiagen) and quantified by real time PCR (2.5 ng/well). Primers for mouse mtDNA (mMitoF1, mMitoR1) and mouse B2M (mB2MF1, mB2MR1) that do not co-amplify nuclear mitochondrial insertion sequences (NumtS), mouse fragments of mitochondrial genome present in the nuclear genome in the form of pseudogenes, were used (Malik et al., 2016). The quantitative real-time PCR were performed in AriaMx Real-Time PCR System."

New Reference:

70 . Malik AN, Czajka A and Cunningham P. Accurate quantification of mouse mitochondrial DNA without co-amplification of nuclear mitochondrial insertion sequences. Mitochondrion 29:59-64 (2016).

2. Since ethidium bromide treatment of dendritic cells in figure 6e decreases the transcript levels of STING and cGAS, it is important to confirm that the pathway is still active to the same level as in vehicle exposed cells. The authors should therefore control for the activity of the cGAS/STING pathway by utilizing ISD transfection to document that ethidium bromide treatment does not generally impair cGAS and or DDX41 dependent interferon/inflammatory responses.

We now verified by immunoblot the activation of the cGAS/STING pathway in dendritic cells stimulated by silica or cDNs in the presence of EtBr. The new Figure 6g shows that EtBr treatment reduced STING protein overexpression, dimerization and phosphorylation in response to silica, while cGAS protein was unchanged. In contrast, EtBr treatment did not affect STING protein overexpression, dimerization and phosphorylation in response to cDN (Fig. 6g). Further, the type I IFN response of both dendritic cells and macrophages to cDNs was not affected after EtBr treatment, as shown in Figure 6h,i.

New Figure 6g:

Revised Figure 6g legend: page 48

"(g) Immunoblot showing cGAS, phospho-STING, STING protein expression and dimerization in DCs after EtdBr treatment, with beta-actin as a reference."

Revised Result section on page 15:

"STING protein expression, phosphorylation and dimerization in response to silica was strongly reduced in DCs after EtBr treatment, while the response to control c-di-AMP (cDN) was spared (Figure 6g). cGAS protein levels were similar after silica or cDN stimulation, without or with EtdBr treatment (Fig. 6g). In mtDNA-depleted WT DCs, type I IFNs expression was totally abolished and CXCL10 strongly decreased (Fig. 6h), while the response to c-di-AMP was unaffected, excluding a cytotoxic effect of the EtdBr treatment."

Furthermore, as the authors do not quantify whether their cells lose mitochondrial DNA after EtBr, it is inconclusive as to whether the changes in the immune phenotype are due to loss of mitochondrial DNA, and thus this key self-ligand, or due to decreased levels/activity of cGAS and STING.

As suggested, we now quantified mitochondrial DNA after EtdBr treatment. Real-time quantitative PCRs were performed on cell fraction from dendritic cells treated or not with EtdBr and exposed to silica, using mMitoF1 and mMitoR1 primers for mitochondrial DNA and mB2MF1 and mB2MR1 for nuclear DNA. The new Figure 6e,f shows that EtdBr decreases intracellular mitochondrial DNA without affecting intracellular nuclear DNA. Thus, EtdBr at a low concentration of 150 ng/mL prevents mtDNA replication yielding a 10-fold diminution in mitochondrial DNA.

New Figures 6 e,f:

Revised Figure 6 e,f legend: page 48

"Fold change of (e) mitochondrial DNA (mMitoF1 and mMitoR1) and (f) nuclear DNA (mB2MF1 and mB2MR1) in untreated versus EtdBr-treated BMDC exposed to silica, as compared to untreated unstimulated cells."

Revised Result section on page 15:

"EtdBr-treated DCs displayed a drastic reduction of mtDNA without affecting nuclear DNA (Fig. 6e, f)."

3. In figure 6 and 7 it is established that the critical ligand for macrophages is extracellular self DNA, whereas extracellular self DNA is dispensable for dendritic cell activation after silica exposure. In contrast, internal self DNA, presumed to be mitochondrial DNA, is the crucial ligand of dendritic cells after silica exposure. Is there more mitochondrial DNA in the cytosol of dendritic cells compared to macrophages **after silica exposure**?

We have not directly answered this question, partly because of the technical difficulty to discriminate between nuclear and mtDNA in the cytosol, but also as DC were reported to contain less endonucleases in their cytoplasm and thus to degrade DNA less efficiently than macrophages (Lindhout et al., 1995; Xu et al., 2017). These authors report that "while no significant change of mtDNA was observed in macrophages and monocytes, mtDNA was markedly increased in DCs in response to CD47 blockade in MC38 tumor model". Further, they conclude "Together, these data suggest that intervention of CD47-SIRPa signaling in DCs specifically modified the phagosomal environment into a less favorable state for degradation."

Indeed, we show by immunofluorescence staining of DNA Draq5 an increase in cytosolic DNA in DCs (Figure 5e) as compared to macrophages (Figure 4h), 18h after silica exposure.

Further, the functional evidence that we provide to document the difference between the two cell types, is now further validated by the mtDNA depletion with EtdBr (point 2).

4. In figures 1, 2, 3, 4, 5, and supplemental figure 2 high AnnexinV and PI are used as markers for late apoptosis/necrosis. Are the macrophages or dendritic cells undergoing inflammatory cell death or programmed cell death? As cell death is an important aspect of this manuscript, the authors should query specific markers of necroptosis (MLKL phosphorylation) and pyroptosis (caspase-1 cleavage). Silica will certainly drive pyroptosis via NLRP3 inflammasome activation, so do alveolar macrophages release self-DNA in an NLRP3-dependent manner? The timing and mode of cell death of all relevant cell types after silica exposure needs to be further clarified, and the inclusion of NLRP3^{-/-} mice and or macrophages would provide important insight into how the NLRP3 inflammasome and cGAS-STING pathways are intertwined to contribute to lung inflammation in this model.

We are thankful for the suggestions and now added information on specific markers of cell death including MLKL phosphorylation, caspase-3 cleavage, and Gasdermin D cleavage.

We originally used AnnexinV/PI as markers for late apoptosis/necrosis as this is a generally accepted method used by prominent authors in the field (Gulen et al., 2017). Silica particles have been reported to drive programmed cell death through apoptosis (Leigh et al., 1997), necrosis (Joshi and Knecht, 2013) and necroptosis (Mulay et al., 2016). However, there was no evidence regarding *in vivo* pyroptosis after silica microparticles intratracheal administration. To address the questions raised, we first assessed Caspase 3 cleavage as a marker of apoptosis, Gasdermin D (GSDMD) cleavage as a marker of pyroptosis, and phosphorylated MLKL as a marker for necroptosis (Joshi and Knecht, 2013). We show in Figure 1j-m that silica drives apoptosis (cleaved Caspase 3), necroptosis (p-MLKL), but also pyroptosis (cleaved Gasdermin D).

New figure 1j-m:

New Figure 1j-m legend: page 39

“(j) Immunoblots on Caspase 3 (Casp3), cleaved Caspase 3 (c-Casp3), Gasdermin D (GSDMD), MLKL and phosphorylated MLKL (p-MLKL), normalized to β -actin protein. Immunoblots relative quantifications of (k) c-Casp3, (l) c-GSDMD and (m) p-MLKL.”

Revised Result section on page 6:

“To better characterize the mode of cell death induced by silica microparticles, we first assessed caspase 3 cleavage as a marker of apoptosis. We show that silica exposure induced caspase 3 cleavage in WT mice, indicative of silica-induced apoptosis (Fig. 1j). The contribution of pyroptosis and necroptosis in silica-induced cell death was assessed by immunoblots of cleaved gasdermin D (GSDMD) and phosphorylated Mixed Lineage Kinase domain Likepseudokinase (MLKL) (Figure 1j). Indeed, silica exposure induced the cleavage of GSDMD and phosphorylation of MLKL in the lung of WT mice (Figure 1j). In summary, airway silica exposure induces cell death through apoptosis, necroptosis and pyroptosis, with nuclear and mitochondrial self-dsDNA release, triggering type I IFN pathway activation.”

Revised M&M section on page 29:

“Primary antibodies used were from rabbit anti-STING (#ab92605; Abcam), rabbit anti-TBK1/NAK (#D1B4; Cell Signaling), rabbit anti-IRF-3 (#D83B9; Cell Signaling), rabbit anti-MLKL (#MABC604; Merck), rabbit anti-GSDMD (#ab209845; Abcam), rabbit anti-NLRP3 (#15101; Cell Signaling), rabbit anti-phospho-STING (Ser366; #19781; Cell Signaling), mouse anti-phospho-MLKL (Ser345; #MABC1158; Merck), rabbit anti-Phospho-TBK1/NAK (Ser172; #D52C2; Cell Signaling), rabbit anti-Phospho-IRF-3 (Ser396; #4D4G; Cell Signaling), goat anti-mouse alpha smooth muscle actin (#ab21027; Abcam) and rabbit anti-human beta actin (#ab227387; Abcam).”

To investigate the role of STING pathway in programmed cell death induced by silica, we analyzed MLKL phosphorylation, caspase-3 cleavage, and Gasdermin D cleavage by immunoblot in STING-deficient mice. The new data is shown on Figure 3j. Indeed, STING drives apoptosis, as there is less cleaved caspase 3 in STING^{-/-} mice, and necroptosis, as there is less phosphorylated MLKL after silica exposure in STING^{-/-} mice as compared to silica exposed WT mice (Fig. 3j). By contrast Gasdermin D is cleaved MLKL in silica exposed STING^{-/-} mice similar to WT mice (Fig. 3j), indicating that pyroptosis is independent of STING after silica exposure (Fig. 3j).

New Figure 3j:

New Figure 3j legend: page 43

“(j) Immunoblots of Caspase 3, cleaved Caspase 3, Gasdermin D, MLKL and phospho-MLKL in the lung of WT and STING^{-/-} mice with β -actin as a reference.”

New Result section on pages 10-11:

“Further, as cell death is induced in a STING-dependent manner after silica exposure, we questioned whether STING could be involved in apoptosis, necroptosis and/or pyroptosis related cell death. Indeed, silica induced apoptosis and necroptosis in a STING-dependent manner since caspase 3 cleavage and MLKL phosphorylation were highly reduced in STING^{-/-} mice (Fig. 3j). By contrast, silica-induced Gasdermin D cleavage was still present in STING^{-/-} mice, showing that STING is not essential

for pyroptosis cell death after silica exposure (Fig. 3j). These results show that silica induces necroptosis and apoptosis in a STING-dependent manner *in vivo*.”

We further analyzed *in vitro* silica induced cell death in macrophages (new Figure 4e) and dendritic cells (new Figure 5d). *In vitro* silica exposure induced apoptosis and necroptosis in macrophages, as revealed by cleaved Caspase 3 and phosphorylated MLKL. Interestingly in STING^{-/-} macrophages, Caspase 3 cleavage was reduced while phosphoMLKL was reduced after silica exposure. In dendritic cells, silica induced Caspase 3 cleavage which was reduced in STING^{-/-} cells, while MLKL was not phosphorylated. Gasdermin was not cleaved after *in vitro* silica exposure, which may reflect the lack of inflammasome activation, due to the absence of first signal. Indeed, we could not detect IL-1 β released *in vitro* in DC or macrophage after silica exposure. Thus *in vitro* silica exposure seems to trigger mostly apoptosis in DCs and apoptosis and necroptosis in macrophages.

New Figure 4e:

New Figure 4e legend: page 45

(e) Immunoblots of Caspase 3, cleaved Caspase 3, Gasdermin D, MLKL and phospho-MLKL in the lung of WT and STING^{-/-} mice with β -actin as a reference.

New Result section on page12:

“Silica exposure induced apoptosis and necroptosis in macrophages, as revealed by cleaved Caspase 3 and phosphorylated MLKL (Fig. 4e). Interestingly in silica-exposed STING^{-/-} macrophages, Caspase 3 cleavage and phosphoMLKL were reduced, as compared to WT macrophages (Fig. 4e). Gasdermin D was not cleaved after *in vitro* silica exposure, which may reflect the lack of inflammasome activation, due to the absence of first NF κ B signal (Fig.4e)²⁵.”

New Figure 5d:

New Figure 5d legend: page 47

(d) Immunoblots of Caspase 3, cleaved Caspase 3, Gasdermin D, MLKL and phospho-MLKL in WT and STING^{-/-} DCs with β -actin as a reference.

New Result section on page13:

“Silica exposure induced STING-dependent apoptosis in DCs, as revealed by cleaved Caspase 3, reduced in STING^{-/-} DCs, while MLKL was not phosphorylated and Gasdermin D was not cleaved (Fig. 5d).”

The reduced susceptibility of NLRP3^{-/-} mice to silica has been well documented (Cassel et al., 2008; Dostert et al., 2008; Gaidt et al., 2017; Hornung et al., 2008; Li et al., 2018). We repeated these experiments and confirm that NLRP3^{-/-} mice have a compromised response to silica in terms of inflammatory cells in the BALF, pulmonary TNF and CXCL10 (New Supplementary Figure 5e-g). As suggested, we now looked at cell death in the absence of NLRP3 *in vivo* after silica exposure (Supplementary Fig 5a-c). *In vivo* silica-induced pulmonary macrophages and DCs death, lung inflammation and type I IFN signaling were considerably reduced in the absence of NLRP3 (Suppl Fig 5b-i). There was no incidence on the overall concentration of dsDNA in the BAL fluid (Supplementary Fig 5d).

New result section: page 11

"Sensing of self-dsDNA by STING modulates NLRP3 expression after silica exposure.

The NLRP3 inflammasome has been recognized as an innate immune signaling receptor important for mediating cell responses and inflammation to crystals such as silica^{14, 16, 20, 21, 22}. To dissect the effect of STING adaptor molecule on NLRP3 expression and pyroptosis, NLRP3-deficient (NLRP3^{-/-}) mice were exposed to silica (Supplementary Fig. 5). The absence of NLRP3 led to a reduction of cell death in interstitial and alveolar macrophages but also apoptosis in DCs after *in vivo* silica exposure (Supplementary Fig. 5a-c), which however did not translate into a reduction of dsDNA release in the airways (Supplementary Fig. 5d). Silica-induced lung inflammation was reduced in NLRP3^{-/-} mice as compared to WT mice (Supplementary Fig 5e-f). Interestingly, type I IFN response was also reduced in the absence of NLRP3 after *in vivo* silica exposure (Supplementary Fig. 5g, h)."

New Supplementary Fig5 a-i:

New Supplementary Fig. 5 legend:

"Supplementary Figure 5. Interplay of STING and NLRP3 to promote silica-induced lung inflammation.

(a-f) Silica microparticles (1 mg/mouse i.t.) or saline vehicle were administered to WT and NLRP3^{-/-} mice and parameters analyzed on day 7 post-exposure.

(a) Gating strategy for flow cytometry analysis. Pre-gated singlet cells using FSC-A/FSC-H followed by SSC-A/SSC-H and FSC-W/FSC-A. CD45 positive cells were gated and stained with followed by F4/80 and CD11c for discriminating subcellular types: F4/80⁺CD11c⁻ interstitial macrophages, F4/80⁺CD11c⁺ alveolar macrophages and CD11c⁺F4/80⁻ dendritic cells.

(b) Dot plots showing Ann V⁺/PI⁺ interstitial and alveolar macrophages and Ann V⁺/PI⁻ DC.

(c) Proportion of dead cells (Ann V⁺/PI⁺) among interstitial and alveolar macrophages and proportion of early apoptotic cells (Ann V⁺/PI⁻) among DC (on 200 000 events).

(d) Concentration of extracellular dsDNA in the BALF acellular fraction.

(e) Neutrophils count in the BALF. Lung (f) TNF and (g) CXCL10 levels measured by ELISA, and (h) *Ifnβ* and (i) *Tmem173* transcripts measured by real-time PCR.

Regarding the interplay between NLRP3 and STING signaling pathways, we investigated STING pathway in the absence of NLRP3 and NLRP3 pathway in the absence of STING *in vivo*. STING-related gene and proteins are reduced in the absence of NLRP3, together with STING phosphorylation and dimerization, and TBK1 phosphorylation after silica exposure (Supplementary Fig 5 i-k). Furthermore, NLRP3-related gene and proteins are also reduced in the absence of STING after silica exposure (Supplementary Fig 5j,l,m). The absence of STING led to a reduction of IL-1β in the lungs after silica exposure (Fig. 3a).

Our data indicate that NLRP3 and STING pathways are interdependent leading to silica-induced lung inflammation. In human myeloid cells, cGAS-STING have been reported to orchestrate a lysosomal cell death program that engages NLRP3 in response to bacterial or viral DNA, independently of STING-induced type I IFN response (Gaidt et al., 2017).

New Supplementary Fig 5 i-m:

New Supplementary Fig 5j-m legend:

(j-m) Silica microparticles (1 mg/mouse i.t.) or saline vehicle were administered to WT, STING^{-/-} and NLRP3^{-/-} mice and lung parameters analyzed on day 7 post-exposure.

(j) Immunoblots of phospho-STING, STING, phospho-TBK1, NLRP3 and β-actin as a reference. Relative quantification of (k) STING and (l) NLRP3 from immunoblots, as compared to β-actin. (m) *Nlrp3* transcripts measured by real-time PCR.

New Result section on page 11:

"In order to understand how type I IFN signaling is reduced in NLRP3^{-/-} mice, we evaluated STING/TBK1 signaling in the absence of NLRP3 *in vivo*. Indeed, *Tmem173* transcripts, STING protein expression, phosphorylation and dimerization but also phosphorylation of TBK1 were partially reduced in the absence of NLRP3 *in vivo* (Supplementary Fig 5i-k). Conversely, STING is important for NLRP3 expression as seen by the decrease of NLRP3 protein and gene expression in the lung of STING-deficient mice after silica exposure (Supplementary Fig. 5j,l,m). Thus STING and NLRP3 pathways are linked to induce lung inflammatory response to silica."

5. The authors indicate that mitochondrial stress and subsequent ROS production are involved in DAMP release in response to silica exposure. However, the mitochondrial ROS measurements by IF staining in figure 4a lacks distinct mitochondrial localization as seen in the literature (Crnkovic et al. 2012 Free Radic Biol Med, Zhang et al. 2013 Cancer Research). An increase in the quality and magnification of the authors' IF images is suggested to enhance the visibility of changes and support the authors' claims. Moreover, it is suggested that the authors use additional markers of mitochondrial stress, such as examining mitochondrial morphology and distribution, or analysis of mitochondrial membrane potential.

We are thankful for this suggestion which helped us improve the quality of the IF images. As suggested, we repeated the experiment using MitoTracker as a marker of mitochondria and now show clearly that the ROS induced after macrophage silica exposure co-localize with mitochondria (Fig. 4a, b). We also added STING^{-/-} macrophages and show that there is very little ROS induction after silica exposure in the absence of STING. Indeed, using Overlap coefficient and Pearson's coefficient we show that the absence of STING leads to lower co-localization between mitochondria and ROS than in WT macrophages after silica exposure. This shows that STING pathway is essential for ROS induction in response to silica.

New Figure 4a,b:

b-Mφ

	WT BMDM		STING ^{-/-} BMDM	
	Medium	Silica	Medium	Silica
Overlap coefficient	0.549±0.03	0.685±0.04	0.650±0.03	0.489±0.04
Pearson's coefficient	0.517±0.03	0.683±0.04	0.595±0.02	0.403±0.06

New Fig. 4a,b legend: page 45

“(a) Brightfield confocal microscopy showing intracellular silica microparticles and MitoTracker (green) labelled mitochondria co-localizing with superoxide production detected by MitoSOX staining (red). Images are representative of 5 slides from n=3 cell cultures. Bars, 5µm.
 (b) Colocalization analysis of MitoTracker versus MitoSOX staining from the slides shown in a. Overlap coefficient and Pearson’s correlation coefficient were determined using ImageJ.”

New Result section on page 12:

“Following silica stimulation, macrophages underwent cellular stress with reactive oxygen species (ROS) production detected using MitoSOX immunofluorescence which co-localized with mitochondria revealed by MitoTracker (Fig. 4a). Interestingly, the induction of ROS after silica stimulation was impaired in STING^{-/-} macrophages (Fig. 4a, b).”

New M&M section on page 28:

“For ROS-mitochondria double staining, BMDMs were incubated at 37°C during 15 min with MitoSOX (1µM), washed, fixed with 4% paraformaldehyde in PBS for 15 min and counterstained with MitoTracker™ Green FM (100 nM) at 37°C for 45 min, before washing and DNA dye Draq5 staining as above.”

6. The STING dimerization immunoblots and the STING puncta staining (4c-d) are not entirely convincing. In fact, the quality of many of the immunoblots is low and would benefit from some sort of quantitative analysis. The authors should utilize a phospho-STING antibody to confirm STING activation in figures 2, 3, 4, 5, suppl 2, and suppl 7. Moreover, the immunoblotting of the human clinical samples in figure 2 should be expanded to more clearly document cGAS/STING pathway activation by examining phospho-TBK1 and phospho-IRF3, as is done in mouse lung samples.

We are thankful for the suggestions which helped us to improve the quality of the immunoblots. We used a phospho-STING specific antibody targeting S366 for immunoblots in Figure 1p, 2d, 2i, 3c, 4g, 5e, 6g and in Supplementary Figures 2f, 5i, 8c and 9i: We show that STING phosphorylation correlated with STING dimerization in the lungs. The specificity of the phospho-STING and STING antibody was controlled in STING^{-/-} mice. However, the role of STING phosphorylation in STING activation is controversial. According to Yang L Wand L, Ketkar H et al., Nat Com 2018, “dimerization of STING is a prerequisite for STING trafficking out of the ER to perinuclear vesicles where it recruits the TBK1 to induce IFN. STING is then phosphorylated and degraded via autophagy and proteasomes.” Indeed, two forms of phosphorylated STING monomer and dimer were described Thus, dimerization would be a marker of STING activation while phosphorylation might occur downstream, prior to degradation. We show now that in response to silica, STING phosphorylation was detected, in addition to STING dimerization, both *in vivo* and *in vitro*. The respective result sections were amended accordingly.

Example: Revised Figure 1p:

Revised Figure 3c:

As suggested, the immunoblotting of the human clinical samples has been expanded to better document cGAS/STING pathway activation in terms of TBK1 and IRF3 phosphorylation. We now show in figure 2d that STING, TBK1 and IRF3 are phosphorylated in the lung tissue of ILD patients, and less so in patients under cortisone treatment. This point is now added in the result section on page 8. We also expanded the immunoblots of human PBMC in Figure 2i: indeed, there is an increase of phospho-TBK1 and phospho-IRF3 after silica exposure, as seen after stimulation with cDNs. This point is now added in the result section on page 8.

Revised Results section on page 8:

“ We show for the first time that STING pathway is activated in the lung of ILD patients, as seen by STING overexpression, phosphorylation and dimer formation, TBK1 and IRF3 phosphorylation, and downstream CXCL10 production (Fig. 2c-e), ”

“Silica induced STING phosphorylation and dimerization, together with phosphorylation TBK1 and IRF3 in PBMCs, similar to cDN (Fig. 2I).”

Revised Figure 2d:

Revised Figure 2I:

Revised Figure 2d, I legend: page 42

“(d) Immunoblots of phosphorylated STING (p-STING), STING, TBK1, phospho-TBK1, IRF3 and phospho-IRF3 in lung homogenate, with β -actin as a reference.

(I) Immunoblots of STING, phospho-TBK1, TBK1, phospho-IRF3, and IRF3 in human PBMCs, with β -actin as a reference.

d; immunoblots shown from left to right for ILD patients H, G, C, A (see Table 2), representative of n= 3 cortisone-treated ILD patients and n=18 untreated-ILD patients, from 3 independent experiments, shown in d-f.

I; immunoblots representative of n= 6 donors from 2 independent experiments.”

A quantitative analysis of the immunoblots is now provided in Figure 1k-m,q, Figure 2d and in Supplementary figure 5k,l. The dimeric form of STING protein was quantified. We also provide quantitative qRT-PCR of STING *Tmem173* gene expression to support the protein data.

Immunoblot quantification in Revised Figure 1:

Revised Figure 1 k-m, q legends: page 41

“(j) Immunoblots of Caspase 3 (Casp3), cleaved Caspase 3 (c-Casp3), Gasdermin D (GSDMD), MLKL and phosphorylated MLKL (p-MLKL), normalized to β -actin protein. Immunoblots relative quantifications of (k) c-Casp3, (l) c-GSDMD and (m) p-MLKL. ”

Immunoblot quantification in Revised Supplementary Fig 5k,l:

Revised Supplementary Fig. 5k,l legend:

"Relative quantification of (k) STING and (l) NLRP3 from immunoblots, as compared to β -actin."

7. In figure 7, *cGAS*^{-/-} cells have a lower response to the addition of cDN. This is an interesting, yet somewhat confusing finding. Can the authors speculate as to why the response to cDN, which directly activate STING, is impaired in *cGAS* KO cells and mice? Absence of *cGAS* should have no effect on the activation of STING by exogenous cDNs, a finding that has been well-documented in other reports.

One should note that the cDN used here is c-di-AMP, and not cGAMP, the product of *cGAS* which directly activates STING.

In Figure 7b, we report that the response of macrophages *cGAS*^{-/-} to c-di-AMP is not significantly affected in terms of CXCL10 and type I *Ifn* α and *Ifn* β gene expression, while the induction of TNF and the expression of *Tmem173* at 18h are significantly reduced by the absence of *cGAS*, as compared to wild-type macrophages. We now added the levels of significance for cDN exposure to the Figure 7b. We actually do not present *in vivo* data on cDN exposure in *cGAS*^{-/-} mice. We found only one report of *cGAS*^{-/-} bone marrow derived macrophages transfected with c-di-AMP in the literature (Dey B et al., Nat Med 2015), where the response in terms of IFN β was essentially not affected (no significance indicated). We did not find report of *cGAS*^{-/-} dendritic cells transfected with c-di-AMP in the literature.

In dendritic cells, we propose that the response to c-di-AMP seen at 18h is both direct and indirect, with a contribution of the cDN inducing ROS, then cell damage/cell death and release of self-dsDNA in the cytosol that can be sensed by *cGAS* to activate STING and type I IFN pathway. *Tmem173* is less upregulated in *cGAS*^{-/-} cells stimulated with cDNs than in WT cells, which can contribute to explain why the 'secondary' response to cDNs through STING pathway is reduced in the *cGAS*^{-/-} cells.

Indeed, we show that c-di-AMP transfection induces a strong production of ROS by macrophages, with cell death and extracellular self-DNA release, as shown below (A):

Transfection of c-di-AMP in dendritic cells also induced cell death and extracellular self-DNA release (B):

Figure legend: Macrophages and dendritic cells respond to c-di-AMP by cellular stress associated with cell death and dsDNA leakage.

(A) WT and STING^{-/-} bone marrow derived macrophages were transfected with the cDN c-di-AMP (6 µg/mL) or with lipofectamine alone for 18 hr. Brightfield confocal microscopy showing superoxide production detected by MitoSOX staining (red). Flow cytometry dot plots of Annexin V/PI staining of pre-gated singlets (SSC-A/SSC-H followed by FSC-W/FSC-A) and CD11b⁺F4/80⁺CD11c⁻ cells followed by a quantification of dead macrophages (Ann V⁺/PI⁺). Concentration of extracellular dsDNA in the macrophages supernatant. Bars: 10 µm.

(B) WT and STING^{-/-} bone marrow dendritic cells were transfected with c-di-AMP, Annexin V/PI flow cytometry analyzed and extracellular dsDNA in the supernatant quantified as in (A).

*p < 0.05, **p < 0.01, ***p < 0.001, ****p < 0.0001 (Kruskal-Wallis test followed by Dunn post-test). Data are presented as mean ± SEM of n= 3-5 independent cultures and are representative of three independent experiments.

Minor concerns:

1. *The work presented here is commendable in that both macrophages and dendritic cells are queried in the experiments, however the figures lack clearly defined labels for cell type and it is ambiguous as to which is used, as both are often presented side by side. Additionally, there are several inconsistencies in labels, presentation of statistical significance, and a lack of information of abbreviations used in the table, which collectively diminish the clarity of the information presented. The authors should indicate which panels are macrophages and which are dendritic cells through labels or color/pattern differences, unify the manner in which statistical significance is presented, and create a legend for the table to enhance readability. Moreover, the authors should indicate which clinical samples noted in the table are used in Fig 2a. blots.*

As suggested, to improve clarity, we added labels for each cell type used in the figures, we revised the labels per panel, and homogenized the presentation of statistical significance.

We added the abbreviations used in Table 1 in the legend.

We now indicate in Figure 2 legend (now Fig.2d) the clinical samples noted in the Table (now Table 2) used for immunoblotting.

Revised Table 2 legend:

Table 2. Clinical characteristics of patients with fibrotic interstitial lung disease

FEV1, forced expiratory volume at 1.0 second; FEV1%, ratio of FEV1 to FVC, forced vital capacity; FEV1% predicted, FEV1% of the patient divided by the average FEV1% in the equivalent population; PY, pack year; Y cess, years of smoking cessation; IPF, idiopathic fibrosis; n.d., not determined.

Revised Figure 2 legend: page 41

"d: immunoblots shown from left to right for ILD patients H, G, C, A (see Table 2), representative of n= 3 cortisone-treated ILD patients and n=18 untreated-ILD patients, from 3 independent experiments, shown in d-f."

2. *There are several grammatical and spelling errors present in the text (i.e. oxidative is misspelled as oxydative).*

We verified and corrected the text throughout

Reviewer #2 (Lung inflammation, nanomedicine)(Remarks to the Author):

This is an interesting and detailed investigation of the role of STING in the development of silica-induced fibrosis. Although STING pathways are increasingly being described to be important in lung disease, this observation is, as far as I know, novel and will be of interest to scientists in the field. The authors have used human material and both animal models and in vitro methods to address their questions.

The authors show that the STING pathway is activated in human IPF tissue and correlates with epithelial damage; I may have missed this, but it is not clear how epithelial damage was evaluated. This needs to be clarified and proven to be rigorous and not subjective. It is also important that the reader appreciates that these individuals have not been diagnosed with silicosis. That this phenomenon is likely related to the disease process, rather than specifically silica exposure.

We appreciate the comments of the reviewer, and had initiated a collaboration to get access to samples from a cohort of patients with silicosis. This cohort of patients formerly working with sand blasting in the denim textile industry in Turkey, is quite unique, with a prevalence of silicosis >96% (Akgun et al., 2015). Indeed the patients are young men, with a defined intense exposure period for ca 4±2 years, and a rapid progression of the disease, often fatal. Because these young adults are in a very fragile condition, invasive procedures could not be performed. Nevertheless, a crucial point was to verify whether self dsDNA was increased in these patients heavily exposed to silica, and whether the type I interferon pathway was activated.

We now report in the new Figure 1a-b that the levels of circulating dsDNA are increased in 21 patients as compared to 10 healthy controls. Because rather elevated circulating dsDNA was found in 2 of the controls, the results are at the border of statistical significance. Interestingly, the level of CXCL10, a type I IFN regulated cytokine, was increased in the sputum of silicosis patients as compared to healthy controls. It should be noted that the exposure to silica occurred in the years 1996-2004, so 10-20 years ago in these patients. The demographic and clinical characteristics of the patients with silicosis and healthy controls are summarized in Table 2. The data are presented in the new Figure 2a,b.

New Figure 2a,b:

New Figure 2 a,b legend: page 41

"Figure 2. Presence of circulating DNA and CXCL10 in the airways of silicosis patients, STING and type I IFN pathway activation in ILD patient lungs and in silica-exposed human PBMCs.

(a,b) Presence of (a) dsDNA in the plasma and (b) CXCL10 in the sputum of patients with silicosis (ILO 5-12, see Table 1), as compared with healthy individuals (ILO 0)."

Revised Result section on page 7:

"Circulating dsDNA and increased CXCL10 secretion in silicosis patients.

Although mining and stone industry were traditional sources of silica exposure, new sources such as sand blasting in the denim industry outgrows the expected prevalence of silicosis to over 96% (Akgun 2015). Unique characteristics of these patients include their young age, a definite start of exposure, an intense exposure for ca 4±2 years, and a rapid progression of the disease, often fatal (Akgun 2015; 2018). Here, we investigated a cohort of 21 young men suffering from silicosis after exposure to silica in denim industry in the mid 2000 (Table 1). We hypothesized that silica might have caused airway cell damage and release of self DNA able to trigger the type I IFN response, as in the silica murine model. Interestingly, patients with silicosis exhibited increased plasma levels of dsDNA (Figure 2a). Further, in their sputum, the concentration of CXCL10 was significantly higher than in healthy controls (Figure 2b). Thus, in silicosis patients, a previous intense exposure to silica leads to high dsDNA circulating levels and trigger of CXCL10 response."

Revised Discussion on page 19:

“Importantly, silicosis patients had increased circulating dsDNA and CXCL10 in sputum,”

Revised Abstract:

“Silicosis patients exhibited circulating dsDNA and increased CXCL10 in sputum”

New Table 1:

Patient	Age	First silica exposure year	Exposure duration (month)	FEV1 (% predicted)	FVC (% predicted)	FEV1/FVC ratio	Chest X-ray Severity score	Chest X-ray Large opacities	Smoking history (Pack Years)
C1	38	NA							15
C2	43	NA							5
C3	29	NA							2
C4	27	NA							0
C5	29	NA							2
C6	40	NA							18
C7	38	NA							10
C8	40	NA							9
C9	42	NA							15
C10	28	NA							0
P1	29	2003	72	83	83	86	8	-	8
P2	36	1996	48	62	93	56	11	A	15
P3	35	2000	60	42	68	53	12	C	7
P4	28	2000	60	44	90	40	11	A	0
P5	34	2002	48	65	73	75	12	-	9
P6	31	2001	60	58	76	66	10	B	20
P7	38	1996	24	92	102	75	11	-	5
P8	34	1996	84	73	91	84	9	-	25
P9	37	2000	60	64	78	68	12	C	5
P10	36	2004	24	96	110	74	11	-	7
P11	36	2001	24	78	87	76	9	-	10
P12	34	1999	41				8	-	5
P13	38	2003	24	78	95	69	7	-	40
P14	49	2000	60	76	102	60	5	-	40
P15	33	2001	12	92	91	85	9	-	17
P16	36	1997	72	39	57	58	12	B	0
P17	30	2000	60	53	73	62	12	B	6
P18	29	2003	24	66	83	69	10	A	15
P19	33	2001	48	42	48	73	12	C	10
P20	35	2000	60	38	71	45	12	C	5
P21	34	2000	36	18	33	46	12	C	1

Table 1. Silicosis patient clinical characteristics.

All 21 patients with silicosis (P1-P21) were males, as were the 10 healthy controls (C1-C10).

Radiological severity score according to International Labor Organization (ILO) small profusion subcategories (1-12, 12 represents the most extensive disease, which is 3/+, highest profusion of small opacities subcategory).

A large opacity is defined as an opacity having the longest dimension exceeding 10 mm. The size increases from A to C and reflects severity of radiological involvement.

FEV1, forced expiratory volume at 1.0 second; FEV1%, ratio of FEV1 to FVC, forced vital capacity; FEV1% predicted, FEV1% of the patient divided by the average FEV1% in the equivalent population; NA: not applicable in the healthy controls.

Epithelial damage was evaluated by light microscopy on pneumonectomy samples before lung transplantation by a trained pathologist (B.R., MD, board certified pathologist). The evaluation of epithelial damage was based on semi-quantitative scoring of condensation of the cytoplasm, chromatin clumping with effacement of nuclear structure, detachment of epithelial cells, with desquamation of necrotic/apoptotic epithelial cells in the airspace. Further a massive interstitial fibrosis and focal, essentially monoclonal cell infiltration was observed.

A typical H&E staining showing typical epithelium damage is now shown in Figure 2f:

Revised Figure 2f legend: page 41

“(f) Correlation between STING dimers vs β -actin protein ratio and epithelial damage scored on human histological tissue sections of these patients. The pathological scoring was established regarding the presence of necrotic cells, desquamation of epithelial barrier, denudation and flattening of the epithelial barrier (indicated by arrows). Control lung tissue area (left, patient G in Table 2) and fibrotic area (right, patient D in Table 2) are shown (scale bar 100 μ m).

c-f: c: confocal image representative of 3 patients; d: immunoblots shown from left to right for ILD patients H, G, C, A (see Table 2), representative of n= 3 cortisone-treated ILD patients and n=18 untreated-ILD patients, from 3 independent experiments, shown in d-f. ”

In fact, the details of patient demographics describe fibrosis or IPF; the difference between these need to be shown. The term IPF suggests nothing is known of the cause of the lung fibrosis, but what about the other patients?

IPF is a clinical term, morphologically manifested by interstitial fibrosis, indistinguishable from particle or toxin induced lung fibrosis. The pathologist use both the clinical and morphological terms indifferently. The only correct morphologic denomination is fibrosis.

We verified with the clinicians and pathologists and confirm that the patients (now listed in Table 2) had no exposure to silica and the cause of lung fibrosis is unknown. All patients had morphological histological signs of the DPLD Group of Interstitial Lung Disease (ILD), while idiopathic pulmonary fibrosis (IPF) in the study Group was defined as the matching of histology of usual Interstitial Fibrosis (UIP) and the HR-CT diagnosis of UIP. All others were probable UIP in HRCT scans.

Thus, fibrotic ILD might better describe the clinical diagnosis and we modified Table 2 and the manuscript text accordingly.

There are emerging studies of the role of STING in pulmonary fibrosis which should be included in the discussion/introduction(eg Qiu et al, Front Immunol 2017, 8:1756; Liu et al NEJM 2014, 371(6):507; Nathan et al Curr Opinions in Pulm Med 2018, 24:253). Indeed, the latter includes the role of genetic status of STING in lung fibrosis.

We included a discussion of recent articles involving STING in pulmonary fibrosis in the Discussion section on page 22:

“Self DNA release including mitochondrial DNA has been implicated in various pathologies such as infections, cancer, autoimmune or inflammatory diseases, with so far limited implication of the STING pathway (West and Shadel, 2017). Extracellular mitochondrial DNA was recently detected in the BALF of idiopathic pulmonary fibrotic patients (Ryu et al., 2017), and mtDNA concentration in plasma identified as a marker of disease progression. However, no mechanistic link has been discussed between mtDNA release and disease. On the other hand, STING has been implicated in pediatric interstitial lung disease (Nathan et al., 2018), and gain of function mutations in Tmem173 were identified in a new autoinflammatory syndrome (STING-associated vasculopathy with onset in

infancy, SAVI), associated with interstitial lung disease in 5 out of 6 patients (Liu et al., 2014). Here we show that silicosis patients with a history of intense silica exposure had increased circulating dsDNA and CXCL10 levels in sputum. Further, we document STING activation in the lung of IPF patients. Airway silica exposure induced self dsDNA release that triggered STING-dependent type I IFN responses in mice. We propose that the DNA present in the airways activates STING pathway and contributes to the progressive lung inflammation.”

The mouse studies indicate that release of self dsDNA, likely by dying macrophages, into the airways is an important mediator of STING activation and type 1 IFN pathway activation. This is supported by studies of human PBMCs. Much work is performed to establish whether the extracellular or intracellular dsDNA is responsible and evidence suggests that for macrophages, the extracellular dsDNA is important, possibly internalised during uptake/phagocytosis of dsDNA coated silica particles. Phagocytosis usually contains (endosomal) and destroys material that is taken up, so it is interesting that the authors think it is subsequently cytosolic and activating STING pathways. What would be the mechanism? Presumably, there is not a receptor for dsDNA at the macrophage cell surface which internalises the DNA directly into the cytosol?

We further discuss this point in the Discussion part on page 20-21:

“ Furthermore, the question arises as to how self-dsDNA released extracellularly could be sensed by intracellular STING adaptor molecule in macrophages. Self-dsDNA co-localized with silica microparticles in the intracellular compartments of macrophages. Extracellular DNA bound to silica is likely transported from the extracellular milieu into macrophage cytosol⁴⁸. For instance, nucleic acid receptors could be involved in promoting extracellular DNA uptake after silica exposure. Indeed, RAGE protein has been shown to interact with DNA promoting DNA uptake by cells via the endosomal route, inducing Nf-κB activation through TLR9⁴⁹. Further, in pathological conditions, extracellular self-dsDNA can also get access to the cytosol through Fc receptors, forming a chromatin-immune complex able to be internalized by the host cell⁵⁰. Beside, CARGO proteins or peptides such as LL37 may contribute to dsDNA transport into the cell⁵¹. Micronuclei formed after silica exposure may also act as DAMPs activating type I IFN response in a STING-dependent manner. Indeed, Mackenzie et al. recently reported a role for DNA sensors in sensing micronuclei arising from genome instability or micronuclear envelope breakdown⁵². Micronuclei formation might occur in macrophages after silica exposure as free-dsDNA was found next to the nucleus. The remaining open questions include whether dsDNA-silica complexes activate STING and whether dsDNA dissociates from silica inside the cell.”

Regarding the in vitro macrophage models, a relatively large amount of silica was used in the exposure and it is perhaps not surprising that there was significant programmed cell death. Normally, apoptotic cells would be internalised by efferocytosis by macrophages. My understanding is that the DNA would not normally be released into the extracellular compartment. Is efferocytosis compromised by silica? So that the cells go into secondary necrosis and release DNA? Certainly the videos show cells stuffed with particles, which one assumes are present in exposed lungs, but which are supplemented by recruitment of fresh macrophages to the affected site.

The authors show that DCs behave differently with respect to STING pathway induction, where mitochondrial DNA drives the process. It should be noted that the authors have recently published on DC and the mechanism of STING activation (J Innate Immunology 2018, 10:239) and some of the work is similar, though using different environmental factors. How likely would they be exposed to silica particles of this size, and would they internalise the particles in vivo? Because it would seem that this is part of the process leading to mtDNA release and STING activation. Which of the cellular responses is most important - macrophage or DCs? And importantly, are there other cells at the gas-liquid interface (eg epithelial cells) that might also have the STING pathway that would be therapeutic targets (particles do not need to get into cells to activate them)?

Whether efferocytosis itself is compromised by silica is not known, but it is well reported that silica induces necrosis (Joshi and Knecht, 2013; Kawasaki, 2015). In our recent article (J Innate Immunology 2018, 10:239) we addressed the role of the cGAS/STING pathway in the host response to *Mycobacterium tuberculosis* infection. In Figure 1a we used cGAMP as a control for stimulating DCs and compared it to M.tb DNA or M.tb infection itself. This is quite different from addressing the effects of silica exposure here, using c-di-AMP as a positive control to activate STING pathway. We showed that DCs do internalize silica particles in vitro, thus it is likely that they would also do it in vivo, although this has not been precisely documented, to the best of our knowledge.

To address the respective relevance of macrophages or DCs, versus other cells such as epithelial cells, we performed experiments in bone marrow chimera mice where the hematopoietic system has

been replaced by either wild-type or STING^{-/-} bone marrow transfer into wild-type or STING^{-/-} recipient mice. We show that the inability of STING deficient mice to activate type I pathway in response to silica was corrected by hematopoietic reconstitution with wild-type bone marrow cells and, conversely, this phenotype was partially transferred to wild-type mice after reconstitution with STING^{-/-} bone marrow. These new results are presented in supplementary Figure 9 and described in the results section on page 18.

New results section on page 18:

"Hematopoietic cell-derived STING is crucial for silica-induced cell death and lung inflammation.

STING protein is expressed in endothelial and epithelial cell types, as well as in hematopoietic cells including macrophages, T cells and DCs(Barber, 2015). To study the cellular source of STING leading to silica-induced lung inflammation, bone marrow reconstituted STING^{-/-} or wild-type mice were exposed to silica microparticles (Supplementary Fig. 9). The inability of STING deficient mice to activate type I pathway in response to silica was corrected by WT hematopoietic cell reconstitution and was partially transferred to WT mice after reconstitution with STING^{-/-} bone marrow cells. Indeed, silica-induced cell death, neutrophil recruitment in the airways, type I IFN expression and the release of CXCL10 and TNF were reduced after reconstitution of WT mice with STING^{-/-} hematopoietic cells (Supplementary Fig 9a-k). Conversely, the reconstitution of STING^{-/-} mice with WT BM restored the response to silica in terms of cell death, neutrophil recruitment in the airways, type I IFN expression and CXCL10 and TNF release (Supplementary Fig 9 c-k). Self-dsDNA release in the BALF after silica exposure was unaffected in the chimeric mice (Supplementary Fig. 9i), similar to STING^{-/-} mice (Fi. 3i). Thus, STING pathway in hematopoietic cells is crucial for silica-induced type I IFN response and lung inflammation. Nevertheless, the role of STING in stromal cells after silica exposure should not be excluded, since STING competent stromal cells provide some CXCL10 response in WT mice reconstituted with STING^{-/-} bone marrow mice as compared with total STING deficiency (Supplementary Fig. 9g)."

New supplementary Figure 9:

New Suppl Fig 9 legend:

Supplementary Figure 9. Hematopoietic STING drives silica-induced lung inflammation

STING^{-/-} mice and WT mice were lethally irradiated and reconstituted with bone marrow from either WT mice or STING^{-/-} mice before silica microparticles (1 mg/mouse i.t.) or saline administration. Parameters were analyzed on day 7 post-exposure.

(a-b) Control of hematopoietic reconstitution. (a) Gating strategy for flow cytometry analysis. Pre-gated singlet cells using FSC-A/FSC-H followed by SSC-A/SSC-H and FSC-W/FSC-A. (b) Whole blood staining using CD45.1 for WT Ly5.1 cells and CD45.2 for STING^{-/-} mice.

(c) Dot plots showing Annexin V/PI staining of lung cells and (d) proportion of dead lung cells (Ann V⁺/PI⁺) in the different groups.

(e) Neutrophils count in the BALF. Lung *Ifnβ1* transcripts quantified by real-time PCR and (g) CXCL10 and (h) TNF levels measured by ELISA.

(i) Concentration of extracellular dsDNA in the BALF acellular fraction.

Dot plots (j) and (k) proportion of CD45⁺ Ly6G^{high} Ly6C^{int} granulocytes in the lungs (on 200 000 events).

*p < 0.05, **p < 0.01, ***p < 0.001, ****p < 0.0001 (one-way ANOVA with Tukey's *post hoc* test). Data are presented as mean ± SEM (mice per group: n = 4 (WT BM → WT NaCl), n = 5 (WT BM → WT Silica), n = 4 (STING^{-/-} BM → WT NaCl), n = 5 (STING^{-/-} BM → WT Silica), n = 3 (WT BM → STING^{-/-} NaCl), n = 4 (WT BM → STING^{-/-} Silica), n = 3 (STING^{-/-} BM → STING^{-/-} NaCl), n = 5 (STING^{-/-} BM → STING^{-/-} Silica). Each symbol represents an individual mouse."

Whilst the authors have meticulously taken the reader through a barrage of studies to establish the import of STING in the fibrotic pathway, there is no control for the effects of these particles, such as a non-fibrogenic (or less fibrogenic) particulate. Consequently, it is not clear whether this is an effect of particles, or the unique effect of crystalline silica.

To address this question we compared the effect of Diesel Exhaust Particles, with those of crystalline silica. Intra-tracheal exposure to Diesel Exhaust Particles, Polycyclic Aromatic Hydrocarbons 16 (DEP-PAH, 10ug i.t., 24h) induced a strong recruitment of neutrophils in the bronchoalveolar space, together with a strong expression of CXCL10 in the BAL fluid, similar to what is seen after silica, in wild-type mice. However, neither DEP-PAH induced neutrophil recruitment nor CXCL10 production were affected by the absence of STING pathway in STING^{-/-} mice (see below). This is in contrast to the response to silica microparticles, which was significantly reduced in STING^{-/-} mice, including in terms of neutrophil recruitment and CXCL10 production (supplementary Figure 3). Thus, the effect seen with crystalline silica are not seen with other types of particles

Figure: Neutrophils and CXCL10 in the BALF of wild-type, STING^{-/-} and cGAS^{-/-} mice exposed to DEP-PAH (10ug i.t.) for 24h.

There are many typographical errors that need to be addressed
We apologize for this and rechecked the whole manuscript.

Macrophages are notoriously difficult to transfect. More details of the measure of efficacy are needed.

Because macrophages are difficult to transfect, we worked with bone marrow cells differentiated for 7 days in the presence of M-CSF plus 20% horse serum. The cells were treated with silencing RNA targeting IFI204, DDX41 and non-targeting RNA diluted in Dharmacon ACCELL siRNA delivery media. There was no actual transfection step. At that stage, the bone marrow progenitors still undergo differentiation and proliferation, and the cells were still maintained for 4 days in culture, to ensure an efficient siRNA knock-down, as recommended by the manufacturer, and a full differentiation of the macrophages. While there was a clear effect of IFI204 and DDX41 siRNA on type I interferon and CXCL10 induction by silica or cDN, they did not affect the level of TBK1 or IRF3 protein expression, as measured by immunoblot.

More details have been added in the Suppl Figure 8 legend:

“(b) WT bone marrow cells were treated with silencing RNA targeting IFI204, DDX41 and non-targeting RNA after harvesting on day 7 of culture in DMEM medium supplemented with 10% FCS, 100 U/mL penicillin and 100 µg/mL streptomycin, 2 mM glutamine, plus 20% horse serum and 30% (v/v) L929 conditioned medium as a source of M-CSF. On day 11, after further 4 days in culture, BMDMs were stimulated as in (a). The purity of the BMDMs culture was verified by flow cytometry on day 7 using CD11b, CD11c and F4/80 staining: 93.7±0.3% CD11b⁺F4/80⁺CD11c⁻ cells.

It is not always clear how many samples were used for Western blotting experiments and not always shown bar graphs and significances of the Western blotting work.

We now added the missing information on how many samples were analyzed by Western blotting experiments in the Figure Legends of Figure 1j,p,r; 2d,l; 3c,j; 4e,g; 5d,f; 6g, and in Supplementary Figure Legends of Supplementary Fig. 2f; Supplementary Fig. 5j; Supplementary Fig. 8c; We added a quantitative bar graph and significance analysis of the new Western blot for cleaved Caspase 3, cleaved Gasdermin D and phosphorylated MLKL in Figure 1k-m, STING dimer in Figure 1q, Figure 2d and Supplementary Fig. 5l, and NLRP3 in Supplementary Fig. 5j (see response to reviewer 1 point 4 and 6). We added such analyses for the main results, considering that all WB data are presented in parallel with quantitative real-time PCR data, excepted for phosphorylated protein where transcriptomic analysis will not necessary indicate an activation of the pathway.

Reviewer #3 (cGAS/STING signalling, DAMP/PAMP)(Remarks to the Author):

In this work Benmerzoug have explored the role of DNA in stimulation of pathological responses in silica-induced inflammation. The work is based on data from a strong mouse model, and the basic observation of a cGAS-STING dependent response upon silica treatment is strong. The subsequent cellular characterization of silica-induced cell death macrophages and DCs and macrophages reveals interesting differences, but is not fully developed. Finally, the attempt to perform detailed cellular/molecular characterization of the pathways for silica-induced cell death has significant flaws, and requires much more work to allow the conclusions drawn by the authors.

SPECIFIC POINTS:

1. Figure 1. The authors should conclusively establish the mode of cell death: apoptosis (cleaved caspase 3, DNA fragmentation), necroptosis (pMLKL), pyroptosis (cleaved GasderminD).

As suggested, we performed new experiments to better assess the mode of cell death.

We first assessed Caspase 3 cleavage as a marker of apoptosis: we show that indeed silica exposure induces caspase 3 cleavage, as determined by immunoblotting, in wild-type mice. However, there was no cleavage of caspase 3 in the lung of STING^{-/-} mice exposed to silica. This indicates that STING pathway is required for silica-induced apoptosis. The new data on silica/STING induced cell death has been added in the Figure 1j-m and Figure 3j.

Second, cytotoxicity of crystals such as silica involves RIPK3-MLKL-mediated necroptosis (Desai et al., 2017; Mulay et al., 2016). We thus performed new experiments to appreciate necroptosis-mediated cell death through phosphorylation of MLKL *in vivo* after silica exposure. Indeed, silica exposure induced MLKL phosphorylation in the lung of wild-type mice. Necroptosis is induced by silica in a STING-dependent manner since the absence of STING led to lower phosphorylation of MLKL. The new data has been added in the figure 1j-m and figure 3j.

Third, although the role of NLRP3 in inducing pyroptosis/apoptosis is well documented (Bergsbaken et al., 2009; Hou et al., 2018; Wree et al., 2014), the role of pyroptosis through gasdermin D cleavage after *in vivo* silica exposure has not been investigated. Here, we now show that silica exposure leads to gasdermin D cleavage that is independent of STING signaling, showing that STING is not involved in pyroptosis programmed cell death after silica exposure. The new data has been added in Figure 1j-m and Figure 3j.

New Figure 1j-m:

New Figure 1j-m legend: page 39

(j) Immunoblots of Caspase 3 (Casp3), cleaved Caspase 3 (c-Casp3), Gasdermin D (GSDMD), MLKL and phosphorylated MLKL (p-MLKL), normalized to β-actin protein. Immunoblots relative quantifications of (k) c-Casp3, (l) c-GSDMD and (m) p-MLKL."

Revised Result section on page 6:

"To better characterize the mode of cell death induced by silica microparticles, we first assessed caspase 3 cleavage as a marker of apoptosis. We show that silica exposure induced caspase 3 cleavage in WT mice, indicative of silica-induced apoptosis (Fig. 1j). The contribution of pyroptosis

and necroptosis in silica-induced cell death was assessed by immunoblots of cleaved gasdermin D (GSDMD) and phosphorylated Mixed Lineage Kinase domain Likepseudokinase (MLKL) (Figure 1j). Indeed, silica exposure induced the cleavage of GSDMD and phosphorylation of MLKL in the lung of WT mice (Figure 1j). In summary, airway silica exposure induces cell death through apoptosis, necroptosis and pyroptosis, with nuclear and mitochondrial self-dsDNA release, triggering type I IFN pathway activation."

New Figure 3j :

New Figure 3j legend: page 43

"(j) Immunoblots of Caspase 3, cleaved Caspase 3, Gasdermin D, MLKL and phospho-MLKL in the lung of WT and STING^{-/-} mice with β -actin as a reference."

New Result section on pages 10-11:

"Further, as cell death is induced in a STING-dependent manner after silica exposure, we questioned whether STING could be involved in apoptosis, necroptosis and/or pyroptosis related cell death. Indeed, silica induced apoptosis and necroptosis in a STING-dependent manner since caspase 3 cleavage and MLKL phosphorylation were highly reduced in STING^{-/-} mice (Fig. 3j). By contrast, silica-induced Gasdermin D cleavage was still present in STING^{-/-} mice, showing that STING is not essential for pyroptosis cell death after silica exposure (Fig. 3j). These results show that silica induces necroptosis and apoptosis in a STING-dependent manner *in vivo*."

2. The functional immunological data should be complemented with results on how cGAS and STING-deficiency impacts on disease development upon silica-treatment:

As suggested we performed experiments to characterize disease development 35 days after silica exposure in STING- and cGAS-deficient mice. We now present histology data, together with inflammation cell recruitment in the airways, pulmonary CXCL10 and extracellular dsDNA in the BALF on day 35 in the new supplementary Figure 3b-c. We show that type I IFN-driven lung inflammation is strongly reduced 35 days post-silica exposure in both STING^{-/-} and cGAS^{-/-} mice (Supplementary Fig 3b). Moreover, histological analysis showed lower cell infiltration and granuloma in the absence of cGAS/STING pathway at day 35 post-silica exposure (Supplementary Fig 3c). The new data has been added in the results section on page 10.

New Supplementary Figure 3b,c:

New Supplementary Fig 3b,c legend:

“(b-c) Silica microparticles (1 mg/mouse i.t.) or saline vehicle were administered to WT, STING^{-/-}, and cGAS^{-/-} mice and parameters analyzed on day 35 post-exposure. (b) Concentration of extracellular dsDNA in the bronchoalveolar lavage fluid (BALF) acellular fraction. Lung levels of *Ifna4* and *Ifnβ1* transcripts measured by RT-qPCR, and CXCL10 by ELISA. (c) Lung HE staining at day 35 post-silica exposure (4X magnification). The pathological scoring was established regarding the presence of necrotic cells, cell infiltration and granuloma.”

New result section on page 10:

“STING pathway was still crucial at day 35 post-silica exposure, as type I IFN response was abolished in the absence of either STING or cGAS (Supplementary Fig. 3b). Interestingly, lung inflammation was also decreased in STING^{-/-} and cGAS^{-/-} mice, with reduced cell infiltration and granuloma at day 35 post-exposure (Supplementary Fig. 3c). Our findings indicate that STING drives lung inflammation both at early and later phases of silica exposure while the role of cGAS DNA sensor seems less prominent early after silica exposure.”

3. Figure 2a. Data from a healthy donor not receiving cortisone treatment should be shown. It is surprising that no STING expression (not even STING monomer) is observed in the cortisone-treated sample. Therefore, the data would also gain if STING immunoblot from a reduced SDS-PAGE was shown.

The frozen lung samples available for qPCR, immunoblot, Immunofluorescence analysis were pneumonectomy samples from patients undergoing lung transplantation. Unfortunately, we have no access to such lung samples from healthy donors.

The fact that no STING expression was detected in the samples from cortisone-treated patients could be due to the short 5 sec exposure time to UV light, used to visualize clearly band doublets for positive samples. We repeated a new series of immunoblots, with an extended exposure time of 15 sec, and now clearly show that low levels of STING monomer are present in the lung tissue from cortisone treated patients, with no STING dimer visible.

All immunoblots were carried out under semi-native conditions (1% 2-mercaptoethanol) to see STING dimer. As suggested, we verified that STING dimer are not visible in immunoblots from SDS-PAGE under 5% 2-mercaptoethanol reducing conditions (see below), as expected (Motani et al., 2015). This is now shown in the new Figure 1r.

New Figure 1r:

New Figure 1r legend: page 40

“(r) Immunoblots of lung homogenates from WT mice exposed to silica, subjected to electrophoresis under 5% 2-mercaptoethanol reducing conditions (2-ME; left panel) or 1% 2-ME semi-native conditions (right panel), with β -actin as a reference. Immunoblots are representative of n = 8 samples from 2 independent experiments (r),”

Revised results section on page 6:

“STING dimer immunoblot was visualized under semi-native conditions, while reducing conditions disrupt the dimeric form of STING (Fig. 1r).”

Revised M&M section on page 29:

“Total protein (30 μ g per sample) were heated 5 min at 95°C in low reducing conditions (1% 2-mercaptoethanol) unless otherwise stated, before loading on 6-15% polyacrylamide gel and run at 160V for 45 min using the Bio-Rad Mini-PROTEAN Tetra Cell.”

4. Figure 3b (Immunoblot data). There is no detectable STING expression in the WT, silica-sample. This is very surprising and somewhat worrying.

As for point 3, the absence of STING expression in the WT Silica – sample was due to the short 5 sec exposure time to UV light used to visualize clearly the band doublets in the STING positive samples. We now repeated a new series of immunoblots, with an exposure time of 15 sec and can visualize low levels of STING that are increased after silica exposure and absent in the lung samples from STING^{-/-} mice. We are thankful and added this new data in the revised Figure 3c.

Revised Figure 3c:

Revised Fig 3 c legend: page 43

“(c) Immunoblots of STING/IRF3 axis in the lung of WT and STING^{-/-} mice including phospho-STING, STING, phospho-TBK1, TBK1, phospho-IRF3, and IRF3, with β -actin as a reference.”

5. Figure 3c. The V+/PI- and V+/PI+ data from all conditions/cell types under investigation should be shown.

The V+/PI- and V+/PI+ data have been added for WT, STING^{-/-} and cGAS^{-/-} mice saline controls not receiving silica in the revised Figure 3e.

Revised Fig. 3e legend: page 45

“(e) Flow cytometry representative dot blots showing Annexin/PI staining of F4/80+CD11c- interstitial macrophages (IM), F4/80+CD11c+ alveolar macrophages (AM), CD11c+F4/80- dendritic cells (DC) and Ly6G+F4/80- neutrophils among CD45+CD11b+ cells in WT, STING^{-/-} and cGAS^{-/-} mice.”

6. The reduced cell death in the cGAS and STING KO cells is interesting, but the consequence (namely that silica-induced cell death and not only the downstream inflammatory response is dependent on cGAS-STING) is quite poorly characterized. This represents a weakness of the study.

We are grateful for this comment, which motivated us to further characterize whether silica-induced cell death is influenced in the absence of cGAS/STING pathway. Silica particles have been reported to drive programmed cell death through apoptosis (Leigh et al., 1997), necrosis (Joshi and Knecht, 2013) and necroptosis (Mulay et al., 2016). However, there was no evidence regarding *in vivo* pyroptosis after silica microparticles intratracheal administration.

We analyzed specific markers of cell death including Caspase 3 cleavage as a marker of apoptosis, Gasdermin D (GSDMD) cleavage as a marker of pyroptosis, and phosphorylated MLKL as a marker for necroptosis.

We now analyzed *in vitro* silica induced cell death in macrophages (new Figure 4e) and dendritic cells (new Figure 5d). *In vitro* silica exposure induced apoptosis and necroptosis in macrophages, as revealed by cleaved Caspase 3 and phosphorylated MLKL. Interestingly in STING^{-/-} macrophages, Caspase 3 cleavage was reduced while phosphoMLKL was still present after silica exposure.

In dendritic cells, silica induced Caspase 3 cleavage which was reduced in STING^{-/-} cells, while MLKL was not phosphorylated.

Gasdermin D was not cleaved after *in vitro* silica exposure, which may reflect the lack of inflammasome activation, due to the absence of first signal. Indeed, we could not detect IL-1b released *in vitro* in DC or Macro after silica exposure.

New Figure 4e: Macrophages

Revised Fig 4e legend: page 45

"(e) Immunoblots of Caspase 3, cleaved Caspase 3, Gasdermin D, MLKL and phospho-MLKL in WT and STING^{-/-} macrophages, with β-actin as a reference."

Revised Results section on page 12:

"Silica exposure induced apoptosis and necroptosis in macrophages, as revealed by cleaved Caspase 3 and phosphorylated MLKL (Fig. 4e). Interestingly in silica-exposed STING^{-/-} macrophages, Caspase 3 cleavage and phosphoMLKL were reduced, as compared to WT macrophages (Fig. 4e). Gasdermin D was not cleaved after *in vitro* silica exposure, which may reflect the lack of inflammasome activation, due to the absence of first NFκB signal (Fig. 4e)²⁵"

New Figure 5d: Dendritic cells

Revised Fig 5d legend: page 46

"(d) Immunoblots of Caspase 3, cleaved Caspase 3, Gasdermin D, MLKL and phospho-MLKL in WT and STING^{-/-} DCs with β-actin as a reference."

Revised Results section on page 13:

"Silica exposure induced STING-dependent apoptosis in DCs, as revealed by cleaved Caspase 3, reduced in STING^{-/-} DCs, while MLKL was not phosphorylated and Gasdermin D was not cleaved (Fig. 5d)."

We show in Figure 1j-m that *in vivo* silica exposure drives apoptosis (as seen by cleaved Caspase 3), necroptosis (with phosphorylated MLKL), but also pyroptosis (with cleaved Gasdermin D) in the lung tissue. (see the response to reviewer 1 point 4)

Interestingly, STING drives apoptosis after silica exposure, as there was less cleaved caspase 3 in STING^{-/-} mice. STING also drives necroptosis, as there was less phosphorylated MLKL in STING^{-/-} mice, as compared to WT mice. By contrast, Gasdermin D was cleaved in silica exposed STING^{-/-} mice similar to WT mice, indicating that pyroptosis is independent of STING pathway after silica exposure. The new data is presented in Figure 3j and in result section on page 10-11.

New figure 1j-m:

New Figure 1j-m legend: page 39

“(j) Immunoblots on Caspase 3 (Casp3), cleaved Caspase 3 (c-Casp3), Gasdermin D (GSDMD), MLKL and phosphorylated MLKL (p-MLKL), normalized to β -actin protein. Immunoblots relative quantifications of (k) c-Casp3, (l) c-GSDMD and (m) p-MLKL.”

Revised Result section on page 6:

“To better characterize the mode of cell death induced by silica microparticles, we first assessed caspase 3 cleavage as a marker of apoptosis. We show that silica exposure induced caspase 3 cleavage in WT mice indicative of silica-induced apoptosis (Fig. 1j). To assess the contribution of pyroptosis and necroptosis in silica-induced cell death, immunoblots for cleaved gasdermin D (GSDMD) and phosphorylated MLKL (Mixed Lineage Kinase domain Likepseudokinase) were performed (Figure 1j). Indeed, silica exposure induced the cleavage of GSDMD and phosphorylation of MLKL in the lung of WT mice (Figure 1j). In summary, airway silica exposure induces cell death through apoptosis, necroptosis and pyroptosis, with nuclear and mitochondrial self-dsDNA release, triggering type I IFN pathway activation.”

New Figure 3j:

New Figure 3j legend: page 43

“(j) Immunoblots of Caspase 3, cleaved Caspase 3, Gasdermin D, MLKL and phospho-MLKL in the lung of WT and STING^{-/-} mice with β -actin as a reference.”

New Result section on pages 10-11:

“Further, as cell death is induced after silica exposure in a STING-dependent manner, we questioned whether STING could be involved in apoptosis, necroptosis and/or pyroptosis related cell death. Indeed, silica induced apoptosis and necroptosis in a STING-dependent manner since caspase 3 cleavage and MLKL phosphorylation were highly reduced in STING^{-/-} mice (Fig. 3j). By contrast, silica-induced Gasdermin D cleavage was still present in STING^{-/-} mice, showing that STING is not essential for pyroptosis cell death after silica exposure (Fig. 3j). These results show that silica induces necroptosis and apoptosis in a STING-dependent manner in vivo.”

We also asked whether STING pathway influences silica-induced ROS. We now show that ROS induction after silica exposure is strongly reduced in STING^{-/-} macrophages. Using MitoTracker, we show that the ROS induced after macrophage silica exposure co-localize with mitochondria in wild-

type macrophages. By contrast, there is very little ROS induction after silica exposure in the absence of STING in *STING*^{-/-} macrophages. Thus, STING pathway is essential for ROS induction in response to silica.

We propose that silica may first induce some mechanical cell death within few hours, as seen in our supplementary video, inducing a first DNA release susceptible to activate STING. STING pathway activation then leads to ROS induction, and more cell death, in an amplification loop. Indeed, agonists of type I IFN pathway and/or STING activation have been identified as important mediators to target and eliminate cancer cells (Burnette et al., 2011; Deng et al., 2014; Gulen et al., 2017; Li and Chen, 2018).

New figure 4a,b:

New Figure 4a Legend: page 45

“(a-b) WT and *STING*^{-/-} BMDMs were unstimulated or stimulated with silica (250 µg/mL) for 18 hr. (a) Brightfield confocal microscopy showing intracellular silica microparticles and MitoTracker (green) labelled mitochondria co-localizing with superoxide production detected by MitoSOX staining (red).”

New result section on page 12:

“Following silica stimulation, macrophages underwent cellular stress with reactive oxygen species (ROS) production detected using MitoSOX immunofluorescence which co-localized with mitochondria revealed by MitoTracker (Fig. 4a,b). Interestingly, the induction of ROS after silica stimulation was impaired in *STING*^{-/-} macrophages (Fig. 4a, b).”

7. Figure 4b. The authors show that the levels of free DNA is lower after silica-stimulation in *STING* KO cells. Is this data not in conflict with the data in the final display item in Figure 3b? Or does the *in vitro* system not mimic the *in vivo* situation?

We showed that *in vivo* the absence of STING leads to some reduction of cell death in specific cell types such as macrophages, DC or neutrophils (Fig 3e-h), although this does not translate into a reduction of dsDNA in the BALF (Fig 3i). This suggests that other cells contribute as sources of dsDNA

(e.g. lymphocytes, airway cells in bronchial and bronchiolar epithelium, fibroblasts, pulmonary vascular cells ...).

In vitro however, the reduced level of dsDNA released in STING KO cells after silica exposure (Fig4d) is the direct reflect of the reduced macrophages death occurring in the absence of STING (Fig4c).

8. Figure 5. As I understand it, the authors proposed that silica-induced ROS formation triggers STING-dependent cell death. What is the effect of ROS inhibitors/scavengers on silica-induced cell death?

This is an interesting point which has already been addressed in the literature: ROS inhibition (NAC) leads to a decrease of cell death through inhibition of caspase 1 cleavage after *in vivo* silica exposure (Bauernfeind et al., 2011). Furthermore, Zhang et al., showed that *in vivo* NAC treatment decreases significantly silica-induced lung fibrosis in rat (Zhang et al., 2013). Finally, another study also showed that NAC inhibits ROS activity leading to an inhibition of the mitochondrial apoptosis pathway *in vivo* (Zhang et al., 2014).

We are thus confident, based on the available literature, that ROS inhibitors decrease silica-induced cell death.

9. Figure 5c. The IF data on Draq5-STING colocalization should be quantified.

We apologize for a misleading wording of the original sentence. We reported that DCs stimulated with silica exhibited multiple DNA-containing sub-cellular structures inside the cytosol that co-localised with silica microparticles. The silica stimulated DCs also showed overexpressed STING and speck formation (Fig. 5c). However we do not believe that STING actually colocalizes with Draq5-DNA, since STING itself does not bind DNA (Barber, 2015). We modified the sentence on page 10 to clarify this point.

Revised Result section on pages 13-14:

"To examine whether dsDNA derived from DCs could be sensed by STING, DCs were double stained with DNA dye Draq5 and STING specific antibody. Interestingly, DCs stimulated with silica exhibited multiple DNA-containing sub-cellular structures inside the cytosol that co-localized with silica microparticles. There was a strong overexpression of STING and speck formation in these cells, beyond the perinuclear region (Fig. 5 e)."

10. Figure 6. The characterization of the difference in the response by macrophages and DCs is interesting but somewhat superficial. For instance, in panel 6, how do the authors explain that EtdBr treatment abolishes STING expression, but still allows response to cDNs? As a very minimum, the transcript data should be complemented by immunoblot data.

As suggested, we verified by immunoblot the activation of the STING pathway, including phosphorylated STING and cGAS, in the presence of EtdBr in dendritic cells stimulated by cDNs. The new Figure 6g shows that while EtdBr treatment reduces STING protein overexpression and dimerization in response to silica, the response of DCs to cDN under EtdBr treatment was preserved.

In addition, we verified that EtdBr decreased mtDNA without affecting nuclear DNA, as shown in Figure 6e,f.

New Figure 6e-g:

Revised Figure 6e-g Legend: page 48

"Fold change of (e) mitochondrial DNA (mMitoF1 and mMitoR1) and (f) nuclear DNA (mB2MF1 and mB2MR1) in untreated versus EtdBr-treated DC exposed to silica, as compared to untreated unstimulated cells.

(g) Immunoblot showing cGAS, phospho-STING, STING protein expression and dimerization in DCs after EtdBr treatment, with β-actin as a reference. "

Revised Result section on page 15:

"EtdBr-treated DCs displayed a drastic reduction of mtDNA without affecting nuclear DNA (Fig. 6e, f). STING protein expression, phosphorylation and dimerization in response to silica was strongly reduced in DCs after EtdBr treatment, while the response to control c-di-AMP (cDN) was spared (Figure 6g). cGAS protein levels were similar after silica or cDN stimulation, without or with EtdBr treatment (Fig. 6g)."

11. *Supplementary Figure 7. This part is very underdeveloped. Non-cGAS cytosolic DNA sensors are very controversial, particularly in mice (e.g. Immunity 45(2):255-66). Therefore, the results presented should be confirmed/replaced by data from KO cells (e.g. using CRISPR). Moreover, the authors should show that the proposed sensors colocalize with DNA in the cytoplasm.*

Here we wanted to propose an explanation for the difference in cGAS dependence between dendritic cells and macrophages. Even if non-cGAS cytosolic DNA sensors are very controversial in mice, we provide some evidence that other DNA sensors such as IFI204 or DDX41 might contribute to DNA sensing after silica exposure in macrophages.

Clearly a whole study would be needed to identify the DNA sensors involved, and those dispensable, as in the Immunity 45(2):255-66 article, especially since new co-sensors are still being discovered, as illustrated with the recently described ZCCHC3 co-sensor for ds DNA recognition (Lian et al., 2018).

Moreover, we added this point to the discussion part:

"STING binds cDN, but whether it directly interacts with dsDNA is still a matter of debate 34, 35. One of the main cytosolic DNA sensors upstream of STING identified is cGAS, but many studies reported DDX41, IFI16 or mouse IFI204, DAI, RNA pol III, LRRFIP1, ZCCHC3 or meiotic recombination 11 homolog A (MRE11) as cytosolic DNA sensors able to activate type I IFN pathway in a cGAS-independent STING-dependent manner^{36, 37,38,39, 40, 41, 42, 43, 44}."

MINOR POINTS

12. *It is very unusual that data referred to in the abstract are not shown in the main figures (data on DDX41 and IFI204).*

We removed the mention to DDX41 and IFI2014 from the abstract.

13. *Each display item should have its own panel. It is very difficult to follow the data presentation in the text.*

We revised the label of the panels displayed, to facilitate data presentation in the text, especially for Figure 1 and 2.

References

- Akgun, M., Araz, O., Ucar, E.Y., Karaman, A., Alper, F., Gorguner, M., and Kreiss, K. (2015). Silicosis Appears Inevitable Among Former Denim Sandblasters: A 4-Year Follow-up Study. *Chest* *148*, 647-654.
- Barber, G.N. (2015). STING: infection, inflammation and cancer. *Nat Rev Immunol* *15*, 760-770.
- Bauernfeind, F., Bartok, E., Rieger, A., Franchi, L., Nunez, G., and Hornung, V. (2011). Cutting edge: reactive oxygen species inhibitors block priming, but not activation, of the NLRP3 inflammasome. *J Immunol* *187*, 613-617.
- Bergsbaken, T., Fink, S.L., and Cookson, B.T. (2009). Pyroptosis: host cell death and inflammation. *Nat Rev Microbiol* *7*, 99-109.
- Burnette, B.C., Liang, H., Lee, Y., Chlewicki, L., Khodarev, N.N., Weichselbaum, R.R., Fu, Y.X., and Auh, S.L. (2011). The efficacy of radiotherapy relies upon induction of type I interferon-dependent innate and adaptive immunity. *Cancer Res* *71*, 2488-2496.
- Cassel, S.L., Eisenbarth, S.C., Iyer, S.S., Sadler, J.J., Colegio, O.R., Tephly, L.A., Carter, A.B., Rothman, P.B., Flavell, R.A., and Sutterwala, F.S. (2008). The Nalp3 inflammasome is essential for the development of silicosis. *Proc. Natl. Acad. Sci. U. S. A.* *105*, 9035-9040.
- Chamilos, G., Gregorio, J., Meller, S., Lande, R., Kontoyiannis, D.P., Modlin, R.L., and Gilliet, M. (2012). Cytosolic sensing of extracellular self-DNA transported into monocytes by the antimicrobial peptide LL37. *Blood* *120*, 3699-3707.
- Deng, L., Liang, H., Xu, M., Yang, X., Burnette, B., Arina, A., Li, X.D., Mauceri, H., Beckett, M., Darga, T., *et al.* (2014). STING-Dependent Cytosolic DNA Sensing Promotes Radiation-Induced Type I Interferon-Dependent Antitumor Immunity in Immunogenic Tumors. *Immunity* *41*, 843-852.
- Desai, J., Foresto-Neto, O., Honarpisheh, M., Steiger, S., Nakazawa, D., Popper, B., Buhl, E.M., Boor, P., Mulay, S.R., and Anders, H.J. (2017). Particles of different sizes and shapes induce neutrophil necroptosis followed by the release of neutrophil extracellular trap-like chromatin. *Sci Rep* *7*, 15003.
- Dostert, C., Petrilli, V., Van Bruggen, R., Steele, C., Mossman, B.T., and Tschopp, J. (2008). Innate immune activation through Nalp3 inflammasome sensing of asbestos and silica. *Science* *320*, 674-677.
- Gaidt, M.M., Ebert, T.S., Chauhan, D., Ramshorn, K., Pinci, F., Zuber, S., O'Duill, F., Schmid-Burgk, J.L., Hoss, F., Buhmann, R., *et al.* (2017). The DNA Inflammasome in Human Myeloid Cells Is Initiated by a STING-Cell Death Program Upstream of NLRP3. *Cell* *171*, 1110-1124 e1118.
- Gulen, M.F., Koch, U., Haag, S.M., Schuler, F., Apetoh, L., Villunger, A., Radtke, F., and Ablasser, A. (2017). Signalling strength determines proapoptotic functions of STING. *Nat Commun* *8*, 427.
- Hornung, V., Bauernfeind, F., Halle, A., Samstad, E.O., Kono, H., Rock, K.L., Fitzgerald, K.A., and Latz, E. (2008). Silica crystals and aluminum salts activate the NALP3 inflammasome through phagosomal destabilization. *Nat Immunol* *9*, 847-856.
- Hou, L., Yang, Z., Wang, Z., Zhang, X., Zhao, Y., Yang, H., Zheng, B., Tian, W., Wang, S., He, Z., and Wang, X. (2018). NLRP3/ASC-mediated alveolar macrophage pyroptosis enhances HMGB1 secretion in acute lung injury induced by cardiopulmonary bypass. *Lab Invest*.
- Joshi, G.N., and Knecht, D.A. (2013). Silica phagocytosis causes apoptosis and necrosis by different temporal and molecular pathways in alveolar macrophages. *Apoptosis* *18*, 271-285.
- Kawasaki, H. (2015). A mechanistic review of silica-induced inhalation toxicity. *Inhal Toxicol* *27*, 363-377.
- Leigh, J., Wang, H., Bonin, A., Peters, M., and Ruan, X. (1997). Silica-induced apoptosis in alveolar and granulomatous cells in vivo. *Environ Health Perspect* *105 Suppl 5*, 1241-1245.
- Li, T., and Chen, Z.J. (2018). The cGAS-cGAMP-STING pathway connects DNA damage to inflammation, senescence, and cancer. *J Exp Med* *215*, 1287-1299.
- Li, X., Yan, X., Wang, Y., Wang, J., Zhou, F., Wang, H., Xie, W., and Kong, H. (2018). NLRP3 inflammasome inhibition attenuates silica-induced epithelial to mesenchymal transition (EMT) in human bronchial epithelial cells. *Exp Cell Res* *362*, 489-497.
- Lian, H., Wei, J., Zang, R., Ye, W., Yang, Q., Zhang, X.N., Chen, Y.D., Fu, Y.Z., Hu, M.M., Lei, C.Q., *et al.* (2018). ZCCHC3 is a co-sensor of cGAS for dsDNA recognition in innate immune response. *Nat Commun* *9*, 3349.
- Lindhout, E., Lakeman, A., and de Groot, C. (1995). Follicular dendritic cells inhibit apoptosis in human B lymphocytes by a rapid and irreversible blockade of preexisting endonuclease. *J Exp Med* *181*, 1985-1995.
- Liu, Y., Jesus, A.A., Marrero, B., Yang, D., Ramsey, S.E., Sanchez, G.A.M., Tenbrock, K., Wittkowski, H., Jones, O.Y., Kuehn, H.S., *et al.* (2014). Activated STING in a vascular and pulmonary syndrome. *N Engl J Med* *371*, 507-518.
- Mackenzie, K.J., Carroll, P., Martin, C.A., Murina, O., Fluteau, A., Simpson, D.J., Olova, N., Sutcliffe, H., Rainger, J.K., Leitch, A., *et al.* (2017). cGAS surveillance of micronuclei links genome instability to innate immunity. *Nature* *548*, 461-465.
- Malik, A.N., Czajka, A., and Cunningham, P. (2016). Accurate quantification of mouse mitochondrial DNA without co-amplification of nuclear mitochondrial insertion sequences. *Mitochondrion* *29*, 59-64.
- Mao, Y., Daniel, L.N., Whittaker, N., and Saffiotti, U. (1994). DNA binding to crystalline silica characterized by Fourier-transform infrared spectroscopy. *Environ Health Perspect* *102 Suppl 10*, 165-171.
- Motani, K., Ito, S., and Nagata, S. (2015). DNA-Mediated Cyclic GMP-AMP Synthase-Dependent and -Independent Regulation of Innate Immune Responses. *J Immunol* *194*, 4914-4923.
- Mulay, S.R., Desai, J., Kumar, S.V., Eberhard, J.N., Thomasova, D., Romoli, S., Grigorescu, M., Kulkarni, O.P., Popper, B., Vielhauer, V., *et al.* (2016). Cytotoxicity of crystals involves RIPK3-MLKL-mediated necroptosis. *Nat Commun* *7*, 10274.
- Nathan, N., Borensztajn, K., and Clement, A. (2018). Genetic causes and clinical management of pediatric interstitial lung diseases. *Curr Opin Pulm Med* *24*, 253-259.

- Ryu, C., Sun, H., Gulati, M., Herazo-Maya, J.D., Chen, Y., Osafo-Addo, A., Brandsdorfer, C., Winkler, J., Blaul, C., Faunce, J., *et al.* (2017). Extracellular Mitochondrial DNA Is Generated by Fibroblasts and Predicts Death in Idiopathic Pulmonary Fibrosis. *Am J Respir Crit Care Med* 196, 1571-1581.
- West, A.P., and Shadel, G.S. (2017). Mitochondrial DNA in innate immune responses and inflammatory pathology. *Nat Rev Immunol* 17, 363-375.
- Wree, A., Eguchi, A., McGeough, M.D., Pena, C.A., Johnson, C.D., Canbay, A., Hoffman, H.M., and Feldstein, A.E. (2014). NLRP3 inflammasome activation results in hepatocyte pyroptosis, liver inflammation, and fibrosis in mice. *Hepatology* 59, 898-910.
- Xu, M.M., Pu, Y., Han, D., Shi, Y., Cao, X., Liang, H., Chen, X., Li, X.D., Deng, L., Chen, Z.J., *et al.* (2017). Dendritic Cells but Not Macrophages Sense Tumor Mitochondrial DNA for Cross-priming through Signal Regulatory Protein alpha Signaling. *Immunity* 47, 363-373 e365.
- Zhang, H., Yin, G., Jiang, H., and Zhang, C. (2013). High-dose N-acetylcysteine decreases silica-induced lung fibrosis in the rat. *J Int Med Res* 41, 1179-1186.
- Zhang, L., He, Y.L., Li, Q.Z., Hao, X.H., Zhang, Z.F., Yuan, J.X., Bai, Y.P., Jin, Y.L., Liu, N., Chen, G., *et al.* (2014). N-acetylcysteine alleviated silica-induced lung fibrosis in rats by down-regulation of ROS and mitochondrial apoptosis signaling. *Toxicol Mech Methods* 24, 212-219.

Reviewers' comments:

Reviewer #1 (Remarks to the Author):

Thank you for providing detailed responses to my questions. I feel the manuscript has been strengthened significantly by this round of review. This is an interesting and timely study that warrants publication in Nature Communications.

A. Phillip West

Reviewer #2 (Remarks to the Author):

A significant amount of extra work has been performed to generate a much improved manuscript that will be of interest to those interested in the STING pathway, and also those interested in mechanisms of silica-induced lung fibrosis.

Reviewer #3 (Remarks to the Author):

The authors have improved the work in the revision, and addressed several points in a satisfactory manner. However, a number of points are either still not addressed, or the data presented are not convincing.

Original point 5. The quantification of cell death in specific cell types reveals only a very modest effect of cGAS or STING-deficiency. Therefore, these data cannot explain the in vivo observation. Either they should be left out of the work, or the authors should continue and identify the cell type(s) responsible for the bulk of the cGAS/STING dependent cell death.

Original point 6. The authors totally fail to address whether the observed apoptosis and necroptosis contributes to the pathogenesis in vivo, or whether this is driven by conventional cGAS-STING pathways such as type I IFN signaling. This can only be addressed in the mouse system, and preferentially using KO mouse strains (e.g. caspase 3^{-/-}, GSDMD^{-/-}, MLKL^{-/-}, IFNAR^{-/-} mice).

Original point 7. The explanation given is very unspecific, and not satisfactory. (i) Is DNA released in a STING-dependent manner following silica to induce cell death?; (ii) is DNA released in a STING-independent manner following silica to induce STING-dependent cell death?; (iii) a combination of i and ii? This needs to be clarified in a precise manner, and based on conclusive data.

Original point 8. The answer is very confusing, and rather disconnected from the data in the manuscript. The authors refer to a study with a ROS scavenger and caspase 1 dependent cell death. Yet, the data presented in this work show that cell death occurs independent of GSDMD cleavage, hence independent of the inflammasome and caspase 1. Consequently, the paper that the authors refer to provides no valuable information for the present study.

Original point 11. The authors cite the literature incorrectly!!! For most of the studies they refer to, it was not tested whether the response was cGAS dependent, since they were published before cGAS was identified. Moreover, for some of the proposed receptors (e.g. IFI16 and ZCCHC3) it was specifically demonstrated that they act in the cGAS-STING pathway, most likely as co-factors for

cGAS-DNA binding. I find it highly disturbing that the authors miscite the literature. For the present study, the fact that cGAS^{-/-} and STING^{-/-} mice largely phenocopy each other on the in vivo data strongly suggest that the effect is driven by the cGAS-STING pathway. Therefore, the data and discussion on other potential DNA sensors weakens the work.

Response to the Reviewers' comments:

Reviewer #1 (Remarks to the Author):

Thank you for providing detailed responses to my questions. I feel the manuscript has been strengthened significantly by this round of review. This is an interesting and timely study that warrants publication in Nature Communications.

A. Phillip West

We are thankful for these positive comments, and for the original review that helped us strengthen the manuscript. We are very honored to know that they come from a most prestigious scientist that certainly inspired our own work.

Reviewer #2 (Remarks to the Author):

A significant amount of extra work has been performed to generate a much improved manuscript that will be of interest to those interested in the STING pathway, and also those interested in mechanisms of silica-induced lung fibrosis.

We appreciate these positive comments and agree that the extra data included in revision has helped improving the manuscript.

Reviewer #3 (Remarks to the Author):

The authors have improved the work in the revision, and addressed several points in a satisfactory manner. However, a number of points are either still not addressed, or the data presented are not convincing.

Original point 5. *The quantification of cell death in specific cell types reveals only a very modest effect of cGAS or STING-deficiency. Therefore, these data cannot explain the in vivo observation. Either they should be left out of the work, or the authors should continue and identify the cell type(s) responsible for the bulk of the cGAS/STING dependent cell death.*

In Fig 3f-i, cell death in specific cell types was expressed as % of Annexin V⁺PI⁺ cells per population, with 200 000 cells analyzed per cell type. Although the % may seem only modestly affected by cGAS or STING-deficiency, this did not reflect the actual differences in total cell numbers per lung.

We now express the results as absolute numbers of Annexin V⁺PI⁺ cells for each cell population. As seen in the revised Figure 3 f-i, neutrophils are the most prominent cell type with cGAS/STING dependent cell death 7 days after silica exposure:

Revised Fig. 3f-i:

Revised Fig.3f-i legend:

“Bargraphs of the Annexin/PI stained cells among (f) interstitial macrophages, (g) alveolar macrophages, (h) DCs and (i) neutrophils, expressed as absolute cell number per lung.”

We originally focused on CD45⁺CD11b⁺ inflammatory cells that reflect silica-induced lung inflammation, and include phagocytic cells. Silica microparticles also cause an initial stress to the airway tissue, reported as ‘connective tissue breakdown’ (Ref 12: Zay K, Loo S, 1999) with epithelium damage. Indeed, silica induces necrosis of type 1 epithelial cells one day after silica intratracheal (Bowden and Adamson, 1984) and we now further analyzed cell death in lung

epithelial cells, and more broadly in CD45⁻ lung cells, as well as in CD45⁺CD11b⁻ cells that we did not show previously.

We now show that silica exposure induced cell death of CD45⁻ EpCAM⁺ cells, and more broadly of CD45⁻ cells, independently of cGAS/STING pathway, while CD45⁺CD11b⁻ cells were less affected by silica. The fact that silica induced cell death of CD45⁻ and epithelial cells independently from cGAS/STING pathway is in line with the limited influence of cGAS or STING-deficiency on the concentration of dsDNA recovered in the BAL fluid.

Thus among the cell populations analyzed from lung homogenates, neutrophil represents a prominent population responsible for the bulk of the cGAS/STING dependent cell death in response to silica exposure.

The new data on silica induced epithelial cell death in WT, STING^{-/-} or cGAS^{-/-} mice is now presented in the new supplementary Fig 5:

Supplementary Fig. 5 legend:

cGAS/STING-independent epithelial cell death after silica exposure.

Silica microparticles (1 mg/mouse i.t.) or saline vehicle were administered to WT, STING^{-/-} and cGAS^{-/-} mice and parameters analyzed on day 7 post-exposure.

(a-c) Dot plots showing Annexin V/PI staining on CD45⁻CD11b⁻Epcam⁺ epithelial cells (a), CD45⁻ resident cells (b) and CD45⁺CD11b⁻ lymphoid cells (c).

(d-f) Bargraphs of the Annexin/PI stained cells among epithelial cells (d), CD45⁻ cells (e) and CD45⁺CD11b⁻ cells (f), expressed as absolute cell number per lung.

*p < 0.05, **p < 0.01, ***p < 0.001, ****p < 0.0001 (one-way ANOVA with Tukey's *post hoc* test). Data are presented as mean ± SEM and are representative of two independent experiments (mice per group: n= 5 (WT NaCl), n= 4 (WT Silica), n= 5 (STING^{-/-} NaCl), n= 5 (STING^{-/-} Silica), n= 4 (cGAS^{-/-} NaCl), n= 5 (cGAS^{-/-} Silica)). Each symbol represents an individual mouse."

Revised Results section on page 11:

"Silica exposure induced cell death of interstitial and alveolar macrophages and tenfold more neutrophils, as well as early apoptosis in DCs from WT mice, while this response was reduced in the lung of STING^{-/-} and cGAS^{-/-} mice (Fig. 3e-i). Interestingly, silica exposure also induced cell death in epithelial and non-hematopoietic, resident lung cells independently of cGAS/STING pathway (Supplementary Fig. 5)."

Original point 6. *The authors totally fail to address whether the observed apoptosis and necroptosis contributes to the pathogenesis in vivo, or whether this is driven by conventional cGAS-STING pathways such as type I IFN signaling. This can only be addressed in the mouse system, and preferentially using KO mouse strains (e.g. caspase 3^{-/-}, GSDMD^{-/-}, MLKL^{-/-}, IFNAR^{-/-} mice).*

As suggested, we now address this point by characterizing cell death in IFNAR^{-/-} mice:

First, we determined dsDNA release in the airways of IFNAR^{-/-} mice 7 day post silica exposure. There was no difference in the amount of dsDNA present in the BAL of IFNAR^{-/-} and wild-type mice (see new suppl Fig 1o), in line with the DNA levels seen in STING^{-/-} mice (Fig 3d).

New Supplementary Figure 1o:

New Supplementary Figure 1o legend:

"(o-q) Silica microparticles or saline vehicle as in (a) were administered to WT, IFNAR^{-/-} and IL-1R1^{-/-} mice and parameters analyzed on day 7 post-exposure.

(o) Concentration of extracellular dsDNA in the BALF."

As cell death at day 7 may be the net result of the original silica insult plus downstream STING activation, STING-dependent ROS and cell death, we also addressed cell death earlier after silica exposure.

We analyzed early cell death 24hr post-silica exposure in $IFNAR^{-/-}$ mice by flow cytometry. We show a partial reduction of silica-induced cell death in $IFNAR^{-/-}$ mice as compared to wild-type mice, similar to $STING^{-/-}$ mice. However, the amount of dsDNA recovered in the BALF of these mice one day post-silica was similar to wild-type mice. Both caspase 3 cleavage and MLKL phosphorylation were detected in the absence of $IFNAR$ 24hr post silica exposure.

The new data on early silica induced cell death in $IFNAR^{-/-}$ mice is now presented in the new supplementary Fig. 6:

New Supplementary Figure 6 legend:

“Partial contribution of Type I IFN signaling in early silica induced lung cell death.

Silica microparticles (1 mg/mouse i.t.) or saline vehicle were administered to WT, $IFNAR^{-/-}$ and $STING^{-/-}$ mice and parameters analyzed on day 1 post-exposure.

(a) Gating strategy showing the exclusion of doublet cells using FSC-H/FSC-A, SSC-H/SSC-A followed by FSC-W/FSC-A.

(b) Dot plots showing Ann V⁺/PI⁺ dead lung cells.

(c) Absolute numbers of Ann V⁺/PI⁺ cells per lung.

(d) Concentration of extracellular dsDNA in the BALF.

(e) Immunoblots of caspase 3, cleaved caspase 3, phospho-MLKL, MLKL and β -actin as a reference.

* $p < 0.05$, ** $p < 0.01$, *** $p < 0.001$, **** $p < 0.0001$ (one-way ANOVA with Tukey’s *post hoc* test). Data are presented as mean \pm SEM (mice per group: $n = 3$ (WT NaCl), $n = 4$ (WT Silica), $n = 3$ ($IFNAR^{-/-}$ NaCl), $n = 4$ ($IFNAR^{-/-}$ Silica), $n = 4$ ($STING^{-/-}$ NaCl), $n = 4$ ($STING^{-/-}$ Silica)). Immunoblots representative of $n = 2$ samples. Each symbol represents an individual mouse.”

Revised results section on page 11:

“We further addressed the involvement of type I IFN pathway in early silica induced cell death. Lung cell death was partially reduced in *IFNAR*^{-/-} and *STING*^{-/-} mice 24h post-silica exposure (Supplementary Fig 6a-c), although there was already a *STING*/*IFN*I-independent release of dsDNA in the airways (Supplementary Fig 6d). At this early stage caspase 3 cleavage and MLKL phosphorylation occurred in the absence of *IFN*I pathway (Supplementary Fig 6e).”

As suggested, we now further addressed the role of Gasdermin D pathway *in vivo* by exposing Gasdermin D deficient mice to silica:

We show a reduction of cell death in the *GSDMD*^{-/-} mice for interstitial and alveolar macrophages and dendritic cells, 7 days post silica exposure (new Supplementary Figure 8). However this did not translate into reduced dsDNA in the BAL, and the inflammatory response was preserved of silica-exposed *GSDMD*^{-/-} mice, in terms of neutrophil recruitment to the alveolar space, pulmonary TNF, CXCL10, *Ifnb* or *Tmem173* expression. Both caspase 3 cleavage and MLKL phosphorylation were detected in the lung of *GSDMD*^{-/-}.

Thus in the absence of Gasdermin D, while there is a reduction in macrophage and dendritic cell death, the *IFN*I response and inflammatory responses to silica exposure are preserved.

This new data is now presented in the new Supplementary Fig. 8:

New Supplementary Figure 8 legend:

“**Silica induced lung inflammation in the absence of Gasdermin D.**

Silica microparticles (1 mg/mouse i.t.) or saline vehicle were administered to WT and *GSDMD*^{-/-} mice and parameters analyzed on day 7 post-exposure.

(a) Dot plots showing Annexin V/PI staining on interstitial macrophages (IM ϕ), alveolar macrophages (AM ϕ) and dendritic cells (DCs).
(b-d) Proportion of Ann V⁺/PI⁺ IM ϕ , AM ϕ and DCs.
(e) Concentration of extracellular dsDNA in the BALF.
(f) Neutrophils count in the BALF. Lung (g) TNF and (h) CXCL10 levels measured by ELISA, and (i) *Ifn β* and (j) *Tmem173* transcripts measured by real-time PCR.
(k) Immunoblots of caspase 3, cleaved caspase 3, phospho-MLKL, MLKL and β -actin as a reference.
* $p < 0.05$, ** $p < 0.01$, *** $p < 0.001$, **** $p < 0.0001$ (one-way ANOVA with Tukey's *post hoc* test). Data are presented as mean \pm SEM (mice per group: $n = 5$ (WT NaCl), $n = 5$ (WT Silica), $n = 2$ (GSDMD^{-/-} NaCl), $n = 5$ (GSDMD^{-/-} Silica)). Immunoblots are representative of $n = 2$ samples. Each symbol represents an individual mouse.”

Revised results section on page 12:

“Although inflammatory cell death was reduced in GSDMD^{-/-} mice, dsDNA release and lung inflammation were not affected 7 days post silica exposure (Supplementary Fig.8).”

Original point 7. *The explanation given is very unspecific, and not satisfactory. (i) Is DNA released in a STING-dependent manner following silica to induce cell death?; (ii) is DNA released in a STING-independent manner following silica to induce STING-dependent cell death?; (iii) a combination of i and ii? This needs to be clarified in a precise manner, and based on conclusive data.*

We are grateful for insisting on the clarification.

The data points to (iii), a combination of i and ii:

(ii) *In vivo*, DNA can be released in a STING-independent manner, based on the STING-independent epithelial cell death after silica exposure (new suppl Fig. 5) and on the unaffected levels of dsDNA in the BAL of silica exposed STING^{-/-} mice (Fig 3i) or dsDNA in IFNAR^{-/-} mice (new suppl Fig. 1o). DNA release can occur very rapidly, based on the 1 μ g/mL dsDNA concentration present in the BAL 24h post silica exposure (Suppl Fig 1h), as compared to the 2 μ g/mL dsDNA seen at day 7 (Fig 1a) or day 28 (suppl Fig.1a). This released dsDNA is then able to induce an early STING dependent response with pulmonary IFN α and IFN β production at 24h, absent in STING^{-/-} mice (Suppl Fig. 3a), plus further STING-dependent cell death, as seen in neutrophils, macrophages and DCs (Fig. 3f-i), and based on the reduction of cCasp3, cGSDMD, pMLKL in the lung of STING^{-/-} mice after silica exposure (Fig. 3j).

(i) Using STING^{-/-} mice we show that DNA can be released in a STING-dependent manner following silica induced cell death, *in vitro* in macrophages (Figure 4c: cell death, d: released DNA, e: cleaved Casp3 and GSDMD, pMLKL) and in dendritic cells (Fig 5b: cell death, c: released DNA, d: cleaved Casp3 and GSDMD, pMLKL), but also *in vivo* in macrophages, dendritic cells or neutrophils (Figure 3d-h: cell death, i: released DNA, j: cleaved Casp3 and GSDMD, pMLKL). This is likely mediated by the induction of ROS by macrophages (Fig 4a) and DCs (Fig 5a), as documented with reduced silica-induced macrophage cell death after ROS scavenger NAC treatment *in vitro* (new suppl Fig.10). In addition, we now document a reduction of silica-induced lung cell death, based on reduced cCasp3 and pMLKL, after DNase I treatment to eliminate extracellular DNA *in vivo* (revised Figure 1p).

(iii) Thus, based on the detailed analysis provided, we trust that there is a combination of i and ii. We have now clarified this point in the Discussion on page 23.

Revised Figure 1p:

Revised Figure 1p legend:

“(p) Immunoblots of STING/IRF3 axis in lung homogenates including phospho-STING, STING, phospho-TBK1, TBK1, phospho-IRF3, IRF3, caspase 3 (Casp3), cleaved Caspase 3 (c-Casp3), MLKL and phosphorylated MLKL (p-MLKL), with β -actin as a reference.”

Revised results section on page 7:

“Silica-induced cell death was reduced by DNase I treatment, in terms of caspase 3 cleavage and MLKL phosphorylation (Fig. 1 p).”

Revised Discussion section on page 23:

“Indeed, silica microparticles cause an initial stress to the airways tissue, with ‘connective tissue breakdown’, lung epithelial cell death and dsDNA release in the airway space independently of STING pathway, and the persistence of microcrystals in the lung can ‘entertain’ this original cellular stress^{12, 54}. Our results indicate that this first response of lung epithelial cell death to silica microparticles is cGAS/STING independent, while a cGAS/STING dependent inflammatory response is induced by the released DNA, inducing ROS and further cell death.”

Original point 8. *The answer is very confusing, and rather disconnected from the data in the manuscript. The authors refer to a study with a ROS scavenger and caspase 1 dependent cell death. Yet, the data presented in this work show that cell death occurs independent of GSDMD cleavage, hence independent of the inflammasome and caspase 1. Consequently, the paper that the authors refer to provides no valuable information for the present study.*

We referred to three studies with the ROS inhibitor NAC in the context of silica exposure. Indeed, the first study addresses the role of ROS scavenger in silica-induced activation of NLRP3 inflammasome, thus using caspase 1 as a read-out (Bauernfeind et al., 2011).

However the *in vivo* study by Zhang *et al.* shows that oral treatment with high-dose NAC during early silica exposure reduced lung fibrosis, IL-8 in BALF, and TNF in serum (Zhang et al., 2013). Further, Zhang et al. exploring the effects of NAC on silica-induced pulmonary fibrosis in rats showed reduced ROS contents, mitochondrial transmembrane potential changes of AM, and reduced lung fibrotic lesions after NAC treatment (Zhang et al., 2014). These authors concluded that the mitochondrial apoptosis pathway was also inhibited by the NAC treatment.

We now tested the effect of ROS scavenger NAC to address more directly the role of silica-induced ROS formation in STING-dependent cell death. Indeed, we showed that ROS formation itself depends of functional STING pathway (Fig. 4a), as ROS formation, associated with mitochondria in silica exposed macrophages was abolished in STING^{-/-} macrophages. This was associated with macrophage cell death (Fig. 4c,e) and dsDNA release (Fig. 4d).

To address more directly the causality link, i.e. whether ROS are needed for silica-induced STING-dependent cell death, we treated macrophages with ROS scavenger NAC prior to silica exposure and analyzed silica-induced macrophage cell death. We now show that silica-induced macrophage cell death was partially prevented by NAC.

This new data is now presented in the new Supplementary Fig. 10:

New Supplementary Figure 10 legend:

“ROS scavenger inhibits macrophages death after *in vitro* silica exposure.

Bone marrow-derived macrophages from WT mice were treated with N-acetyl cysteine (NAC) during 1h prior to silica exposure (250µg/mL) for 18 hr. The purity of the macrophage population was verified by flow cytometry using CD11b, CD11c and F4/80 staining: 99.7±0.5% CD11b⁺ F4/80⁺ CD11c⁻ cells (a).

(b-c) Dot plots showing Annexin V/PI staining and (c) proportion of Ann V⁺/PI⁺ macrophages in the different groups.

p < 0.01, *p < 0.001, ****p < 0.0001 (Mann-Whitney U analysis). Data are presented as mean ± SD of n= 2 independent cultures.”

Revised Results section on page 13:

“Macrophage cell death after silica exposure was dependent on ROS induction as it could be partially prevented in the presence of ROS scavenger (Supplementary Figure 10).”

Original point 11. *The authors cite the literature incorrectly!!! For most of the studies they refer to, it was not tested whether the response was cGAS dependent, since they were published before cGAS was identified. Moreover, for some of the proposed receptors (e.g. IFI16 and ZCCHC3) it was specifically demonstrated that they act in the cGAS-STING pathway, most likely as co-factors for cGAS-DNA binding. I find it highly disturbing that the authors miscite the literature. For the present study, the fact that cGAS^{-/-} and STING^{-/-} mice largely phenocopy each other on the in vivo data strongly suggest that the effect is driven by the cGAS-STING pathway. Therefore, the data and discussion on other potential DNA sensors weakens the work.*

First, we apologize for indicating that different DNA sensors activate type I IFN pathway in a cGAS independent manner. This was a wrong shortcut. We meant to list non-cGAS DNA sensors reported to act with STING to trigger type I IFN. As pointed by the reviewer, reference 36 (Wu et al., Science 2013) and reference 39 (Kondo et al., PNAS 2013) were published in early 2013, essentially at the time the first cGAS major articles were published. Indeed, IFI16 was reported to cooperate with cGAS (Ref 40, 41), as a cofactor (Ref 42), and ZCCHC3 as a cosensor for cGAS (Ref 43). We modified the sentence in the Discussion section on page 21.

In the present study, we agree that the fact that cGAS^{-/-} and STING^{-/-} mice largely phenocopy each other 7 to 35 days after silica exposure *in vivo* strongly suggests that this effect is driven by the cGAS-STING pathway. However, we showed differences in the early response to silica between STING^{-/-} and cGAS^{-/-} mice at 24h post-silica exposure *in vivo* (Supplementary Fig.3a). This is why we started to look for other potential DNA sensors that may be involved in silica induced inflammation. In particular, the early silica-induced lung production of IFN α and IFN β was abolished in STING^{-/-} mice while IFN α was not affected and IFN β only partially reduced in cGAS^{-/-} mice on day 1 post silica. The difference between STING^{-/-} and cGAS^{-/-} mice was also seen for lung CXCL10, IL-1b and TNF α , and for the neutrophil recruitment in the airways on day 1 post silica. This suggests that the early response to silica might involve other DNA sensors, beside cGAS (as mentioned on page 10).

Further, the fact that cGAS^{-/-} macrophages respond to silica exposure as wild-type macrophages while STING^{-/-} macrophages have a completely impaired response to silica (Figure 7) suggested that other DNA sensors beside cGAS are involved in the response to silica. We wished to give a first attempt at identifying the potential alternative DNA sensors involved in the response to silica (suppl Fig 7). Additional work will clearly be needed in future investigations to further settle these points.

Revised Discussion section on page 21:

“The main cytosolic DNA sensors upstream of STING identified is cGAS, and other molecules contributing to cytosolic DNA sensing and type I IFN pathway activation in a STING-dependent manner include DDX41, IFI16 or mouse IFI204, DAI, RNA pol III, LRRFIP1, ZCCHC3 or meiotic recombination 11 homolog A (MRE11), IFI16 and IFI204 cooperating with cGAS, or ZCCHC3 acting as co-factors of cGAS^{36, 37,38,39, 40, 41, 42, 43, 44}”

Reference

Bowden DH, Adamson IY. The role of cell injury and the continuing inflammatory response in the generation of silicotic pulmonary fibrosis. *J Pathol.* 144:149-61 (1984)

REVIEWERS' COMMENTS:

Reviewer #3 (Remarks to the Author):

I think the authors have now addressed most of my points in a satisfactory manner.